# GraphCliff: Short-Long Range Gating for Subtle Differences but Critical Changes

## Abstract

Quantitative structure–activity relationship assumes a smooth relationship between molecular structure and biological activity. However, activity cliffs defined as pairs of structurally similar compounds with large potency differences break this continuity. Recent benchmarks targeting activity cliffs have revealed that classical machine learning models with extended connectivity fingerprints outperform graph neural networks. Our analysis shows that graph embeddings fail to adequately separate structurally similar molecules in the embedding space, making it difficult to distinguish between structurally similar but functionally different molecules. Despite this limitation, molecular graph structures are inherently expressive and attractive, as they preserve molecular topology. To preserve the structural representation of molecules as graphs, we propose a new model, GraphCliff, which integrates short- and long-range information through a gating mechanism. Experimental results demonstrate that GraphCliff consistently improves performance on both non-cliff and cliff compounds. Furthermore, layer-wise node embedding analyses reveal reduced over-smoothing and enhanced discriminative power relative to strong baseline graph models.

## 1 Introduction

Quantitative Structure–Activity Relationship (QSAR) is based on the premise that molecules with similar structures have similar biological activity. QSAR modeling plays a crucial role in drug discovery as it reduces the number of compounds that require experimental testing, thereby saving both cost and time. In particular, QSAR-guided drug discovery enables virtual screening for hit identification, lead optimization, and ADMET (absorption, distribution, metabolism, excretion, and toxicity) evaluation, thus streamlining the experimental workflow (Cherkasov et al., 2014). To support such virtual screening efforts, a wide range of machine learning and deep learning models have recently been developed to directly predict molecular properties and biological activities from molecular structures (Hu et al., 2019; Wang et al., 2022; Heid et al., 2023; Li et al., 2023; Qiao et al., 2025). However, there exists a class of cases that breaks the continuity of the typical structure–activity relationship, known as *activity cliffs*. Unlike the conventional assumption that structurally similar molecules exhibit similar activities, activity cliffs describe cases where minor structural differences lead to large and abrupt changes in activity. They are formally quantified as the ratio of the activity difference between two compounds to their distance in a given chemical space (Maggiora, 2006). In practical terms, activity cliffs are defined as pairs or groups of structurally similar compounds that are active against the same target protein but exhibit large potency differences (Stumpfe et al., 2019). Although analog groups corresponding to activity cliffs may deviate from general QSAR assumptions, they highlight the importance of local structural changes and provide valuable insight into processes such as hit-to-lead optimization and structural alert development (Stumpfe & Bajorath, 2012; Wedlake et al., 2019).

Motivated by activity cliffs' importance in drug discovery, Van Tilborg et al. (2022) curated the MoleculeACE dataset from ChEMBL (Gaulton et al., 2012) and evaluated a wide range of models. The results revealed that machine learning models with extended connectivity fingerprints (ECFPs) consistently outperform deep learning approaches, with CNNs and LSTMs using SMILES providing moderate success, while transformer and GNNs generally underperformed. The strong performance of ECFPs can be attributed to their design, in which binary bit vectors represent radius-based substructures that are highly sensitive to chemical modifications (Rogers & Hahn, 2010). This represen-

tation introduces a strong inductive bias and low variance, expanding atom-centered neighborhoods within a fixed radius and hashing them into sparse vectors that suppress noise and yield stable encodings. In small-data regimes, such inductive bias allows ECFP-based models to generalize more reliably than flexible deep models. In contrast, GNNs introduce numerous parameters and high modeling flexibility, which increase variance under limited data (Baptista et al., 2022). Moreover, as layers deepen, node embeddings become homogenized due to Laplacian smoothing, leading to the over-smoothing phenomenon where fine-grained local distinctions vanish (Wu et al., 2023). This explains why LSTM and CNN models, which emphasize local structural changes, often perform better than transformer and GNNs. Nevertheless, molecular graphs inherently preserve rich structural information, where atoms are represented as nodes and bonds as edges, with extensions to 3D coordinates, charges, or bond orders directly incorporated (Kearnes et al., 2016). Unlike ECFPs that rely on predefined radius-based hashing, graph representations can adaptively capture complex topological patterns, stereochemistry, and long-range dependencies. The central challenge is therefore to design graph architectures that preserve the expressiveness of molecular graph structures while mitigating over-smoothing and achieving ECFP-level sensitivity to local patterns.

To confirm that GNNs have difficulty preserving the same level of local sensitivity as ECFPs, we performed an analysis based on the MoleculeACE results, comparing the ability of ECFPs and graph embeddings to capture local structural changes within activity cliff pairs. For ECFPs, each molecule in a cliff pair was represented as a 1024-dimensional fingerprint, and the dissimilarity between a pair of molecules was measured as $1 - \mathrm{TanimotoSimilarity}(A, B)$, where $A$ and $B$ denote the ECFPs of the two molecules in the pair. For graph embeddings, we extracted embeddings from graph-based models for each molecule in a cliff pair and calculated the Euclidean distance (Liberti et al., 2014) between them. To ensure a fair comparison, we applied min–max normalization separately to the ECFP dissimilarities and graph embedding Euclidean distances, scaling each to the range $[0, 1]$. Appendix Figure 6 compares ECFP dissimilarities (x-axis) and graph embedding Euclidean distances (y-axis) for activity cliff pairs, with the diagonal line $y = x$ (red) serving as a reference. If the two measures were similar, the points would align closely with this line. However, most points lie below the diagonal for GCN, GAT, and MPNN, indicating that ECFP dissimilarities tend to be larger than the corresponding graph embedding distances. The fitted regression lines (green) further confirm this trend, with slopes below 1, indicating that ECFPs capture larger bit-level differences between cliff pairs. Thus, ECFPs are more sensitive to local structural changes than graph embeddings. The slopes computed for each model across all individual datasets are reported in Appendix Table 2.

These findings highlight a critical limitation of existing GNNs: despite their expressive capacity, they fail to preserve ECFP-level sensitivity to local structural changes. This limitation motivates the need for graph models that can preserve the inherent expressiveness of molecular graph structures while matching the local sensitivity of ECFPs. Therefore, we aimed to create a graph-based model that effectively integrates global context with local structural details. Similar efforts to combine local and global dependencies have also been explored in sequence modeling. StripedHyena2 (Ku et al., 2025) is a multi-hybrid sequence architecture that extends the original Hyena long convolution by introducing short explicit (SE), middle regularized (MR), and long implicit (LI) convolutional components to jointly capture short-, middle-, and long-range dependencies. While StripedHyena2 operates on 1D token sequences, we adapt the same principle to molecular graphs by combining short-range message passing layers with long-range propagation modules. This design shares the same goal of capturing both short- and long-range information through a gating mechanism that selectively integrates local features with global context. Our contributions are as follows:

- We introduce **GraphCliff**, a novel graph neural architecture that explicitly integrates local structural details and global context through a gating mechanism over short- and long-range representations, with the explicit goal of overcoming the loss of local sensitivity and over-smoothing issues observed in existing GNNs.

- We provide extensive empirical evidence on the benchmark, demonstrating **consistent improvements on both non-cliff and activity cliff compounds**.

- We present a comprehensive analysis which shows that our model mitigates over-smoothing in node representations, yielding **more discriminative representations** than existing GNNs.

## 2 RELATED WORKS

**Contextual dependencies at varying ranges**   Modeling dependencies across multiple contextual ranges is essential for tasks that require both fine-grained local detail and broad long-range coherence. StripedHyena2 addresses this challenge with a convolution-centric architecture that operates entirely on 1D convolutional modules optimized for sequence modeling. StripedHyena used the transformed Hyena block (Poli et al., 2023), which was converted into short-, middle-, and long-range variants that are sequentially connected to capture information across multiple scales. These variants are implemented as three convolutional operators with distinct receptive fields: Short-Explicit (SE), Medium-Regularized (MR), and Long-Implicit (LI), which are combined into sequential compositions such as SE–MR–LI. Each module is specialized to capture a different scale of interaction. SE focuses on local recall through short explicit filters and has been empirically shown to be particularly effective at capturing short-range dependencies. MR models medium-range interactions using regularized filters. LI aggregates information across the entire sequence via implicit long convolutions. This hierarchical design is particularly advantageous for ultra-long sequence domains such as genomic data, and has been shown to scale to contexts of up to one million tokens. To unify representations obtained at different scales, StripedHyena2 employs a learnable gating mechanism that adaptively balances short-, middle- and long-range features. The gating formulation allows the model to dynamically adjust the contribution of local versus global signals, thereby preserving critical short-range information while maintaining coherence across long-range contexts.

**Activity cliff**   Stumpfe et al. (2019) established two key criteria to enable a systematic and quantitative investigation of activity cliffs. The first criterion concerns the structural similarity between two compounds, while the second considers the magnitude of their potency difference. To define analog groups, the authors employed the concept of Matched Molecular Pairs (MMPs), identifying pairs of molecules with single or multiple substitution sites that exhibit changes in potency (or $\Delta$pKi) greater than 2. In another study, Van Tilborg et al. (2022) formalized the concept of activity cliffs by constructing a curated dataset from ChEMBL specifically designed for activity cliff analysis. Structural similarity was quantified using three complementary measures: (i) substructure similarity, computed as the Tanimoto coefficient on ECFPs to capture shared radial, atom-centered substructures between molecules, thereby reflecting global differences across their entire substructural composition, (ii) scaffold similarity, based on ECFPs computed on molecular scaffolds, to detect compounds differing in their core structures, and (iii) SMILES similarity, measured via Levenshtein distance, to account for character insertions, deletions, and translocations in the string representation of molecules. Activity cliffs were then defined as compound pairs with at least a 10-fold difference in Ki. The benchmark further evaluated a broad range of models, including deep learning approaches such as graph-based methods, Attentive Fingerprint (AFP) (Xiong et al., 2019), Graph Attention Networks (GAT) (Veličković et al., 2017), Graph Convolutional Networks (GCN) (Kipf, 2016), and Message Passing Neural Networks (MPNN) (Gilmer et al., 2017), as well as Convolutional Neural Networks (CNN) (Kimber et al., 2021), Long Short-Term Memory networks (LSTM) (Hochreiter & Schmidhuber, 1997), Multilayer Perceptrons (MLP), and Transformer (Vaswani et al., 2017). In addition, traditional machine learning algorithms were assessed, including Gradient Boosting Machines (GBM) (Friedman, 2001), k-Nearest Neighbors (KNN) (Fix, 1985), Random Forests (RF) (Predictors, 1996), and Support Vector Machines (SVM) (Cristianini, 2000).

## 3 METHODS

### 3.1 DATASETS

We utilized MoleculeACE as benchmark, which comprises curated compound–protein interaction data extracted from ChEMBL. Each dataset contains potency values (Ki) for compounds targeting a specific protein, where low-quality samples were removed during curation based on predefined criteria. In total, the benchmark includes 30 datasets, each corresponding to a distinct protein target, where Ki values serve as regression labels. Each of the 30 datasets in MoleculeACE is associated with a different protein target and can be used independently to assess model generalization across diverse biological contexts. In addition to MoleculeACE, we also employed the benchmark datasets introduced by  Group (2023). The Low-Sample Size and Narrow Scaffold (LSSNS) datasets con-

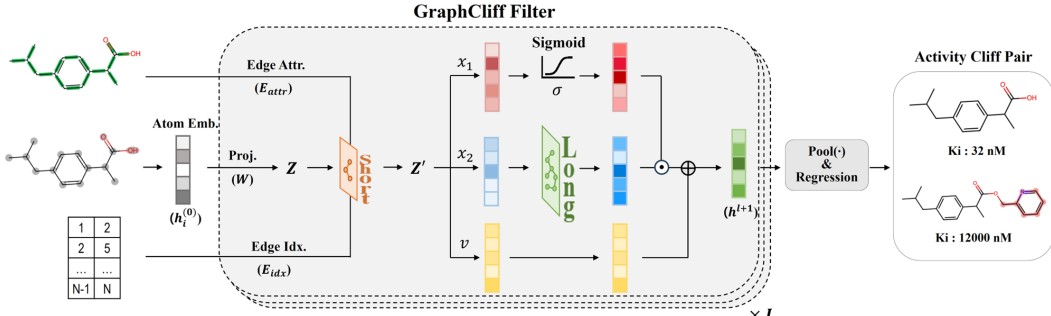

Figure 1: Overall architecture of GraphCliff.

sist of small molecules built around highly conserved scaffolds, with each dataset containing ranging from a few dozen to slightly over one hundred compounds. These data were compiled from fragment-to-lead medicinal chemistry studies and the ChEMBL database. As the LSSNS collection does not provide pre-defined activity cliff annotations, we applied the same criteria used in MoleculeACE to annotate cliff molecules. Structural similarity between compounds was defined using three complementary metrics: substructure similarity, scaffold similarity, and SMILES string similarity. Two compounds were considered structurally similar if at least one of these similarity scores exceeded 0.9. If their Ki values differed by more than the predefined threshold (i.e., at least a 10-fold difference), the pair was regarded as exhibiting a significant potency change. Compound pairs that satisfied both criteria were designated as activity cliffs. In the benchmark, activity cliffs were encoded in a binary manner, indicating whether each compound belongs to a cliff, without explicitly enumerating pairs or groups. Train–test splits were constructed to preserve the overall ratio of activity cliffs across datasets. Additional details are provided in Appendix Tables 3 and 4.

## 3.2 SHORT- AND LONG-RANGE GATING

We take inspiration from StripedHyena2, which transforms the Hyena block into short-, middle-, and long-range variants that are sequentially connected to capture information across multiple scales. Extending this design principle to molecular graphs, we construct a graph architecture where conventional graph modules are adapted to process short- and long-range dependencies and their outputs are fused through a learnable gating mechanism. An overview of this architecture is illustrated in Figure 1, and the formulation of our model is presented in the following equations.

**Problem Setup and Notation** We represent each molecule as a graph $G = (V, E)$, where $V$ is the set of atoms and $E$ is the set of chemical bonds. Let $N = |V|$ and $M = |E|$ denote the numbers of atoms and bonds, respectively. Each atom is associated with an input feature vector, forming the node feature matrix $x \in \mathbb{R}^{N \times d_{\text{in}}}$. Bond connectivity is represented by the edge index $E_{idx} \in \mathbb{R}^{M \times 2}$, and each bond carries an edge attribute vector stored in $E_{attr} \in \mathbb{R}^{M \times d_{attr}}$. Given a molecular graph $G$, the learning objective is to predict a target property $y$ through a parametric mapping $f(G; \theta)$. Unless otherwise stated, we use $d$ to denote the hidden dimensionality of intermediate node embeddings.

**Atom encoding** Given the node feature matrix $x$, the model first maps each atom to a $d$-dimensional hidden representation via $h^{(0)} = \phi_{\text{atom}}(x) \in \mathbb{R}^{N \times d}$, where $\phi_{\text{atom}}$ denotes an MLP followed by normalization and a nonlinear activation function.

**GraphCliff filter** Adopted from StripedHyena2, in which a single projection yields three separate components (input, gating, and output), our projection layer maps $d$-dimensional space into a $3d$-dimensional space to facilitate decomposition into functionally distinct streams. Each filter layer processes short- and long-range information, and their outputs are subsequently integrated via a gating mechanism. The short-range filter adopts a GINE (Xu et al., 2018a) message passing operator to capture local neighborhood interactions. In contrast, the long-range filter captures

multi-hop dependencies within a single layer using Chebyshev polynomials (Hammond et al., 2011), thereby avoiding the need to stack multiple GNN layers. Recent work demonstrates that Chebyshev polynomials operate directly on the normalized Laplacian, propagating information across multiple hops without explicit edge rewiring or architectural modifications that distort the original topology (Hariri et al., 2025b).

At layer $\ell$, given hidden node representations $h^{(\ell)} \in \mathbb{R}^{N \times d}$, we first apply normalization followed by a linear projection:

$$Z = h^{(\ell)}W, \quad Z \in \mathbb{R}^{N \times 3d}, \tag{1}$$

where $W \in \mathbb{R}^{d \times 3d}$ is a trainable projection matrix.

**SHORT FILTER** The short-range filter applies a GINE message passing operator to the projected features $Z$:

$$Z' = \text{GINE}(Z, E_{idx}, E_{attr}), \tag{2}$$

where $E_{idx}$ denotes edge index and $E_{attr}$ denotes edge features. The output $Z' \in \mathbb{R}^{N \times 3d}$ is then split along the feature dimension into three parts:

$$Z' = [\, x_2 \parallel x_1 \parallel v \,], \quad x_2, x_1, v \in \mathbb{R}^{N \times d}. \tag{3}$$

The GINE operator is defined as:

$$z'_i = \psi \left( (1 + \epsilon)z_i + \sum_{j \in \mathcal{N}(i)} (z_j + \phi(e_{ij})) \right), \tag{4}$$

where $e_{ij}$ denotes the edge attribute associated with the directed edge from source node $j$ to target node $i$, $\phi$ is an MLP applied to edge attributes, $\psi$ is a node-wise MLP, and $\epsilon$ is a learnable scalar parameter. Since $Z \in \mathbb{R}^{N \times 3d}$, each node embedding $z_i$ and $z_j$ lies in $\mathbb{R}^{3d}$, and the edge MLP $\phi$ maps $e_{ij}$ into the same $3d$-dimensional space to ensure dimensional consistency in the aggregation term.

**LONG FILTER** To capture global context, we compute Chebyshev polynomials over the normalized adjacency matrix $\hat{A}$, applied to the short-path feature $x_2$:

$$T_0 = x_2, \quad T_1 = \hat{A}x_2, \quad T_k = 2\hat{A}T_{k-1} - T_{k-2} \; (k \geq 2). \tag{5}$$

The long-range module computes $\text{Long}(x_2) = \sum_{k=0}^{K} \alpha_k T_k$, where $\alpha_k$ are learnable coefficients.

**GATED FUSION** We combine short- and long-range information using a sigmoid gating function:

$$g = \sigma(x_1), \qquad u = g \odot \text{Long}(x_2) + v, \tag{6}$$

where $\sigma$ denotes the element-wise sigmoid function and $\odot$ is the element-wise product. Gating mechanisms have been shown to alleviate over-smoothing in GNNs by adaptively regulating information flow (Xin et al., 2020), providing empirical support for our design. Finally, the filter layer output is updated via a residual connection as $h^{(\ell+1)} = u^{(\ell)} + h^{(\ell)}$.

We stack $L$ GraphCliff filters sequentially, where the output of each layer serves as the input to the next:

$$h^{(\ell+1)} = \text{GraphCliffFilter}^{(\ell)}\big(h^{(\ell)}, E_{idx}, E_{attr}\big), \qquad \ell = 0, \dots, L-1. \tag{7}$$

**Pooling and regression** Finally, we apply an attention-based graph pooling operation to adaptively select and aggregate informative nodes. Specifically, we employ SAGPool (Lee et al., 2019), and the resulting pooled representation is passed to a regression head to produce the final output corresponding to the target property. Formally, after obtaining the final layer node embeddings $h^{(L)}$, the graph-level representation is constructed as:

$$\hat{y} = \phi_{\text{reg}}\Big( \text{SAGPool}\Big(h^{(L)}\Big) \Big), \tag{8}$$

where the graph-level representation is obtained via SAGPool and subsequently passed to the regression MLP $\phi_{\text{reg}}$.

Table 1: RMSE ($\downarrow$) and RMSE$_{\text{cliff}}$ ($\downarrow$) values for each algorithm on six ChEMBL targets. The best results are highlighted in **bold**, and the second-best results are underlined.

| Algorithm | Descriptor | CHEMBL1871 (Ki) | CHEMBL204 (Ki) | CHEMBL2147 (Ki) | CHEMBL228 (Ki) | CHEMBL239 (EC50) | CHEMBL244 (Ki) |
|---|---|---|---|---|---|---|---|
| GraphCliff | GRAPH | **0.628** / 0.797 | **0.691** / **0.821** | **0.560** / **0.579** | **0.651** / **0.674** | **0.670** / **0.792** | **0.668** / **0.752** |
| SVM | ECFP | 0.665 / 0.873 | 0.723 / 0.859 | 0.576 / 0.580 | 0.662 / 0.676 | 0.678 / 0.819 | 0.715 / 0.797 |
| GINE + PairNorm | GRAPH | 0.630 / 0.867 | 0.745 / 0.898 | 0.693 / 0.679 | 0.693 / 0.714 | 0.733 / 0.864 | 0.786 / 0.859 |
| GINE + NodeNorm | GRAPH | 0.672 / 0.834 | 0.738 / 0.863 | 0.779 / 0.696 | 0.702 / 0.756 | 0.693 / 0.823 | 0.758 / 0.836 |
| GINE + Residual | GRAPH | 0.678 / **0.781** | 0.800 / 0.938 | 0.711 / 0.733 | 0.708 / 0.793 | 0.709 / 0.839 | 0.765 / 0.815 |
| Chemprop | GRAPH | 0.704 / 0.919 | 0.811 / 0.851 | 0.649 / 0.639 | 0.670 / 0.695 | 0.819 / 0.827 | 0.726 / 0.797 |
| LSTM | SMILES | 0.662 / 0.850 | 0.822 / 0.930 | 0.647 / 0.725 | 0.779 / 0.884 | 0.765 / 0.905 | 0.800 / 0.913 |
| MLP | ECFP | 0.737 / 0.958 | 0.815 / 0.962 | 0.723 / 0.704 | 0.755 / 0.757 | 0.756 / 0.901 | 0.796 / 0.850 |
| GINE + DropEdge | GRAPH | 0.770 / 0.954 | 0.866 / 0.978 | 0.785 / 0.802 | 0.810 / 0.858 | 0.792 / 0.930 | 0.847 / 0.910 |
| SCAGE (w/o 3D) | GRAPH | 0.758 / 0.823 | 0.814 / 0.888 | 0.910 / 0.858 | 0.771 / 0.786 | 0.796 / 0.851 | 0.892 / 0.972 |
| GCN | GRAPH | 0.769 / 1.009 | 1.056 / 1.201 | 0.840 / 0.825 | 0.958 / 1.000 | 0.906 / 1.024 | 1.075 / 1.060 |
| GAT | GRAPH | 0.798 / 1.042 | 1.138 / 1.281 | 0.966 / 0.917 | 1.026 / 1.028 | 0.902 / 1.012 | 1.088 / 1.117 |
| MPNN | GRAPH | 1.058 / 1.154 | 1.458 / 1.581 | 1.025 / 0.934 | 1.000 / 1.015 | 1.288 / 1.481 | 1.660 / 1.557 |
| MolCLR_gcn | GRAPH | 0.948 / 0.863 | 1.592 / 1.616 | 1.551 / 1.545 | 1.340 / 1.317 | 0.968 / 1.025 | 1.831 / 1.837 |
| MolCLR_gin | GRAPH | 1.077 / 1.020 | 1.689 / 1.709 | 1.226 / 1.015 | 1.459 / 1.364 | 1.054 / 1.112 | 1.849 / 1.837 |
| Contextpred | GRAPH | 1.647 / 1.687 | 2.012 / 2.269 | 1.295 / 1.857 | 1.663 / 1.803 | 1.893 / 2.168 | 1.922 / 2.056 |
| KPGT | GRAPH | 1.976 / 1.822 | 2.302 / 2.210 | 1.676 / 1.201 | 1.856 / 1.847 | 2.751 / 2.660 | 2.126 / 2.103 |

# 4 RESULTS

==The section has been revised.==

We evaluated our method on all 30 benchmark datasets provided by MoleculeACE. As baselines, we included the machine learning and deep learning models reported in the original MoleculeACE study. These include graph-based models (AFP, GAT, GCN, MPNN), SMILES-based models (CNN, LSTM, Transformer), and ECFP-based models (MLP, GBM, KNN, RF, SVM). To provide a comprehensive evaluation, we incorporated additional models known for their strong performance in molecular property prediction. **ContextPred** (Hu et al., 2019) is a pretraining method that learns to predict masked subgraphs using contextual information, thereby enhancing structural awareness. **MolCLR** (Wang et al., 2022) applies contrastive learning to molecular graphs, encouraging structurally similar molecules to be mapped closer in the learned embedding space. **Chemprop** (Heid et al., 2023) is based on a message passing neural network (MPNN) architecture that incorporates directed edge information (D-MPNN). **KPGT** (Li et al., 2023) is a knowledge-guided pretraining method designed to integrate domain-specific chemical insights into the representation learning process. **SCAGE** (Qiao et al., 2025) is a self-conformation-aware graph transformer that incorporates 3D geometric information and functional group tagging through multitask pretraining. We excluded the 3D atom-distances for fair comparison with our 2D graph setting.

We also evaluate against anti-oversmoothing methods to examine whether architectures designed to mitigate representation collapse can better handle activity cliff compounds. These methods include GINE + PairNorm (Zhao & Akoglu, 2019), GINE + NodeNorm (Zhou et al., 2021), GINE + Residual, GINE + DropEdge (Rong et al., 2019), APPNP (Gasteiger et al., 2018), PPNP (Gasteiger et al., 2018), GCNII (Chen et al., 2020b), and JK-Net (Xu et al., 2018b). Additionally, we test two Graph-Cliff variants. GraphCliff w/ JK-Net replaces gated fusion with JK aggregation, and GraphCliff w/ Residual substitutes gating with residual connections. These variants help isolate the contribution of our gating mechanism. We used two evaluation metrics: root mean squared error (RMSE) and RMSE$_{\text{cliff}}$. RMSE is computed over all molecules in the test set and measures the overall accuracy of the predicted $-\log_{10}(\text{Ki})$ values. In contrast, RMSE$_{\text{cliff}}$ is calculated specifically on compounds identified as activity cliffs, thereby quantifying the prediction error on these particularly challenging and structure-sensitive samples.

We present results for six datasets in Table 1, while the complete results across all 30 benchmark datasets are provided in Appendix Tables 5, 6, and 7 due to space limitations. Among pretrained models, ContextPred and MolCLR capture broad structural patterns but emphasize global similarity. They struggle with the localized perturbations characteristic of activity cliffs. Chemprop demonstrates stronger performance through bond directionality, enabling sharper discrimination of functional motifs. Anti-oversmoothing methods preserve feature diversity through uniform operations across nodes or layers, yet they lack selective modulation at structurally sensitive sites where activity cliffs occur. In contrast, GraphCliff's gating mechanism adapts propagation scale per node.

This enables strong local discrimination at cliff-defining substructures while capturing global SAR continuity. GraphCliff achieves the best average rank across all 30 datasets for both RMSE and $RMSE_{cliff}$, demonstrating that gated short- and long-range propagation is more effective than existing graph-based approaches and uniform anti-oversmoothing strategies for activity-cliff prediction. These findings are also consistent with the inherent expressiveness limitations of 1-dimensional Weisfeiler-Leman (1-WL) GNNs, which we further analyze in Section 8.6 through theoretical and empirical examples.

We also evaluated our approach on the nine datasets in LSSNS benchmark. As shown in Appendix Tables 8 and 9, GraphCliff does not uniformly dominate across all LSSNS protein targets. This outcome is expected, given that LSSNS was deliberately designed under low-data conditions. Each dataset is composed of only a few dozen to slightly over one hundred molecules and all built around narrow and highly conserved scaffolds. In such conditions, training a high-capacity graph neural network from scratch often leads to overfitting, unstable optimization, and limited generalization. To address this limitation, we investigated whether knowledge transfer from related, larger-scale datasets could provide more stable initialization. For each protein target in LSSNS, we identified a biologically similar target in the MoleculeACE. The results of this transfer-initialization strategy are reported in Tables 8 and 9. Across most protein targets, Transferred GraphCliff substantially reduced both RMSE and $RMSE_{cliff}$ compared to training from scratch. For instance, in the PKC$\iota$ and mGluR2 tasks, transferred models achieved notable gains in predictive accuracy. While performance was not uniformly improved for every target due to imperfect biological similarity, the overall trend clearly demonstrates that leveraging prior knowledge from related MoleculeACE datasets mitigates the difficulties of learning in data-scarce scenarios. These findings suggest that transfer learning is a practically useful strategy for extending the applicability to real-world settings, where data availability is often limited. The mapping from each LSSNS target to its MoleculeACE counterpart is provided in the Appendix Table 10.

## 5 ANALYSIS

### 5.1 ABLATION STUDIES

We conducted ablation studies to assess the individual contributions of the short-range filter, long-range filter, and gating mechanism in our architecture. As shown in Appendix Table 11, removing the short-range filter caused the most substantial performance drop, underscoring its critical role in the model. The short-range filter captures essential one-hop message-passing information and serves multiple functions: feeding into the long-range filter, providing input to the gating mechanism, and contributing to the final sum fusion. These pathways ensure that localized chemical information is effectively preserved and propagated throughout the network. Removing the long-range filter also degraded performance, though to a lesser extent. Because it captures broader structural context up to three hops, its absence restricts the model's ability to integrate global molecular features. The gating mechanism, while having the smallest standalone effect, still made a positive contribution through the adaptive combination of short- and long-range information. This mechanism proved more effective than naive feature summation and enhanced the model's ability to balance local and global information. We evaluated different graph pooling strategies and found that other methods (mean, sum, max) exacerbate over-smoothing by uniformly aggregating indistinguishable node embeddings. To overcome this, we adopted SAGPool, which adaptively selects informative nodes based on learned importance scores. Empirically, SAGPool outperformed basic pooling methods, resulting in lower RMSE and $RMSE_{cliff}$, confirming that adaptive node selection helps preserve both local and global information.

As shown in Appendix Table 12, we investigated the effect of using different GNN architectures in the short- and long-range filters. Our default configuration employs **GINE** for the short-range filter and **Chebyshev polynomials** for the long-range filter. We replace either component with GCN, GAT, or GIN. GINE, which incorporates edge features into the message-passing process, is particularly beneficial for molecular graphs, as edge attributes such as bond type, aromaticity, and stereochemistry encode important chemical information. In contrast, GCN and GIN do not explicitly utilize bond features, limiting their expressiveness. For long-range propagation, Chebyshev polynomials outperformed stacked GNNs such as GIN and GAT. This improvement is likely due to its use of spectral polynomials, which efficiently encode multi-hop neighborhood information within a

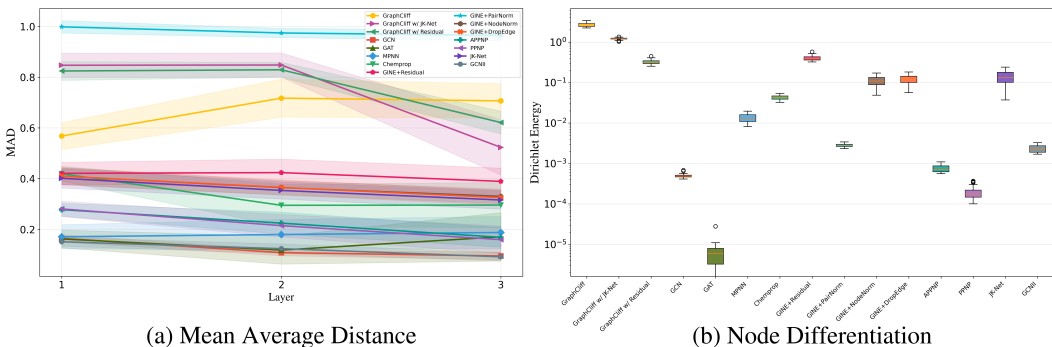

(a) Mean Average Distance                    (b) Node Differentiation

Figure 2: Layer-wise Mean Average Distance (MAD) and Dirichlet Energy across GNN architectures. (a) Layer-wise MAD quantifies the degree of node-level differentiation preserved at each propagation depth. Higher MAD indicates better preservation of distinct node representations. (b) Dirichlet Energy measures smoothness of node embeddings along edges. Lower energy indicates stronger smoothing.

single layer. These results underscore the importance of selecting GNNs that are structurally aligned with the type of information local or global being modeled. Moreover, they demonstrate that efficient compression of global context is particularly advantageous for representing complex molecular graphs.

## 5.2 ANALYSIS OF OVER-SMOOTHING MITIGATION IN GRAPHCLIFF

This section has been revised.

Over-smoothing is a well-known pathology in deep GNNs where repeated message passing causes node representations to become increasingly similar, losing their discriminative power. To quantify this phenomenon, we follow the methodology of Chen et al. (2020a) and Hariri et al. (2025a) and employ two metrics: Mean Average Distance (MAD) and Dirichlet Energy. For MAD, we quantify how much the node embeddings of a given graph diverge from one another at each layer. Specifically, we compute the cosine distance between every pair of node representations within a layer, ignore self-comparisons, and then average these distances for each node. The MAD for that layer is obtained by averaging these node-level distances. When MAD decreases as depth increases, it indicates that node representations are becoming more alike, signaling the onset of over-smoothing. We also evaluate the Dirichlet energy of the final-layer embeddings. This metric captures how much adjacent nodes differ in their representations. Low Dirichlet energy means that neighboring nodes have become nearly identical whereas higher values indicate that important local variations are still preserved.

Examining the MAD results, most baseline models show a clear decrease as layers deepen. This progressive loss of discriminative power confirms that repeated propagation causes node representations to collapse. Even models equipped with anti-smoothing mechanisms such as residual connections, DropEdge, and NodeNorm exhibit limited effectiveness in deeper layers. In contrast, GraphCliff displays an entirely different trend, its MAD either increases or remains consistently high across layers, indicating that it not only preserves but sometimes reinforces node-level distinctions during multi-hop propagation. GINE+PairNorm also maintains relatively high MAD, as PairNorm explicitly re-centers and re-scales embeddings at each layer, preventing representation contraction.

The Dirichlet energy results further strengthen these observations. Standard architectures like GAT exhibit extremely low values (as low as $10^{-5}$), indicating that neighboring node embeddings become nearly identical—strong evidence of severe smoothing. Anti-smoothing variants yield higher energy values but still within a limited range. By contrast, GraphCliff achieves the highest Dirichlet energy among all models, demonstrating that it maintains sharp differences across adjacent nodes throughout propagation. Overall, while most GNNs progressively flatten node-wise information and lose discriminability with depth, GraphCliff exhibits a striking contrast in both MAD and Dirichlet energy. Such behavior is particularly advantageous in activity-cliff scenarios, where subtle struc-

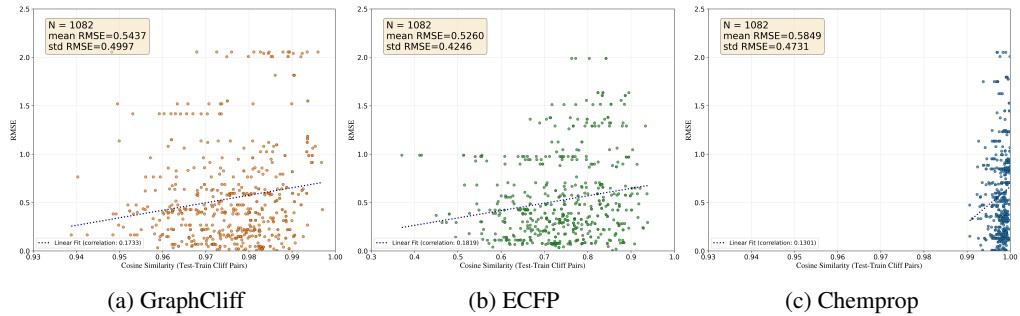

Figure 3: Correlation between representation similarity and predictive error for activity-cliff pairs on CHEMBL4792 (Ki).

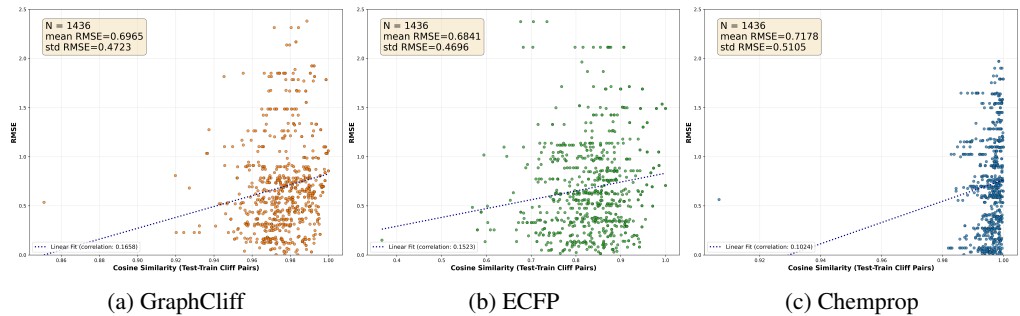

Figure 4: Correlation between representation similarity and predictive error for activity-cliff pairs on CHEMBL237 (Ki).

tural modifications must be reflected as distinct changes in molecular representations. The ability to preserve fine-grained local variations while capturing long-range dependencies makes GraphCliff especially well-suited for molecular property prediction.

### 5.3 ERROR ANALYSIS

==A new subsection has been added.==

In cases where GraphCliff underperformed, many failures coincided with SVM using an ECFP descriptor achieving lower error, suggesting that ECFP preserved certain local structural differences more effectively than GraphCliff's learned embedding. To probe this behavior further, we examined the cosine similarity of each cliff pair across three representations: ECFP, GraphClif, and ChemProp and related these values to their RMSE. While interpreting these similarities, it is important to note that the absolute scales differ across representations because ECFP is binary whereas ChemProp and GraphCliff produce continuous embeddings. Accordingly, ChemProp and GraphCliff similarities cluster near 0.9–1.0, whereas ECFP spans a wider range (approximately 0.3–1.0). This scale difference is a structural property of the representations and should be considered when comparing similarities across models.

ECFP provides strong local discrimination when the decisive modification falls within its hashing radius, enabling it to capture subtle structural changes sharply. GraphCliff, in contrast, combines short-range and long-range spectral propagation, which often produces high cosine similarity when molecules share broadly similar global structure. This can cause activity cliff pairs to appear relatively compressed in the embedding space. Even so, GraphCliff's short-range component distinguishes subtle differences more clearly than ChemProp. Consequently, when compared with both ECFP and ChemProp, GraphCliff does not always preserve the fine-grained structural distinctions that are functionally important in difficult cliff cases. These tendencies are also reflected in the mean RMSE values. In Figure 3, GraphCliff (0.5437) lies between ECFP (0.5260) and ChemProp (0.5849). A similar pattern appears in Figure 4, where GraphCliff (0.6965) again falls between

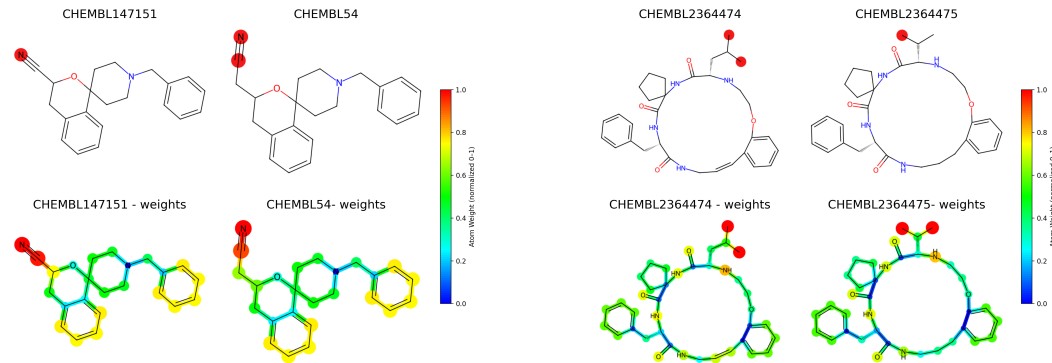

Figure 5: Visualization of a cliff pair with large functional divergence. Top: atoms responsible for the activity cliff (highlighted in red). Bottom: attention weights from the sigmoid gating vector $\sigma(x_1)$, with warmer colors indicating higher importance.

ECFP (0.6841) and ChemProp (0.7178). Overall, GraphCliff generally outperforms ChemProp, but its sensitivity to fine-grained perturbations still leaves room for improvement relative to ECFP.

## 5.4 QUALITATIVE ANALYSIS

To qualitatively assess whether our model identifies functionally relevant substructures, we visualized atom-level importance scores derived from the gating vector $\sigma(x_1) \in \mathbb{R}^{N \times d}$, where the sigmoid function assigns attention weights to each node. Specifically, we investigated whether atoms with high gating values align with those responsible for activity cliffs. Figure 5 shows two representative activity cliff pairs, where the top row highlights the difference atoms (shown in red) between the two compounds in each pair, and the bottom row visualizes atom importance scores obtained from the sigmoid-gated vector. The importance values are normalized between 0 and 1, and colored accordingly. We observe that atoms with the highest attention weights (indicated by warmer colors such as red and orange) are frequently aligned with the structural differences responsible for activity cliffs. This suggests that the gating mechanism successfully highlights substructures that are functionally discriminative, rather than relying solely on global molecular context. These results provide qualitative evidence that the gating path captures meaningful local information and contributes to the model's robustness in handling activity cliff compounds.

## 6 CONCLUSION

In this work, we introduced GraphCliff, a novel graph neural architecture designed to address two key limitations of existing GNNs: the loss of local sensitivity and the tendency toward over-smoothing. By explicitly integrating local structural details with global context through a gating mechanism over short- and long-range representations, GraphCliff provides a more balanced and chemically meaningful representation of molecules. Our extensive evaluation on the MoleculeACE benchmark demonstrated that GraphCliff consistently achieves improved performance across both non-cliff and activity cliff compounds, highlighting its robustness in challenging prediction settings. Furthermore, our in-depth analysis confirmed that GraphCliff effectively alleviates node over-smoothing, yielding more discriminative representations than conventional GNNs. Taken together, these findings suggest that explicitly combining local and global information is a promising direction for molecular gaph representation. Future research may build on this framework by further incorporating chemically informed descriptors, such as fingerprint-derived substructures, to bridge the gap between domain knowledge and learned graph representations.

## 7 LLM USAGE

Large language models (LLMs) were used in a limited assistive role during the preparation of this paper. LLMs were employed for grammar checking, rephrasing, and improving clarity of sentences. Some sentences were rephrased with the help of LLMs to improve readability, without altering the technical content. LLMs were occasionally used to identify relevant related work and papers, which were subsequently verified and selected by the authors.

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

# 8 APPENDIX

## 8.1 GRAPH EMBEDDING VS. ECFPs DISTANCE ANALYSIS

Figure 6 illustrates the relationship between ECFP fingerprint dissimilarities (x-axis) and graph embedding Euclidean distances (y-axis) across different GNN architectures. If the two measures were aligned, points would concentrate along the diagonal $y = x$, but conventional GNNs (GCN, GAT, and MPNN) generally underestimate distances, placing most points below the diagonal. GraphCliff achieves a distribution closer to the reference line, indicating that its embeddings more faithfully reflect structural differences captured by ECFPs. The regression slopes reported in Table 2 quantify this trend, confirming that GraphCliff narrows the gap between graph- and fingerprint-based representations.

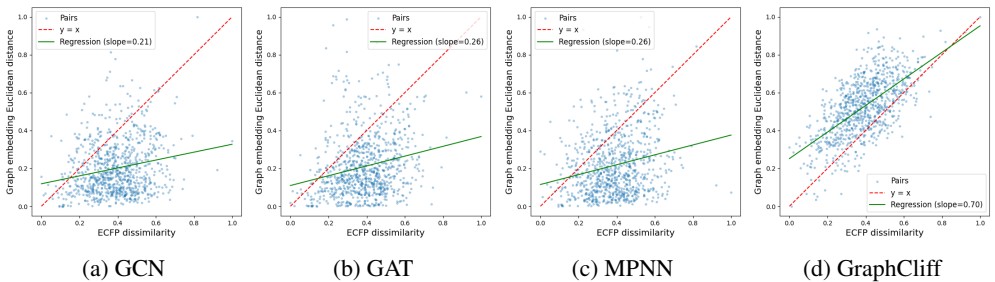

(a) GCN        (b) GAT        (c) MPNN        (d) GraphCliff

Figure 6: Comparison of graph embedding Euclidean distances with ECFP fingerprint dissimilarities across different models.

Table 2: Slopes of fitted regression lines comparing ECFP dissimilarities (x-axis) and graph embedding Euclidean distances (y-axis) for activity-cliff pairs across different GNN models.

| Dataset | GCN | GAT | MPNN | GraphCliff |
|---|---|---|---|---|
| CHEMBL1862_Ki | 0.121 | 0.508 | **0.718** | 0.564 |
| CHEMBL1871_Ki | -0.282 | 0.259 | 0.261 | **0.733** |
| CHEMBL2034_Ki | -0.055 | 0.388 | **0.533** | 0.433 |
| CHEMBL204_Ki | 0.2 | 0.188 | 0.138 | **0.684** |
| CHEMBL2047_EC50 | -0.047 | 0.401 | **0.543** | 0.541 |
| CHEMBL214_Ki | 0.158 | 0.095 | 0.328 | **0.686** |
| CHEMBL2147_Ki | 0.535 | 0.398 | 0.65 | **0.821** |
| CHEMBL218_EC50 | -0.028 | -0.095 | 0.429 | **0.572** |
| CHEMBL219_Ki | 0.051 | 0.23 | 0.318 | **0.742** |
| CHEMBL228_Ki | 0.234 | 0.156 | 0.489 | **0.635** |
| CHEMBL231_Ki | 0.207 | 0.431 | 0.389 | **0.672** |
| CHEMBL233_Ki | 0.144 | 0.44 | 0.366 | **0.628** |
| CHEMBL234_Ki | 0.034 | 0.001 | 0.138 | **0.75** |
| CHEMBL235_EC50 | 0.129 | 0.127 | 0.18 | **0.733** |
| CHEMBL236_Ki | 0.029 | -0.037 | 0.073 | **0.55** |
| CHEMBL237_EC50 | 0.222 | 0.144 | 0.226 | **0.67** |
| CHEMBL237_Ki | 0.153 | 0.305 | 0.383 | **0.577** |
| CHEMBL238_Ki | 0.02 | 0.056 | 0.302 | **0.556** |
| CHEMBL239_EC50 | -0.071 | -0.077 | -0.018 | **0.527** |
| CHEMBL244_Ki | 0.209 | 0.084 | 0.152 | **0.701** |
| CHEMBL262_Ki | **0.513** | -0.135 | -0.301 | 0.473 |
| CHEMBL264_Ki | 0.365 | 0.241 | 0.379 | **0.655** |
| CHEMBL2835_Ki | 0.557 | 0.172 | 0.398 | **0.924** |
| CHEMBL287_Ki | 0.255 | -0.01 | 0.02 | **0.643** |
| CHEMBL2971_Ki | **0.627** | -0.062 | 0.216 | 0.555 |
| CHEMBL3979_EC50 | 0.12 | **0.549** | **0.549** | 0.477 |
| CHEMBL4005_Ki | 0.237 | 0.134 | 0.307 | **0.549** |
| CHEMBL4203_Ki | 0.459 | 0.084 | 0.417 | **0.708** |
| CHEMBL4616_EC50 | 0.008 | -0.065 | -0.027 | **0.667** |
| CHEMBL4792_Ki | 0.047 | -0.073 | 0.123 | **0.573** |

## 8.2 DATASET METADATA

Tables 3 and 4 summarize the datasets used in our experiments. Table 3 provides general statistics of the MoleculeACE benchmark datasets, including the number of compounds, activity labels, and target proteins. Table 4 reports metadata of the LSSNS datasets, which consist of small sample sizes with narrow scaffold diversity.

Table 3: Statistics of MoleculeACE datasets corresponding to the ChEMBL targets used in this study.

| Dataset | ChEMBL ID | Type | Target name | Receptor Class | Train compounds (Train cliff) | Test compounds (Test cliff) | Total compounds (cliff) |
|---|---|---|---|---|---|---|---|
| CHEMBL1862_Ki | CHEMBL1862 | Ki | Tyrosine-protein kinase ABL1 | Kinase | 633 (202) | 161 (51) | 794 (253) |
| CHEMBL1871_Ki | CHEMBL1871 | Ki | Androgen Receptor | NR | 525 (126) | 134 (31) | 659 (157) |
| CHEMBL2034_Ki | CHEMBL2034 | Ki | Glucocorticoid receptor | NR | 598 (183) | 152 (47) | 750 (230) |
| CHEMBL2047_EC50 | CHEMBL2047 | EC50 | Farnesoid X receptor | NR | 503 (195) | 128 (50) | 631 (245) |
| CHEMBL204_Ki | CHEMBL204 | Ki | Thrombin | Protease | 2201 (790) | 553 (199) | 2754 (989) |
| CHEMBL2147_Ki | CHEMBL2147 | Ki | Serine/threonine-protein kinase PIM1 | Kinase | 1162 (387) | 294 (98) | 1456 (485) |
| CHEMBL214_Ki | CHEMBL214 | Ki | Serotonin 1a receptor | GPCR | 2651 (917) | 666 (230) | 3317 (1147) |
| CHEMBL218_EC50 | CHEMBL218 | EC50 | Cannabinoid receptor 1 | GPCR | 823 (292) | 208 (75) | 1031 (367) |
| CHEMBL219_Ki | CHEMBL219 | Ki | Dopamine D4 receptor | GPCR | 1485 (572) | 374 (143) | 1859 (715) |
| CHEMBL228_Ki | CHEMBL228 | Ki | Serotonin transporter | Other | 1362 (479) | 342 (120) | 1704 (599) |
| CHEMBL231_Ki | CHEMBL231 | Ki | Histamine H1 receptor | GPCR | 776 (178) | 197 (46) | 973 (224) |
| CHEMBL233_Ki | CHEMBL233 | Ki | u-opioid receptor | GPCR | 2512 (889) | 630 (222) | 3142 (1111) |
| CHEMBL234_Ki | CHEMBL234 | Ki | Dopamine D3 receptor | GPCR | 2923 (1150) | 734 (291) | 3657 (1441) |
| CHEMBL235_EC50 | CHEMBL235 | EC50 | PPAR gamma | NR | 1879 (703) | 470 (178) | 2349 (881) |
| CHEMBL236_Ki | CHEMBL236 | Ki | Delta opioid receptor | GPCR | 2077 (772) | 521 (193) | 2598 (965) |
| CHEMBL237_EC50 | CHEMBL237 | EC50 | Kappa opioid receptor | GPCR | 762 (319) | 193 (81) | 955 (400) |
| CHEMBL237_Ki | CHEMBL237 | Ki | Kappa opioid receptor | GPCR | 2081 (753) | 521 (188) | 2602 (941) |
| CHEMBL238_Ki | CHEMBL238 | Ki | Dopamine transporter | Other | 839 (209) | 213 (54) | 1052 (263) |
| CHEMBL239_EC50 | CHEMBL239 | EC50 | PPAR alpha | NR | 1377 (568) | 344 (141) | 1721 (709) |
| CHEMBL244_Ki | CHEMBL244 | Ki | Coagulation factor X | Protease | 2476 (1080) | 621 (270) | 3097 (1350) |
| CHEMBL262_Ki | CHEMBL262 | Ki | GSK-3 beta | Kinase | 683 (127) | 173 (31) | 856 (158) |
| CHEMBL264_Ki | CHEMBL264 | Ki | Histamine H3 receptor | GPCR | 2288 (865) | 574 (219) | 2862 (1084) |
| CHEMBL2835_Ki | CHEMBL2835 | Ki | Janus kinase 1 | Kinase | 489 (36) | 126 (10) | 615 (46) |
| CHEMBL287_Ki | CHEMBL287 | Ki | Sigma opioid receptor | Other | 1061 (371) | 267 (93) | 1328 (464) |
| CHEMBL2971_Ki | CHEMBL2971 | Ki | Janus kinase 2 | Kinase | 779 (95) | 197 (25) | 976 (120) |
| CHEMBL3979_EC50 | CHEMBL3979 | EC50 | PPAR delta | NR | 900 (373) | 225 (94) | 1125 (467) |
| CHEMBL4005_Ki | CHEMBL4005 | Ki | PI3K p110-alpha subunit | Transferase | 767 (281) | 193 (70) | 960 (351) |
| CHEMBL4203_Ki | CHEMBL4203 | Ki | CLK4 | Kinase | 582 (51) | 149 (13) | 731 (64) |
| CHEMBL4616_EC50 | CHEMBL4616 | EC50 | Ghrelin receptor | GPCR | 543 (262) | 139 (68) | 682 (330) |
| CHEMBL4792_Ki | CHEMBL4792 | Ki | Orexin receptor 2 | GPCR | 1174 (610) | 297 (153) | 1471 (763) |

Table 4: Statistics of LSSNS datasets corresponding to the protein targets used in this study.

| Dataset | ChEMBL ID | Type | Target name | Receptor Class | Train compounds (Train cliff) | Test compounds (Test cliff) | Total compounds (cliff) |
|---|---|---|---|---|---|---|---|
| USP7 | CHEMBL4251701 | – | Ubiquitin carboxyl-terminal hydrolase 7 | Protease | 36 (19) | 9 (5) | 45 (24) |
| RIP2 | CHEMBL4266012; CHEMBL4130524 | – | Serine/threonine-protein kinase RIPK2 | Kinase | 36 (16) | 10 (4) | 46 (20) |
| PKCι | CHEMBL4184321 | – | Protein kinase C iota | Kinase | 38 (12) | 10 (3) | 48 (15) |
| PHGDH | CHEMBL4373702 | – | D-3-phosphoglycerate dehydrogenase | Other Enzyme | 40 (13) | 11 (3) | 51 (16) |
| PLK1 | CHEMBL4406868; CHEMBL4138231 | – | Serine/threonine-protein kinase PLK1 | Kinase | 58 (22) | 15 (6) | 73 (28) |
| IDO1 | CHEMBL4364294 | – | Indoleamine 2,3-dioxygenase | Other Enzyme | 62 (34) | 16 (9) | 78 (43) |
| RXFP1 | CHEMBL3714716 | – | Relaxin receptor 1 | GPCR | 93 (52) | 24 (14) | 117 (66) |
| BRAF | CHEMBL3638563 | – | Serine/threonine-protein kinase B-raf | Kinase | 102 (27) | 26 (7) | 128 (34) |
| mGluR2 | CHEMBL3886984 | – | Metabotropic glutamate receptor 2 | GPCR | 195 (92) | 49 (23) | 244 (115) |

## 8.3 ALL RESULTS OF 30 DATASETS IN MOLECULEACE

Tables 5, 6, and 7 report the complete results across all 30 protein–ligand datasets included in MoleculeACE. Each table includes the performance of baseline machine learning models, graph-based neural networks, and our proposed GraphCliff. The results include both RMSE and $RMSE_{cliff}$, allowing a direct comparison of overall predictive accuracy and sensitivity to activity cliffs.

Among graph-based models (excluding anti-oversmoothing methods), **Chemprop** achieved the strongest performance. Its advantage can be attributed to its D-MPNN architecture, which incorporates bond directionality into message passing. This design enables the model to distinguish chemically distinct but structurally similar motifs, such as C=O versus O=C. Following closely, **SCAGE (w/o 3D)** also delivered competitive results. The presence of explicit functional group annotations guides the model to attend to chemically meaningful substructures. In contrast, **MolCLR** exhibited relatively weaker performance. While its contrastive learning objective promotes generalization by aligning embeddings of structurally similar molecules, it primarily captures global molecular similarity. This global bias may limit MolCLR's sensitivity to functionally important substructures, thereby contributing to its higher RMSE and $RMSE_{cliff}$ scores. **ContextPred**, based on a GIN (Xu et al., 2018a) backbone, showed limited performance because it failed to capture contextual substructure information without pretraining. Performance improved when context-based pretraining was applied followed by fine-tuning. However, the learned structural context alone was still insufficient to fully address the challenges posed by activity-cliff compounds. Finally, **KPGT**, a knowledge-guided pretraining method that incorporates domain-specific chemical information such as pharmacophore patterns and functional groups, showed modest performance compared to other models. This suggests that, despite being chemically informed, KPGT struggles to capture the subtle structure–activity discontinuities characteristic of activity cliff compounds. Overall, while several baselines demonstrated competitive performance, our results indicate that GraphCliff achieves consistently strong performance relative to prior approaches. This underscores the effectiveness of explicitly balancing local substructural sensitivity with global molecular context in addressing both general prediction tasks and activity cliff scenarios.

In addition to pretrained and supervised graph-based models, we also benchmark several anti-oversmoothing methods that have been proposed to mitigate representation collapse in deep GNNs. Specifically, we evaluate: GINE + PairNorm, GINE + NodeNorm, GINE + Residual, GINE + DropEdg, APPNP, PPNP, GCNII, JK-Net, as well as two GraphCliff variants, GraphCliff w/ JK-Net (replacing gated fusion with JK aggregation) and GraphCliff w/ Residual (substituting gating with residual connection). Although these methods are effective at counteracting oversmoothing, their operations act uniformly across all nodes or layers. Techniques such as PairNorm and NodeNorm normalize embeddings globally, residual-based architectures inject uniform skip signals, and propagation-based models like APPNP/PPNP blend predictions with fixed teleportation probabilities. These mechanisms preserve global distinguishability but lack the ability to modulate propagation selectively at the level of individual atoms, particularly at structurally sensitive sites where activity cliffs occur. Consequently, while many of these anti-oversmoothing baselines yield improved RMSE compared to standard GNNs, they often fail to adequately preserve the fine-grained local distinctions required for cliff-sensitive prediction tasks. By contrast, GraphCliff's gating mechanism adapts the scale of propagation per node, enabling strong local discrimination at cliff-defining substructures while still capturing global SAR continuity. As shown in Tables 5–7, this leads to consistently competitive or superior performance across both RMSE and $RMSE_{cliff}$, highlighting the importance of context-dependent message modulation rather than uniform anti-oversmoothing strategies.

Table 5: RMSE / RMSE$_{cliff}$ (Part 1/3) with best (**bold**) and second-best (underlined) highlighted.

| Algorithm | Descriptor | CHEMBL1862 (Ki) | CHEMBL1871 (Ki) | CHEMBL2034 (Ki) | CHEMBL2047 (EC50) | CHEMBL204 (Ki) | CHEMBL2147 (Ki) | CHEMBL214 (Ki) | CHEMBL218 (EC50) | CHEMBL219 (Ki) | CHEMBL228 (Ki) |
|---|---|---|---|---|---|---|---|---|---|---|---|
| GraphCliff | GRAPH | 0.7810 / 0.6740 | 0.6279 / 0.7975 | 0.7473 / 0.8580 | 0.5924 / 0.6549 | **0.6914** / **0.8212** | 0.5600 / **0.5790** | **0.6210** / 0.7260 | 0.6974 / 0.7811 | 0.6636 / 0.7435 | **0.6513** / **0.6739** |
| GraphCliff w/ JK-Net | GRAPH | 0.8194 / 0.7496 | **0.6042** / **0.7342** | 0.7718 / 0.9196 | 0.6289 / 0.7330 | 0.6947 / 0.8226 | **0.5581** / 0.6186 | 0.6302 / 0.7485 | 0.7252 / 0.7997 | **0.6634** / **0.7328** | 0.6672 / 0.6902 |
| SVM | ECFP | 0.7736 / **0.6739** | 0.6649 / 0.8734 | **0.6737** / **0.8129** | 0.6136 / 0.6874 | 0.7231 / 0.8588 | 0.5762 / 0.5796 | 0.6345 / **0.7238** | 0.7190 / 0.7614 | 0.7096 / 0.7883 | 0.6621 / 0.6759 |
| GraphCLiff w/ Residual | GRAPH | 0.8148 / 0.7696 | 0.6615 / 0.7796 | 0.7369 / 0.8556 | 0.6177 / 0.6779 | 0.7126 / 0.8479 | 0.5815 / 0.6362 | 0.6415 / 0.7612 | 0.7669 / 0.8055 | 0.7052 / 0.7818 | 0.7010 / 0.6994 |
| GBM | ECFP | 0.7977 / 0.7465 | 0.6775 / 0.9221 | 0.7674 / 0.8637 | 0.6020 / 0.6396 | 0.7533 / 0.9320 | 0.5811 / 0.6161 | 0.6775 / 0.7611 | 0.7116 / 0.7464 | 0.7117 / 0.7646 | 0.6862 / 0.7234 |
| RF | ECFP | 0.8051 / 0.6859 | 0.6600 / 0.9064 | 0.7269 / 0.8518 | 0.6275 / 0.6646 | 0.7628 / 0.8988 | 0.6616 / 0.6763 | 0.7003 / 0.8010 | 0.7055 / 0.7596 | 0.7227 / 0.7650 | 0.7093 / 0.7733 |
| GINE + PairNorm | GRAPH | 0.8408 / 0.7460 | 0.6301 / 0.8670 | 0.7245 / 0.8453 | 0.6125 / 0.5953 | 0.7451 / 0.8983 | 0.6933 / 0.6793 | 0.7222 / 0.8089 | 0.6903 / 0.7711 | 0.7198 / 0.7691 | 0.6931 / 0.7139 |
| JK-Net | GRAPH | 0.9715 / 0.9612 | 0.6400 / 0.8227 | 0.7143 / 0.8245 | **0.5921** / **0.5779** | 0.7783 / 0.8997 | 0.6663 / 0.6807 | 0.6665 / 0.7683 | 0.7348 / 0.7825 | 0.7400 / 0.8127 | 0.7129 / 0.7710 |
| GINE + NodeNorm | GRAPH | 0.9213 / 0.8562 | 0.6720 / 0.8341 | 0.7187 / 0.8524 | 0.6542 / 0.7203 | 0.7376 / 0.8635 | 0.7788 / 0.6964 | 0.6806 / 0.7934 | 0.7653 / 0.7763 | 0.7021 / 0.7969 | 0.7017 / 0.7565 |
| GINE + Residual | GRAPH | 0.8660 / 0.7848 | 0.6778 / 0.7810 | 0.7300 / 0.8964 | 0.6696 / 0.6731 | 0.8000 / 0.9378 | 0.7111 / 0.7330 | 0.7098 / 0.8114 | 0.7265 / 0.7927 | 0.7472 / 0.8188 | 0.7080 / 0.7927 |
| KNN | ECFP | 0.8983 / 0.8218 | 0.6498 / 0.8170 | 0.6965 / 0.9133 | 0.6418 / 0.7353 | 0.8207 / 0.9954 | 0.6618 / 0.6821 | 0.7335 / 0.8742 | 0.7355 / 0.7848 | 0.7754 / 0.8159 | 0.7173 / 0.8112 |
| Chemprop | GRAPH | 0.8149 / 0.6928 | 0.7043 / 0.9193 | 0.7748 / 0.8857 | 0.6933 / 0.7203 | 0.8109 / 0.8506 | 0.6491 / 0.6392 | 0.6600 / 0.8247 | 0.7580 / 0.7938 | 0.6916 / 0.7740 | 0.6705 / 0.6951 |
| LSTM | SMILES | **0.7614** / 0.7927 | 0.6621 / 0.8502 | 0.7553 / 0.9441 | 0.6955 / 0.7904 | 0.8224 / 0.9302 | 0.6465 / 0.7246 | 0.7232 / 0.8506 | 0.7479 / 0.8178 | 0.7797 / 0.8549 | 0.7788 / 0.8836 |
| GBM | MACCS | 0.8602 / 0.7781 | 0.6859 / 0.9039 | 0.7234 / 0.8565 | 0.6703 / 0.7019 | 0.7998 / 0.9579 | 0.7915 / 0.7775 | 0.7295 / 0.8565 | 0.7152 / 0.7655 | 0.8159 / 0.8619 | 0.7701 / 0.8074 |
| SVM | MACCS | 0.8905 / 0.8592 | 0.6702 / 0.8905 | 0.6947 / 0.7363 | 0.6808 / 0.8602 | 0.8059 / 0.9832 | 0.7937 / 0.8061 | 0.7463 / 0.8888 | 0.7030 / 0.7289 | 0.8204 / 0.8839 | 0.7450 / 0.7924 |
| RF | MACCS | 0.8739 / 0.8439 | 0.7025 / 0.9044 | 0.7002 / 0.8283 | 0.6765 / 0.7071 | 0.8149 / 0.9706 | 0.8060 / 0.7925 | 0.7467 / 0.8872 | **0.6688** / **0.7124** | 0.8070 / 0.8375 | 0.7690 / 0.8097 |
| MLP | ECFP | 0.8776 / 0.7811 | 0.7367 / 0.9579 | 0.7423 / 0.8488 | 0.6766 / 0.7283 | 0.8154 / 0.9622 | 0.7232 / 0.7035 | 0.6932 / 0.7803 | 0.7851 / 0.8059 | 0.7557 / 0.8319 | 0.7550 / 0.7574 |
| GINE + DropEdge | GRAPH | 1.0060 / 0.9994 | 0.7704 / 0.9536 | 0.7485 / 0.9009 | 0.7415 / 0.7301 | 0.8658 / 0.9784 | 0.7846 / 0.8020 | 0.7644 / 0.8741 | 0.8521 / 0.8753 | 0.8325 / 0.8942 | 0.8104 / 0.8577 |
| SCAGE | GRAPH | 0.8753 / 0.6768 | 0.7582 / 0.8230 | 0.7446 / 0.8250 | 0.7137 / 0.6054 | 0.8135 / 0.8884 | 0.9104 / 0.8584 | 0.7793 / 0.8336 | 0.7379 / 0.7833 | 0.8765 / 0.8630 | 0.7713 / 0.7865 |
| GCNII | GRAPH | 0.9601 / 0.9193 | 0.7403 / 0.9383 | 0.7593 / 0.8566 | 0.7248 / 0.7001 | 0.8258 / 0.9342 | 1.0474 / 0.9713 | 0.7543 / 0.8342 | 0.7576 / 0.8331 | 0.8156 / 0.8606 | 0.7838 / 0.8280 |
| KNN | MACCS | 1.0494 / 0.8676 | 0.7051 / 0.9412 | 0.7682 / 0.9102 | 0.7057 / 0.6895 | 0.8910 / 1.0433 | 0.9438 / 0.8668 | 0.7988 / 0.9017 | 0.7071 / 0.7560 | 0.8726 / 0.8878 | 0.7785 / 0.8362 |
| APPNP | GRAPH | 1.0084 / 0.9476 | 0.7624 / 0.9685 | 0.7650 / 0.8998 | 0.7595 / 0.7709 | 0.8719 / 0.9734 | 1.1075 / 1.0046 | 0.8000 / 0.8887 | 0.8527 / 0.8863 | 0.8865 / 0.9236 | 0.8442 / 0.8787 |
| RF | PHYSCHEM | 0.9085 / 0.9395 | 0.7530 / 1.0577 | 0.7054 / 0.8348 | 0.8137 / 0.9004 | 1.1046 / 1.1609 | 0.9720 / 0.9162 | 0.9090 / 0.9892 | 0.8228 / 0.7997 | 0.9297 / 0.9700 | 0.9214 / 0.9166 |
| GBM | PHYSCHEM | 0.9427 / 0.9033 | 0.7244 / 1.0485 | 0.7292 / 0.8670 | 0.8130 / 0.9096 | 1.1445 / 1.2108 | 0.9486 / 0.8935 | 0.8949 / 0.9881 | 0.8286 / 0.8262 | 0.9496 / 0.9798 | 0.9072 / 0.9228 |
| Transformer | TOKENS | 0.9606 / 0.9637 | 0.8085 / 1.0729 | 0.8036 / 0.9380 | 0.7667 / 0.7253 | 1.0979 / 1.2547 | 0.9026 / 0.8930 | 0.8617 / 0.9523 | 0.8692 / 0.8951 | 0.8781 / 0.9398 | 0.9076 / 0.9787 |
| GCN | GRAPH | 0.9416 / 0.9420 | 0.7689 / 1.0090 | 0.8100 / 0.9282 | 0.7973 / 0.7807 | 1.0563 / 1.2010 | 0.8400 / 0.8251 | 1.0068 / 1.0836 | 0.9276 / 0.9518 | 1.0263 / 1.0553 | 0.9583 / 1.0005 |
| KNN | PHYSCHEM | 1.0008 / 0.9602 | 0.7969 / 1.0635 | 0.7609 / 0.9014 | 0.7655 / 0.8450 | 1.1148 / 1.1689 | 1.0530 / 0.9226 | 0.9710 / 1.0487 | 0.8828 / 0.8324 | 0.9701 / 0.9853 | 0.9893 / 0.9343 |
| CNN | SMILES | 1.0489 / 0.8998 | 0.8100 / 1.0410 | 0.7978 / 0.9335 | 0.7761 / 0.7740 | 1.1314 / 1.2338 | 0.9246 / 0.9338 | 0.9310 / 1.0071 | 0.9584 / 0.9763 | 0.9768 / 0.9726 | 0.9652 / 0.9440 |
| SVM | PHYSCHEM | 0.9663 / 0.9201 | 0.8026 / 1.0663 | 0.7632 / 0.9102 | 0.7146 / 0.6672 | 1.2052 / 1.3129 | 0.9655 / 0.9065 | 0.9567 / 0.9649 | 1.0259 / 1.0046 | 0.9602 / 1.0090 | 0.9297 / 0.8905 |
| GAT | GRAPH | 0.9869 / 1.0007 | 0.7984 / 1.0424 | 0.8091 / 0.9415 | 0.8403 / 0.7896 | 1.1377 / 1.2815 | 0.9661 / 0.9167 | 1.0510 / 1.1372 | 0.9569 / 0.9835 | 0.9790 / 0.9824 | 1.0265 / 1.0278 |
| RF | WHIM | 0.8645 / 0.8864 | 0.8817 / 1.1043 | 0.7819 / 0.8769 | 0.8331 / 0.7674 | 1.2578 / 1.3597 | 1.0416 / 0.9957 | 1.0260 / 1.0795 | 0.8840 / 0.8941 | 0.9895 / 1.0222 | 1.0496 / 1.0408 |
| GBM | WHIM | 0.8732 / 0.8501 | 0.9027 / 1.1041 | 0.7699 / 0.8551 | 0.8528 / 0.7324 | 1.2664 / 1.3814 | 1.0032 / 0.9607 | 1.0479 / 1.0890 | 0.8906 / 0.9158 | 1.0327 / 1.0755 | 1.0791 / 1.0201 |
| SVM | WHIM | 1.0280 / 0.9655 | 0.8571 / 1.0881 | 0.8420 / 0.9637 | 0.8511 / 0.8248 | 1.3113 / 1.4359 | 1.1029 / 1.0025 | 1.0600 / 1.0669 | 0.8907 / 0.9255 | 1.0407 / 1.0740 | 1.0948 / 1.0355 |
| KNN | WHIM | 0.9865 / 0.8491 | 0.8810 / 1.0691 | 0.8047 / 0.9293 | 0.8407 / 0.8327 | 1.3652 / 1.4822 | 1.1407 / 1.0130 | 1.0693 / 1.1120 | 0.9208 / 0.9195 | 1.0611 / 1.0863 | 1.0756 / 1.0168 |
| MPNN | GRAPH | 0.9477 / 0.8884 | 1.0583 / 1.1539 | 0.9048 / 0.9268 | 1.0302 / 0.9509 | 1.4584 / 1.5806 | 1.0245 / 0.9340 | 1.1833 / 1.2433 | 1.0531 / 1.0623 | 0.9030 / 0.9193 | 0.9995 / 1.0145 |
| AFP | GRAPH | 1.3474 / 1.1578 | 1.1428 / 1.2745 | 0.9309 / 0.9486 | 0.9703 / 0.9023 | 1.5527 / 1.7434 | 1.9063 / 1.3681 | 1.0832 / 1.1343 | 1.0459 / 1.0402 | 0.9523 / 0.9656 | 1.1923 / 1.1603 |
| MolCLR$_{gcn}$ | GRAPH | 0.9896 / 1.1098 | 0.9482 / 0.8629 | 1.3034 / 1.3151 | 0.7810 / 0.7568 | 1.5917 / 1.6165 | 1.5509 / 1.5455 | 1.0088 / 1.0386 | 1.0999 / 1.1056 | 0.9866 / 0.9428 | 1.3398 / 1.3170 |
| MolCLR$_{gcn}^{pretrained}$ | GRAPH | 1.0680 / 1.1939 | 0.9248 / 0.8320 | 1.2919 / 1.3054 | 0.7672 / 0.7390 | 1.5568 / 1.5791 | 1.9046 / 2.0633 | 1.0503 / 1.0759 | 1.0720 / 1.1182 | 0.9838 / 0.9396 | 1.3396 / 1.3288 |
| PPNP | GRAPH | 1.4631 / 1.4367 | 1.1400 / 1.3332 | 1.0041 / 0.9308 | 1.2564 / 0.9284 | 1.5242 / 1.7086 | 1.9766 / 1.8805 | 1.1338 / 1.1517 | 1.0523 / 0.9986 | 1.0711 / 1.0656 | 1.1956 / 1.1603 |
| MolCLR$_{gin}^{pretrained}$ | GRAPH | 1.0512 / 1.1760 | 1.0775 / 1.0196 | 1.5104 / 1.5500 | 0.7840 / 0.7643 | 1.6887 / 1.7093 | 1.2265 / 1.0147 | 1.1438 / 1.1834 | 1.0999 / 1.1056 | 1.0222 / 0.9912 | 1.4587 / 1.3643 |
| MolCLR$_{gin}$ | GRAPH | 1.0512 / 1.1760 | 1.0775 / 1.0196 | 1.4538 / 1.4921 | 0.7840 / 0.7643 | 1.6887 / 1.7093 | 1.2265 / 1.0147 | 1.1438 / 1.1834 | 1.0697 / 1.0587 | 1.0222 / 0.9912 | 1.4587 / 1.3643 |
| Contextpred$^{pretrained}$ | GRAPH | 1.3513 / 1.8131 | 1.7136 / 1.7663 | 1.4155 / 1.4123 | 1.6540 / 2.0069 | 1.9574 / 2.2021 | 1.2663 / 1.8716 | 1.5010 / 1.5931 | 1.7257 / 1.9468 | 1.6610 / 1.6946 | 1.4937 / 1.6573 |
| Contextpred | GRAPH | 1.3711 / 1.8173 | 1.6469 / 1.6866 | 1.4749 / 1.4535 | 1.7401 / 2.0606 | 2.0118 / 2.2692 | 1.2950 / 1.8566 | 1.6802 / 1.7679 | 1.5827 / 1.7777 | 1.6858 / 1.7417 | 1.6626 / 1.8030 |
| KPGT$^{pretrained}$ | GRAPH | 1.6267 / 1.3956 | 1.9951 / 1.8846 | 1.4158 / 1.5493 | 2.8480 / 2.7568 | 2.2878 / 2.1817 | 1.6150 / 1.1296 | 1.7811 / 1.7774 | 2.3303 / 2.3055 | 1.9625 / 1.9738 | 1.8300 / 1.8475 |
| KPGT | GRAPH | 1.6682 / 1.4652 | 1.9764 / 1.8220 | 1.4203 / 1.5942 | 2.8369 / 2.6945 | 2.3020 / 2.2098 | 1.6755 / 1.2009 | 1.8017 / 1.7803 | 2.1985 / 2.0988 | 2.1214 / 2.0995 | 1.8556 / 1.8465 |

Table 6: RMSE / RMSE$_{cliff}$ (Part 2/3) with best (**bold**) and second-best (underlined) highlighted.

| Algorithm | Descriptor | CHEMBL231 (Ki) | CHEMBL233 (Ki) | CHEMBL234 (Ki) | CHEMBL235 (EC50) | CHEMBL236 (Ki) | CHEMBL237 (EC50) | CHEMBL237 (Ki) | CHEMBL238 (Ki) | CHEMBL239 (EC50) | CHEMBL244 (Ki) |
|---|---|---|---|---|---|---|---|---|---|---|---|
| GraphCliff | GRAPH | **0.7080** / 0.8660 | 0.7860 / 0.8803 | 0.6175 / 0.6316 | 0.6504 / **0.7544** | 0.6965 / 0.7850 | **0.7075** / **0.7668** | 0.7046 / 0.7833 | **0.5966** / 0.7293 | **0.6701** / **0.7919** | **0.6683** / 0.7516 |
| GraphCliff w/ JK-Net | GRAPH | 0.7292 / 0.8745 | **0.7642** / 0.8663 | **0.5991** / **0.5979** | 0.6551 / 0.7642 | **0.6873** / 0.7958 | 0.7294 / 0.8026 | 0.6985 / 0.7567 | 0.6102 / 0.7264 | 0.6728 / 0.8019 | 0.6830 / 0.7683 |
| SVM | ECFP | 0.7503 / 0.9322 | 0.7742 / 0.8591 | 0.6217 / 0.6318 | 0.6397 / 0.7743 | 0.6981 / 0.7978 | 0.7200 / 0.7816 | 0.6771 / 0.7350 | 0.6101 / 0.6808 | 0.6776 / 0.8191 | 0.7152 / 0.7966 |
| GraphCLiff w/ Residual | GRAPH | 0.7155 / 0.8663 | 0.7929 / 0.8847 | 0.6331 / 0.6409 | 0.6513 / 0.7766 | 0.7023 / **0.7592** | 0.7776 / 0.8797 | 0.7174 / 0.7876 | 0.6006 / 0.6958 | 0.7263 / 0.8391 | 0.6826 / **0.7485** |
| GBM | ECFP | 0.7737 / 0.9548 | 0.8017 / 0.8857 | 0.6293 / 0.6486 | 0.6631 / 0.8063 | 0.6962 / 0.7935 | 0.8026 / 0.8803 | 0.7127 / 0.7887 | 0.6180 / **0.6377** | 0.6825 / 0.8205 | 0.7361 / 0.8208 |
| RF | ECFP | 0.8215 / 0.9074 | 0.8007 / 0.8796 | 0.6600 / 0.6829 | **0.6385** / 0.7644 | 0.7108 / 0.7921 | 0.7620 / 0.7927 | 0.7288 / 0.7995 | 0.6247 / 0.6562 | 0.6890 / 0.8248 | 0.7407 / 0.8231 |
| GINE + PairNorm | GRAPH | 0.7292 / 0.7855 | 0.7959 / 0.8899 | 0.6819 / 0.7028 | 0.6473 / 0.7946 | 0.7575 / 0.8165 | 0.7571 / 0.8735 | 0.7384 / 0.8103 | 0.6578 / 0.7359 | 0.7331 / 0.8636 | 0.7861 / 0.8591 |
| JK-Net | GRAPH | 0.7756 / 0.8577 | 0.7766 / 0.8589 | 0.6553 / 0.6557 | 0.6716 / 0.7849 | 0.7412 / 0.8329 | 0.7637 / 0.8654 | 0.7474 / 0.8238 | 0.6437 / 0.7385 | 0.7858 / 0.9106 | 0.7580 / 0.8343 |
| GINE + NodeNorm | GRAPH | 0.7566 / 0.8443 | 0.7912 / 0.8769 | 0.6717 / 0.6545 | 0.6563 / 0.7793 | 0.7300 / 0.8169 | 0.7381 / 0.8174 | 0.7279 / 0.7953 | 0.6663 / 0.7469 | 0.6928 / 0.8229 | 0.7576 / 0.8361 |
| GINE + Residual | GRAPH | 0.7953 / 0.8026 | 0.7937 / 0.8812 | 0.6940 / 0.6995 | 0.6882 / 0.8191 | 0.7357 / 0.7980 | 0.7709 / 0.8544 | 0.7528 / 0.8101 | 0.7026 / 0.7650 | 0.7095 / 0.8393 | 0.7655 / 0.8153 |
| KNN | ECFP | 0.7763 / 1.0202 | 0.8160 / 0.9123 | 0.6752 / 0.6989 | 0.6881 / 0.7946 | 0.7458 / 0.8650 | 0.8100 / 0.8758 | 0.7362 / 0.8507 | 0.6466 / 0.7346 | 0.7136 / 0.8646 | 0.7674 / 0.8735 |
| Chemprop | GRAPH | 0.7596 / 0.8169 | 0.7990 / 0.8438 | 0.6868 / 0.6532 | 0.7081 / 0.7995 | 0.7992 / 0.8827 | 0.7674 / 0.8394 | 0.7426 / 0.8003 | 0.6730 / 0.7361 | 0.8194 / 0.8271 | 0.7256 / 0.7972 |
| LSTM | SMILES | 0.8086 / 1.0703 | 0.8501 / 0.9418 | 0.7381 / 0.7970 | 0.7270 / 0.8474 | 0.8118 / 0.9051 | 0.7832 / 0.9028 | 0.7735 / 0.8618 | 0.6536 / 0.7925 | 0.7648 / 0.9051 | 0.8003 / 0.9128 |
| GBM | MACCS | 0.7543 / 0.7590 | 0.8463 / 0.9134 | 0.7189 / 0.7241 | 0.7131 / 0.8604 | 0.8392 / 0.9308 | 0.8312 / 0.9041 | 0.7994 / 0.8857 | 0.6783 / 0.6999 | 0.7207 / 0.8409 | 0.7989 / 0.8789 |
| SVM | MACCS | 0.7829 / 0.8372 | 0.8685 / 0.9534 | 0.7392 / 0.7293 | 0.6961 / 0.8382 | 0.8498 / 0.9377 | 0.8316 / 0.8853 | 0.7791 / 0.8726 | 0.6687 / 0.6821 | 0.7178 / 0.8533 | 0.8183 / 0.8697 |
| RF | MACCS | 0.7386 / 0.8280 | 0.8284 / 0.9060 | 0.7443 / 0.7498 | 0.7052 / 0.8232 | 0.8281 / 0.9565 | 0.8488 / 0.9113 | 0.8140 / 0.9274 | 0.7024 / 0.7082 | 0.7346 / 0.8681 | 0.8458 / 0.9174 |
| MLP | ECFP | 1.3335 / 1.2722 | 0.8452 / 0.9157 | 0.6693 / 0.6759 | 0.7180 / 0.8178 | 0.7329 / 0.8098 | 0.9018 / 0.9504 | 0.7216 / 0.7653 | 0.6839 / 0.7320 | 0.7559 / 0.9010 | 0.7955 / 0.8497 |
| GINE + DropEdge | GRAPH | 0.7923 / 0.8716 | 0.8611 / 0.9405 | 0.7518 / 0.7418 | 0.7117 / 0.8541 | 0.8138 / 0.9103 | 0.8950 / 0.9487 | 0.7821 / 0.8438 | 0.7444 / 0.8308 | 0.7924 / 0.9296 | 0.8467 / 0.9104 |
| SCAGE | GRAPH | 0.9145 / 0.7844 | 0.8244 / **0.8004** | 0.7993 / 0.8114 | 0.6941 / 0.7875 | 0.7987 / 0.9285 | 0.9673 / 0.9601 | 0.7623 / 0.8222 | 0.7261 / 0.6813 | 0.7959 / 0.8506 | 0.8923 / 0.9723 |
| GCNII | GRAPH | 0.8574 / 0.7778 | 0.8372 / 0.9122 | 0.7294 / 0.7103 | 0.6928 / 0.8318 | 0.7863 / 0.8299 | 1.0908 / 1.0403 | 0.7692 / 0.8311 | 0.7274 / 0.7441 | 0.7561 / 0.8674 | 0.8574 / 0.9347 |
| KNN | MACCS | 0.8366 / 0.9586 | 0.8523 / 0.9106 | 0.7817 / 0.7580 | 0.7503 / 0.8682 | 0.8963 / 1.0465 | 0.9619 / 0.9503 | 0.8249 / 0.9426 | 0.7077 / 0.7353 | 0.7910 / 0.8951 | 0.9016 / 0.9473 |
| APPNP | GRAPH | 0.8659 / 0.8776 | 0.9045 / 0.9578 | 0.7938 / 0.7788 | 0.7608 / 0.9047 | 0.8622 / 0.9159 | 0.9286 / 0.9318 | 0.8286 / 0.8655 | 0.7846 / 0.8673 | 0.8233 / 0.9480 | 0.9900 / 0.9899 |
| RF | PHYSCHEM | 0.9078 / **0.7321** | 1.0193 / 0.9963 | 0.8866 / 0.8586 | 0.7949 / 0.8915 | 0.9893 / 0.9983 | 0.9660 / 0.8935 | 0.9612 / 0.9795 | 0.9034 / 0.8508 | 0.8839 / 1.0317 | 1.1161 / 1.1190 |
| GBM | PHYSCHEM | 0.9531 / 0.8270 | 1.0502 / 1.0373 | 0.8832 / 0.8464 | 0.8210 / 0.9114 | 1.0141 / 1.0165 | 0.9992 / 0.9688 | 0.9609 / 0.9688 | 0.9018 / 0.8363 | 0.8745 / 1.0227 | 1.1047 / 1.1251 |
| Transformer | TOKENS | 0.9641 / 1.0284 | 1.0720 / 1.1179 | 0.8626 / 0.8483 | 0.8015 / 0.9136 | 1.0236 / 1.1023 | 1.1260 / 1.1842 | 0.9961 / 1.0624 | 0.8802 / 0.8515 | 0.9095 / 1.0319 | 1.0781 / 1.0707 |
| GCN | GRAPH | 0.8783 / 0.7969 | 1.0561 / 1.1056 | 0.9344 / 0.9186 | 0.9008 / 1.0389 | 0.9425 / 1.0003 | 1.1316 / 1.0944 | 1.1119 / 1.1521 | 0.9370 / 0.9249 | 0.9055 / 1.0237 | 1.0745 / 1.0602 |
| KNN | PHYSCHEM | 1.0454 / 0.8996 | 1.0372 / 1.0296 | 0.9109 / 0.8977 | 0.8736 / 0.9376 | 1.0324 / 1.0278 | 0.9983 / 0.9304 | 0.9686 / 0.9554 | 0.9394 / 0.8525 | 0.8738 / 0.9912 | 1.1384 / 1.1425 |
| CNN | SMILES | 1.0080 / 1.0442 | 1.0730 / 1.0796 | 0.8979 / 0.8811 | 0.8925 / 0.9623 | 1.0179 / 1.0669 | 1.0608 / 1.0216 | 1.0401 / 1.0403 | 0.9175 / 0.8975 | 0.9102 / 0.9857 | 1.0945 / 1.0706 |
| SVM | PHYSCHEM | 0.9936 / 0.8922 | 1.1595 / 1.1296 | 0.9436 / 0.8892 | 0.9126 / 1.0058 | 1.1155 / 1.2229 | 1.0847 / 1.0709 | 1.0199 / 1.0438 | 1.0011 / 0.9546 | 0.9641 / 1.0548 | 1.1602 / 1.1158 |
| GAT | GRAPH | 0.9909 / 0.9696 | 1.0659 / 1.0986 | 0.9499 / 0.9119 | 0.8692 / 1.0117 | 1.0023 / 1.0999 | 1.1029 / 1.0637 | 1.0850 / 1.0978 | 0.9276 / 0.9458 | 0.9016 / 1.0123 | 1.0881 / 1.1173 |
| RF | WHIM | 0.9533 / 0.9391 | 1.1316 / 1.1480 | 0.9691 / 0.9023 | 1.0037 / 1.1015 | 1.1178 / 1.1631 | 1.3018 / 1.3140 | 1.1205 / 1.1400 | 0.9935 / 0.9634 | 0.9979 / 1.0783 | 1.2490 / 1.2071 |
| GBM | WHIM | 0.9591 / 0.9619 | 1.1469 / 1.1585 | 0.9991 / 0.9123 | 1.0001 / 1.1008 | 1.1319 / 1.1788 | 1.3569 / 1.3497 | 1.1247 / 1.1285 | 0.9907 / 0.9495 | 1.0475 / 1.1333 | 1.2866 / 1.2301 |
| SVM | WHIM | 0.9558 / 0.9001 | 1.1497 / 1.1909 | 0.9871 / 0.9221 | 0.9920 / 1.0815 | 1.1393 / 1.2000 | 1.3074 / 1.2991 | 1.1750 / 1.2217 | 0.9733 / 0.9897 | 1.0164 / 1.1355 | 1.3053 / 1.2769 |
| KNN | WHIM | 1.0074 / 0.9088 | 1.1947 / 1.2158 | 1.0171 / 0.9445 | 0.9782 / 1.0792 | 1.1499 / 1.2141 | 1.3190 / 1.3189 | 1.1743 / 1.1873 | 1.0473 / 1.0123 | 1.0187 / 1.1004 | 1.2742 / 1.2174 |
| MPNN | GRAPH | 1.3047 / 1.2233 | 1.0740 / 1.1382 | 0.9586 / 0.9216 | 1.0579 / 1.1940 | 1.3639 / 1.4539 | 1.4020 / 1.3336 | 1.0528 / 1.1094 | 1.1417 / 1.2077 | 1.2883 / 1.4815 | 1.6595 / 1.5569 |
| AFP | GRAPH | 1.2615 / 1.1557 | 1.2106 / 1.2335 | 0.8850 / 0.8642 | 1.2019 / 1.3078 | 1.3705 / 1.4229 | 1.3606 / 1.3041 | 1.3104 / 1.4115 | 1.2157 / 1.2252 | 1.3611 / 1.5731 | 1.7060 / 1.5913 |
| MolCLR$_{gcn}$ | GRAPH | 1.2319 / 1.2398 | 1.2528 / 1.2835 | 1.2779 / 1.2919 | 1.0446 / 1.0717 | 1.3564 / 1.3406 | 1.1112 / 1.1396 | 1.3807 / 1.4115 | 1.1568 / 1.2928 | 0.9683 / 1.0252 | 1.8312 / 1.8374 |
| MolCLR$_{gcn}^{pretrained}$ | GRAPH | 1.3950 / 1.4198 | 1.2290 / 1.2562 | 1.3255 / 1.3353 | 1.0275 / 1.0452 | 1.3146 / 1.2956 | 1.2088 / 1.2518 | 1.3378 / 1.3655 | 1.1677 / 1.3067 | 0.9439 / 0.9968 | 1.8971 / 1.9297 |
| PPNP | GRAPH | 1.3204 / 1.2262 | 1.2973 / 1.3029 | 1.1354 / 1.0602 | 1.0675 / 1.1460 | 1.3322 / 1.3214 | 1.4430 / 1.3491 | 1.3400 / 1.4205 | 1.1516 / 1.1805 | 1.0983 / 1.1795 | 1.6043 / 1.5392 |
| MolCLR$_{gin}^{pretrained}$ | GRAPH | 1.6430 / 1.6588 | 1.2987 / 1.3599 | 1.3471 / 1.3572 | 1.3126 / 1.3911 | 1.4144 / 1.3823 | 1.0887 / 1.1027 | 1.3450 / 1.3897 | 1.1978 / 1.3359 | 1.0543 / 1.1123 | 1.8490 / 1.8369 |
| MolCLR$_{gin}$ | GRAPH | 1.6430 / 1.6588 | 1.3137 / 1.3302 | 1.3471 / 1.3572 | 1.3126 / 1.3911 | 1.4144 / 1.3823 | 1.0887 / 1.1027 | 1.4369 / 1.4828 | 1.1978 / 1.3359 | 1.0543 / 1.1123 | 1.8490 / 1.8369 |
| Contextpred$^{pretrained}$ | GRAPH | 1.9750 / 2.1353 | 1.8107 / 1.9078 | 1.3493 / 1.4886 | 1.6154 / 1.8623 | 1.4829 / 1.7134 | 1.6653 / 1.7664 | 1.5641 / 1.7102 | 1.6095 / 2.0372 | 1.8364 / 2.0995 | 1.8936 / 2.0112 |
| Contextpred | GRAPH | 1.8816 / 1.9981 | 1.8158 / 1.9027 | 1.4666 / 1.5984 | 1.7811 / 1.9887 | 1.6644 / 1.9059 | 1.7069 / 1.8184 | 1.6768 / 1.8274 | 1.6665 / 2.0558 | 1.8929 / 2.1681 | 1.9221 / 2.0561 |
| KPGT$^{pretrained}$ | GRAPH | 2.2765 / 2.4227 | 1.8560 / 1.8244 | 1.6561 / 1.6645 | 2.6720 / 2.6705 | 2.1335 / 2.0196 | 1.7889 / 1.8235 | 1.8541 / 1.8355 | 2.4356 / 2.3289 | 2.7528 / 2.6857 | 2.0671 / 2.0234 |
| KPGT | GRAPH | 2.6051 / 2.6221 | 1.6700 / 1.6562 | 1.6763 / 1.7084 | 2.6704 / 2.6356 | 2.1538 / 2.0568 | 1.6908 / 1.7617 | 2.1583 / 2.0943 | 2.3888 / 2.2401 | 2.7510 / 2.6604 | 2.1260 / 2.1025 |

Table 7: RMSE / RMSE$_{cliff}$ (Part 3/3) with best (**bold**) and second-best (underlined) highlighted.

| Algorithm | Descriptor | CHEMBL262 (Ki) | CHEMBL264 (Ki) | CHEMBL2835 (Ki) | CHEMBL287 (Ki) | CHEMBL2971 (Ki) | CHEMBL3979 (EC50) | CHEMBL4005 (Ki) | CHEMBL4203 (Ki) | CHEMBL4616 (EC50) | CHEMBL4792 (Ki) |
|---|---|---|---|---|---|---|---|---|---|---|---|
| GraphCliff | GRAPH | 0.7523 / _0.7019_ | _0.6189_ / _0.6715_ | 0.3959 / 0.7953 | 0.7055 / 0.7977 | 0.6154 / 0.7777 | **0.6226** / **0.6536** | **0.6174** / **0.7122** | 0.8999 / 1.1768 | **0.6342** / 0.7186 | 0.6351 / _0.6513_ |
| GraphCliff w/ JK-Net | GRAPH | **0.7108** / 0.7118 | 0.6424 / 0.6957 | 0.3986 / 0.8506 | **0.7016** / 0.7356 | **0.5893** / 0.7058 | 0.6635 / 0.7087 | 0.6516 / 0.7400 | _0.8724_ / 1.1417 | 0.6631 / 0.7244 | 0.6401 / 0.6595 |
| SVM | ECFP | 0.7236 / **0.6556** | **0.6150** / 0.6741 | 0.4198 / 0.7427 | 0.7142 / 0.8115 | _0.6050_ / 0.6590 | _0.6295_ / _0.6737_ | 0.6462 / 0.7415 | 0.8798 / **1.0014** | _0.6353_ / _0.6924_ | **0.6334** / **0.6381** |
| GraphCLiff w/ Residual | GRAPH | 0.7886 / 0.9097 | 0.6488 / 0.7051 | 0.3903 / 0.8178 | 0.7363 / 0.8150 | 0.6695 / 0.7589 | 0.6951 / 0.7653 | 0.6386 / _0.7277_ | **0.8198** / 1.0360 | 0.6973 / 0.7425 | **0.6332** / 0.6818 |
| GBM | ECFP | 0.7498 / 0.7274 | 0.6486 / 0.7215 | 0.4053 / 0.7893 | 0.7591 / 0.8473 | 0.6157 / 0.6667 | 0.6595 / 0.7217 | 0.6468 / 0.7482 | 0.9194 / 1.0753 | 0.6857 / 0.7675 | 0.6737 / 0.6865 |
| RF | ECFP | _0.7212_ / 0.7749 | 0.6586 / 0.7418 | _0.3876_ / 0.8021 | 0.7758 / 0.8911 | 0.6297 / **0.6428** | 0.6497 / 0.7078 | 0.6475 / 0.7321 | 0.8816 / 1.0809 | 0.6816 / 0.7695 | 0.7086 / 0.7211 |
| GINE+PairNorm | GRAPH | 0.7891 / 0.9355 | 0.6431 / 0.7241 | **0.3840** / 0.7499 | 0.7338 / 0.7459 | 0.6835 / 0.8511 | 0.6308 / 0.6898 | 0.6671 / 0.7668 | 0.9310 / 1.2276 | 0.6916 / 0.7698 | 0.6760 / 0.6968 |
| JK-Net | GRAPH | 0.8369 / 0.8611 | 0.6578 / 0.7186 | 0.4208 / 0.7796 | _0.7023_ / _0.7269_ | 0.6544 / 0.7893 | 0.6838 / 0.7149 | _0.6221_ / 0.7734 | 0.9765 / 1.0937 | 0.7140 / 0.7502 | 0.6805 / 0.6948 |
| GINE+NodeNorm | GRAPH | 0.8071 / 0.8037 | 0.6582 / 0.7159 | 0.4147 / 0.7563 | 0.7331 / 0.7663 | 0.6468 / 0.7698 | 0.6894 / 0.7025 | 0.6738 / 0.7833 | 0.9556 / 1.2305 | 0.6746 / 0.6998 | 0.7496 / 0.7339 |
| GINE+Residual | GRAPH | 0.8593 / 0.8792 | 0.6628 / 0.7499 | 0.4287 / 0.8235 | 0.7421 / 0.7571 | 0.6921 / 0.7991 | 0.6784 / 0.7153 | 0.6498 / 0.7412 | 0.9403 / 1.2689 | 0.6603 / 0.7098 | 0.7617 / 0.7714 |
| KNN | ECFP | 0.8337 / 0.8994 | 0.6739 / 0.8053 | 0.4362 / 0.8584 | 0.8100 / 0.7955 | 0.6634 / 0.7824 | 0.6836 / 0.7389 | 0.6557 / 0.7540 | 0.9718 / 1.0740 | 0.7398 / 0.8282 | 0.6955 / 0.7244 |
| Chemprop | GRAPH | 0.8680 / 1.0281 | 0.6368 / **0.6520** | 0.4331 / 0.7623 | 0.7086 / **0.7153** | 0.7445 / 0.9535 | 0.7108 / 0.7705 | 0.7088 / 0.8083 | 1.0027 / 1.4844 | 0.7036 / 0.7947 | 0.6751 / 0.7131 |
| LSTM | SMILES | 0.7672 / 0.7814 | 0.6652 / 0.7666 | 0.4314 / 0.8404 | 0.7913 / 0.8942 | 0.6887 / 0.8860 | 0.7404 / 0.7903 | 0.7637 / 0.9005 | 0.9071 / 1.3177 | 0.7392 / 0.8308 | 0.6910 / 0.7499 |
| GBM | MACCS | 0.8090 / 0.8785 | 0.6960 / 0.7901 | 0.4807 / 0.9261 | 0.7876 / 0.8361 | 0.6585 / _0.6456_ | 0.6611 / 0.7034 | 0.6755 / 0.7899 | 0.9836 / 1.4244 | 0.7151 / 0.7949 | 0.7564 / 0.7935 |
| SVM | MACCS | 0.8339 / 0.9592 | 0.7205 / 0.8131 | 0.4643 / 0.7648 | 0.7380 / 0.7890 | 0.6568 / 0.6987 | 0.6728 / 0.7146 | 0.7233 / 0.8439 | 0.9817 / 1.4668 | 0.7173 / 0.7795 | 0.7488 / 0.7798 |
| RF | MACCS | 0.8847 / 0.8843 | 0.7379 / 0.8311 | 0.4372 / 0.8238 | 0.7891 / 0.8333 | 0.6369 / 0.6565 | 0.6963 / 0.7267 | 0.7013 / 0.8448 | 0.9292 / 1.3265 | 0.7168 / 0.7721 | 0.7867 / 0.8266 |
| MLP | ECFP | 0.9041 / 0.9484 | 0.6719 / 0.7309 | 0.4878 / 0.8757 | 0.7325 / 0.8519 | 0.6743 / 0.7636 | 0.6610 / 0.7244 | 0.6798 / 0.7694 | 0.9468 / 1.0265 | 0.7268 / 0.7779 | 0.6915 / 0.6821 |
| GINE+DropEdge | GRAPH | 0.9186 / 0.8750 | 0.7444 / 0.7897 | 0.4785 / 0.9170 | 0.8330 / 0.8733 | 0.6790 / 0.8311 | 0.7519 / 0.7878 | 0.7042 / 0.8006 | 0.9499 / 1.1955 | 0.7476 / 0.7609 | 0.8374 / 0.8613 |
| SCAGE | GRAPH | 0.9228 / 0.8449 | 0.7048 / 0.7383 | 0.5052 / 0.6936 | 0.8011 / 0.8126 | 0.7453 / 0.7260 | 0.9212 / 0.8939 | 0.7048 / 0.7698 | 1.0241 / _1.0160_ | 0.7396 / **0.6857** | 0.7808 / 0.8590 |
| GCNII | GRAPH | 0.9118 / 0.9378 | 0.7039 / 0.7747 | 0.4760 / 0.9362 | 0.8044 / 0.8662 | 0.8598 / 0.8850 | 0.7640 / 0.7988 | 0.8057 / 0.8503 | 0.9767 / 1.1498 | 0.7375 / 0.7396 | 0.7916 / 0.8003 |
| KNN | MACCS | 0.9168 / 1.1287 | 0.7698 / 0.8905 | 0.4669 / 0.8819 | 0.8423 / 0.9244 | 0.7316 / 0.6706 | 0.7069 / 0.7492 | 0.7657 / 0.8732 | 1.0308 / 1.5023 | 0.7113 / 0.7821 | 0.8627 / 0.8859 |
| APPNP | GRAPH | 0.9799 / 0.9536 | 0.7942 / 0.8439 | 0.5020 / 0.9794 | 0.7939 / 0.8640 | 0.8402 / 0.8530 | 0.8222 / 0.7956 | 0.7212 / 0.8305 | 1.0061 / 1.2026 | 0.7972 / 0.7732 | 0.8115 / 0.8130 |
| RF | PHYSCHEM | 0.8634 / 0.8645 | 0.8511 / 0.8861 | 0.5017 / 0.8914 | 0.7844 / 0.7866 | 0.8153 / 0.7967 | 0.8731 / 0.8188 | 0.8000 / 0.9034 | 1.0024 / 1.4555 | 0.8165 / 0.8403 | 0.8440 / 0.8257 |
| GBM | PHYSCHEM | 0.8746 / 0.9121 | 0.8779 / 0.9155 | 0.5389 / 0.9050 | 0.8057 / 0.8152 | 0.8588 / 0.8879 | 0.8673 / 0.7997 | 0.7819 / 0.8625 | 1.0136 / 1.5379 | 0.8280 / 0.8525 | 0.8464 / 0.8221 |
| Transformer | TOKENS | 0.9765 / 1.0519 | 0.8221 / 0.8820 | 0.4850 / 0.7722 | 0.8694 / 0.9269 | 0.8262 / 0.9542 | 0.8335 / 0.8804 | 0.8546 / 0.9406 | 0.9594 / 1.1445 | 0.7842 / 0.8165 | 0.9124 / 0.9113 |
| GCN | GRAPH | 0.9337 / 1.0044 | 0.8553 / 0.9102 | 0.5052 / 0.9256 | 0.8861 / 0.9000 | 0.7806 / 0.9167 | 0.8120 / 0.8047 | 0.8752 / 0.9087 | 0.9747 / 1.2114 | 0.8674 / 0.8306 | 0.9226 / 0.9264 |
| KNN | PHYSCHEM | 0.9064 / 0.9357 | 0.8786 / 0.9003 | 0.4907 / 0.8244 | 0.8490 / 0.8381 | 0.7766 / 0.7524 | 0.9549 / 0.8427 | 0.8197 / 0.8410 | 0.9828 / 1.3259 | 0.8384 / 0.8566 | 0.9096 / 0.8561 |
| CNN | SMILES | 0.9477 / 0.9534 | 0.8901 / 0.9151 | 0.5596 / 0.8712 | 0.8908 / 0.9212 | 0.8311 / 0.8609 | 0.9074 / 0.8588 | 0.8381 / 0.9279 | 1.0128 / 1.2306 | 0.8190 / 0.8207 | 0.9673 / 0.9662 |
| SVM | PHYSCHEM | 0.9489 / 1.0021 | 0.9085 / 0.9167 | 0.4135 / **0.6398** | 0.8179 / 0.8092 | 0.9504 / 0.8548 | 0.9186 / 0.8843 | 0.7993 / 0.8523 | 1.0052 / 1.2213 | 0.8413 / 0.8255 | 0.8697 / 0.8703 |
| GAT | GRAPH | 0.9943 / 1.0323 | 0.8958 / 0.9356 | 0.5554 / 0.9240 | 0.9470 / 0.9944 | 0.8026 / 0.9663 | 0.9232 / 0.9139 | 0.8613 / 0.9013 | 1.0039 / 1.2080 | 0.8734 / 0.8351 | 1.0036 / 1.0141 |
| RF | WHIM | 0.9294 / 1.0229 | 0.9363 / 0.9698 | 0.4783 / 0.7437 | 0.9168 / 1.0236 | 0.7499 / 0.8256 | 0.9965 / 0.9384 | 0.8845 / 0.9045 | 0.9974 / 1.1101 | 0.9122 / 0.8879 | 1.0399 / 1.0222 |
| GBM | WHIM | 0.9356 / 1.0257 | 0.9700 / 1.0206 | 0.5030 / 0.7909 | 0.9407 / 1.0316 | 0.7812 / 0.8209 | 1.0372 / 0.9752 | 0.8817 / 0.9400 | 1.0340 / 1.2519 | 0.9515 / 0.9042 | 1.0398 / 1.0337 |
| SVM | WHIM | 0.8984 / 0.9927 | 0.9743 / 1.0097 | 0.5116 / 0.8028 | 0.9465 / 1.0426 | 0.8666 / 0.9746 | 1.0445 / 1.0198 | 0.9007 / 0.9405 | 1.0251 / 1.1688 | 0.9100 / 0.9003 | 1.0910 / 1.0968 |
| KNN | WHIM | 0.9133 / 0.8676 | 0.9742 / 1.0089 | 0.5335 / 0.9013 | 0.9741 / 1.0728 | 0.8099 / 0.8314 | 1.0187 / 0.9775 | 0.9285 / 0.9975 | 1.0704 / 1.3052 | 0.8847 / 0.8458 | 1.0522 / 1.0205 |
| MPNN | GRAPH | 1.0213 / 1.0356 | 1.0822 / 1.0124 | 0.6679 / 1.0670 | 0.9267 / 0.9732 | 0.9727 / 0.9446 | 1.1831 / 1.1452 | 0.9978 / 1.0160 | 1.0564 / 1.1486 | 0.9345 / 0.8603 | 1.1221 / 1.1141 |
| AFP | GRAPH | 1.1158 / 1.1835 | 1.1024 / 1.0617 | 0.7473 / 1.1064 | 1.1485 / 1.2151 | 1.0907 / 1.1009 | 1.0797 / 0.9897 | 1.0592 / 1.0789 | 1.0617 / 1.1308 | 0.9467 / 0.8722 | 1.2335 / 1.2376 |
| MolCLR$_{gcn}$ | GRAPH | 1.2110 / 1.0167 | 1.0812 / 1.0247 | 1.1784 / 0.9831 | 0.8821 / 0.8618 | 1.7887 / 1.7890 | 1.0663 / 0.9434 | 1.0722 / 1.0537 | 1.0671 / 1.4879 | 0.9018 / 0.8160 | 1.2255 / 1.2214 |
| MolCLR$_{gcn}^{pretrained}$ | GRAPH | 1.2197 / 1.0901 | 1.0675 / 1.0304 | 1.3432 / 1.1817 | 0.8728 / 0.8503 | 1.8176 / 1.8397 | 1.0532 / 0.9213 | 1.0446 / 1.0433 | 1.0725 / 1.5024 | 0.8989 / 0.8162 | 1.3483 / 1.3562 |
| PPNP | GRAPH | 1.0920 / 1.1605 | 1.0769 / 1.0153 | 0.9767 / 1.3259 | 1.0419 / 1.1170 | 1.3838 / 1.3053 | 1.1359 / 1.0032 | 1.0788 / 1.0877 | 1.0695 / 1.0985 | 0.9291 / 0.8418 | 1.1759 / 1.1685 |
| MolCLR$_{gin}^{pretrained}$ | GRAPH | 1.1739 / 1.0662 | 1.0812 / 1.0247 | 1.0659 / 0.9023 | 0.8728 / 0.8503 | 1.2315 / 1.1016 | 0.9871 / 0.8284 | 1.1207 / 1.1324 | 1.0580 / 1.4872 | 0.9090 / 0.8243 | 1.4978 / 1.4986 |
| MolCLR$_{gin}$ | GRAPH | 1.1739 / 1.0662 | 1.1757 / 1.1259 | 1.0659 / 0.9023 | 1.0022 / 0.9877 | 1.2315 / 1.1016 | 0.9871 / 0.8284 | 1.1207 / 1.1324 | 1.0345 / 1.4518 | 0.9604 / 0.8568 | 1.4978 / 1.4986 |
| Contextpred$^{pretrained}$ | GRAPH | 1.9809 / 2.0611 | 1.6605 / 1.5609 | 1.5424 / 1.0246 | 1.5156 / 1.5724 | 1.7642 / 1.6268 | 1.6231 / 1.8675 | 1.5573 / 1.4940 | 1.7366 / 2.0266 | 1.2383 / 1.3317 | 1.7062 / 1.7833 |
| Contextpred | GRAPH | 1.9713 / 2.0585 | 1.5504 / 1.4692 | 1.6920 / 1.1823 | 1.3156 / 1.3855 | 1.4843 / 1.4337 | 1.7480 / 1.9870 | 1.5978 / 1.5115 | 1.9036 / 2.0851 | 1.6043 / 1.6481 | 1.7222 / 1.8009 |
| KPGT$^{pretrained}$ | GRAPH | 2.6248 / 2.7108 | 1.3685 / 1.4650 | 0.5715 / 0.7474 | 1.6315 / 1.6767 | 1.3323 / 1.4566 | 2.3793 / 2.3392 | 1.5268 / 1.6350 | 2.5748 / 2.7290 | 1.3843 / 1.4016 | 2.1907 / 2.1513 |
| KPGT | GRAPH | 2.5337 / 2.6635 | 1.5136 / 1.5888 | 0.6072 / _0.6824_ | 1.4728 / 1.5559 | 1.3366 / 1.4283 | 2.1415 / 2.0541 | 1.5060 / 1.5896 | 2.5364 / 2.6396 | 1.4745 / 1.4685 | 2.3039 / 2.2114 |

## 8.4 ALL RESULTS OF NINE DATASETS IN LSSNS

Tables 8 and 9 present the complete results on all nine LSSNS datasets. We report both RMSE and $\text{RMSE}_{\text{cliff}}$ for baseline machine learning models, graph-based neural networks, and Graph-Cliff. These results provide a detailed view of model performance in small-sample, narrow-scaffold regimes, highlighting the challenges posed by limited data diversity and the relative robustness of different approaches. Table 10 provides a mapping between LSSNS targets and similar MoleculeACE datasets, enabling cross-dataset comparison and transfer evaluation.

Table 8: Comparison of performance (RMSE) across protein targets in LSSNS.

| Algorithm | Descriptor | USP7 | RIP2 | PKC$\iota$ | PHGDH | PLK1 | IDO1 | RXFP1 | BRAF | mGluR2 |
|---|---|---|---|---|---|---|---|---|---|---|
| GCN | GRAPH | 0.5419 | 0.7355 | 0.8603 | 1.0886 | 0.6306 | 0.6485 | 0.6747 | 0.4889 | 0.4228 |
| GAT | GRAPH | 0.5062 | 0.7870 | 0.8039 | 0.4396 | 0.4873 | 0.6595 | 0.6333 | 0.5551 | 0.4412 |
| AFP | GRAPH | 0.5080 | 0.7947 | 1.1758 | 0.4010 | 0.5048 | 0.6555 | 0.4803 | 0.4771 | 0.3578 |
| MPNN | GRAPH | 0.5833 | 0.7779 | 0.9875 | 1.1800 | 0.5036 | 0.6287 | 0.6349 | 0.4436 | 0.4558 |
| SVM | ECFP | 0.5350 | 0.5787 | 0.8224 | 0.6174 | 0.5026 | 0.7858 | 0.4181 | 0.4120 | 0.2927 |
| MLP | ECFP | 0.5082 | 0.5875 | 0.8188 | 0.6865 | 0.4293 | 0.7047 | 0.4234 | 0.3778 | 0.3260 |
| GraphCliff | GRAPH | 0.5181 | 0.4824 | 1.8550 | 1.2460 | 0.6134 | 0.7354 | 0.6909 | 0.4557 | 0.4463 |
| Transferred GraphCliff | GRAPH | 0.3409 | 0.5476 | 0.6478 | – | 0.4837 | – | 0.6537 | 0.4550 | 0.2884 |

Table 9: Comparison of performance ($\text{RMSE}_{\text{cliff}}$) across protein targets in LSSNS.

| Algorithm | Descriptor | USP7 | RIP2 | PKC$\iota$ | PHGDH | PLK1 | IDO1 | RXFP1 | BRAF | mGluR2 |
|---|---|---|---|---|---|---|---|---|---|---|
| GCN | GRAPH | 0.4338 | 0.6928 | 1.4547 | 1.2228 | 0.5635 | 0.6335 | 0.8078 | 0.7702 | 0.5221 |
| GAT | GRAPH | 0.5759 | 0.6655 | 1.3094 | 0.6524 | 0.4489 | 0.7711 | 0.7475 | 0.8501 | 0.5134 |
| AFP | GRAPH | 0.6049 | 0.7494 | 1.9475 | 0.6833 | 0.4461 | 0.7975 | 0.5126 | 0.6437 | 0.3853 |
| MPNN | GRAPH | 0.4499 | 0.6920 | 1.6844 | 1.2934 | 0.4217 | 0.5570 | 0.7389 | 0.6052 | 0.5605 |
| SVM | ECFP | 0.6939 | 0.6234 | 1.3064 | 0.8643 | 0.4099 | 0.9141 | 0.5158 | 0.6290 | 0.3117 |
| MLP | ECFP | 0.5824 | 0.6667 | 1.3662 | 0.9428 | 0.4078 | 0.8142 | 0.5173 | 0.4697 | 0.3434 |
| GraphCliff | GRAPH | 0.5322 | 0.5665 | 1.4120 | 1.1136 | 0.5711 | 0.7090 | 0.8589 | 0.6860 | 0.5380 |
| Transferred GraphCliff | GRAPH | 0.4350 | 0.4818 | 0.9259 | – | 0.4885 | – | 0.8121 | 0.6950 | 0.3515 |

Table 10: Mapping between LSSNS protein targets and similar MoleculeACE datasets.

| LSSNS Target | Class | Similar MoleculeACE datasets (Class) |
|---|---|---|
| USP7 | Protease (cysteine protease) (UniProt, p) | CHEMBL204_Ki (Thrombin, serine protease) (UniProt, c) |
| | | CHEMBL244_Ki (Factor X, serine protease) (UniProt, d) |
| RIP2 | Kinase (Ser/Thr kinase) (UniProt, b) | CHEMBL2147_Ki (PIM1, Ser/Thr kinase) (UniProt, f) |
| | | CHEMBL262_Ki (GSK3$\beta$, Ser/Thr kinase) (UniProt, m) |
| PKC$\iota$ | Kinase (Ser/Thr kinase) (UniProt, l) | CHEMBL2147_Ki (PIM1, Ser/Thr kinase) (UniProt, f) |
| | | CHEMBL262_Ki (GSK3$\beta$, Ser/Thr kinase) (UniProt, m) |
| PHGDH | Other enzyme (oxidoreductase) (UniProt, a) | – |
| PLK1 | Kinase (Ser/Thr kinase) (UniProt, n) | CHEMBL2147_Ki (PIM1, Ser/Thr kinase) (UniProt, f) |
| | | CHEMBL262_Ki (GSK3$\beta$, Ser/Thr kinase) (UniProt, m) |
| IDO1 | Other enzyme (oxidoreductase) (UniProt, g) | – |
| RXFP1 | GPCR (Class A) (UniProt, q) | CHEMBL214_Ki (5-HT1A, class A) (UniProt, e) |
| | | CHEMBL219_Ki (D4, class A) (UniProt, i) |
| | | CHEMBL231_Ki (Histamine H1, class A) (UniProt, j) |
| | | CHEMBL234_Ki (D3, class A) (UniProt, k) |
| BRAF | Kinase (Ser/Thr kinase) (UniProt, h) | CHEMBL2147_Ki (PIM1, Ser/Thr kinase) (UniProt, f) |
| | | CHEMBL262_Ki (GSK3$\beta$, Ser/Thr kinase) (UniProt, m) |
| mGluR2 | GPCR (Class C) (UniProt, o) | CHEMBL214_Ki (5-HT1A, class A) (UniProt, e) |
| | | CHEMBL219_Ki (D4, class A) (UniProt, i) |
| | | CHEMBL231_Ki (Histamine H1, class A) (UniProt, j) |
| | | CHEMBL234_Ki (D3, class A) (UniProt, k) |

## 8.5 ABLATION STUDY

Table 11 summarizes the effects of ablating short- and long-range filters, gating, and pooling strategies. Removing any component leads to substantial performance degradation, while replacing SAG-Pool with simple pooling further increases error, underscoring the importance of each design choice. Table 12 reports the performance of different GNN variants used in the short- and long-range filter components. Across all tested combinations, the configuration with GINE as the short-range filter and Chebyshev polynomials as the long-range operator consistently achieved the best performance in terms of both RMSE and $RMSE_{cliff}$.

Table 11: Performance comparison across different module ablations and pooling methods.

| Short | Long | Gating | Pooling | RMSE | $\Delta$RMSE (%) | $RMSE_{cliff}$ | $\Delta RMSE_{cliff}$ |
|---|---|---|---|---|---|---|---|
| O | O | O | SAGPool | 0.673 | – | 0.766 | – |
| O | O | – | SAGPool | 0.725 | +7.7% | 0.798 | +4.2% |
| O | – | O | SAGPool | 0.856 | +27.2% | 0.933 | +21.8% |
| – | O | O | SAGPool | 1.288 | +91.3% | 1.287 | +68.0% |
| O | – | – | SAGPool | 1.001 | +48.6% | 1.038 | +35.6% |
| – | O | – | SAGPool | 1.327 | +97.2% | 1.314 | +71.6% |
| – | – | O | SAGPool | 1.361 | +102.2% | 1.286 | +67.9% |
| O | O | O | Max | 0.811 | +20.5% | 0.871 | +13.7% |
| O | O | O | Mean | 0.874 | +29.9% | 0.950 | +24.0% |
| O | O | O | Sum | 0.963 | +43.1% | 1.024 | +33.7% |

Table 12: Comparison of different GNN types used in the short- and long-range filters. Bold rows correspond to our default configuration (Short:GINE + Long:Chebyshev), which achieved the best overall performance.

| Short | Long | RMSE | $RMSE_{cliff}$ | Short | Long | RMSE | $RMSE_{cliff}$ |
|---|---|---|---|---|---|---|---|
| GCN | GCN | 0.713 | 0.798 | GAT | GCN | 0.695 | 0.786 |
| | GIN | 0.712 | 0.791 | | GIN | 0.703 | 0.784 |
| | GAT | 0.692 | 0.780 | | GAT | 0.706 | 0.794 |
| | Chebyshev | 0.724 | 0.819 | | Chebyshev | 0.689 | 0.778 |
| GIN | GCN | 0.704 | 0.792 | GINE | GCN | 0.715 | 0.803 |
| | GIN | 0.710 | 0.799 | | GIN | 0.694 | 0.774 |
| | GAT | 0.699 | 0.795 | | GAT | 0.696 | 0.778 |
| | Chebyshev | 0.688 | 0.777 | | **Chebyshev** | **0.673** | **0.766** |

## 8.6 THEORETICAL AND EMPIRICAL EXAMPLES OF 1-WL GNN LIMITATIONS

==A new subsection has been added.==

In the benchmark datasets used in the study, activity cliff pairs were defined based on structural similarity together with a large activity discrepancy. Three complementary similarity notions were considered: (i) substructure similarity, computed as the Tanimoto coefficient on ECFPs to capture shared atom-centered radial environments, (ii) scaffold similarity, obtained from ECFPs restricted to Bemis–Murcko scaffolds to detect differences in core frameworks, and (iii) SMILES similarity, measured using Levenshtein distance to capture localized edit operations in the string representation. Pairs exhibiting at least a 10-fold Ki difference under any of these similarity measures were labeled as activity cliffs.

Theoretically, such cliff pairs create instances that are difficult for standard 1-WL GNNs. Consider two molecules differing only by a single atom substitution at one node while all other atoms, bonds, and neighborhoods remain unchanged. In this setting, the substituted node possesses the same multiset of neighbor features and structural context, resulting in identical message-passing updates at each layer. Consequently, the two molecules remain indistinguishable to standard message-passing GNNs, even though the small local perturbation yields a substantial change in biological activity.

In the first case in Figure 7a, a Levenshtein-type cliff arises from a small local modification where two hydrogens are replaced by a double-bonded oxygen, yet the potency changes sharply from 4.1 Ki to 127.0 Ki. In Figure 7b, the target (indicated by the star) and the cliff-pair molecule in the ChemProp embedding space (indicated by the green triangle) appear in nearly the same position, revealing ChemProp's difficulty in distinguishing this subtle atom-level perturbation. In contrast, the distance between the target and the cliff-pair molecule is slightly larger in the GraphCliff embedding space (shown as the blue circle). The ground-truth target label, converted to $-\log_{10}(\mathrm{Ki})$ for the 4.1 Ki molecule, is $-0.6128$. GraphCliff predicts $-0.5936$, whereas ChemProp predicts $-1.2879$.

The second case in Figure 8 represents a scaffold-level cliff with a large potency gap (220.0 Ki versus 7.4 Ki). In the ChemProp embedding space, the target and the cliff-pair molecule appear almost indistinguishable, whereas GraphCliff places them slightly farther apart, capturing the scaffold-level perturbation more effectively. The ground-truth label for the 220.0 Ki molecule is $-2.3424$, and GraphCliff provides a closer prediction ($-2.3631$) than ChemProp ($-2.1282$).

In both cases, ChemProp embeds the molecules extremely close, reflecting its difficulty in distinguishing the structural change. In contrast, GraphCliff assigns a larger embedding distance and produces a more faithful prediction of the activity gap, indicating higher sensitivity to the underlying structural change.

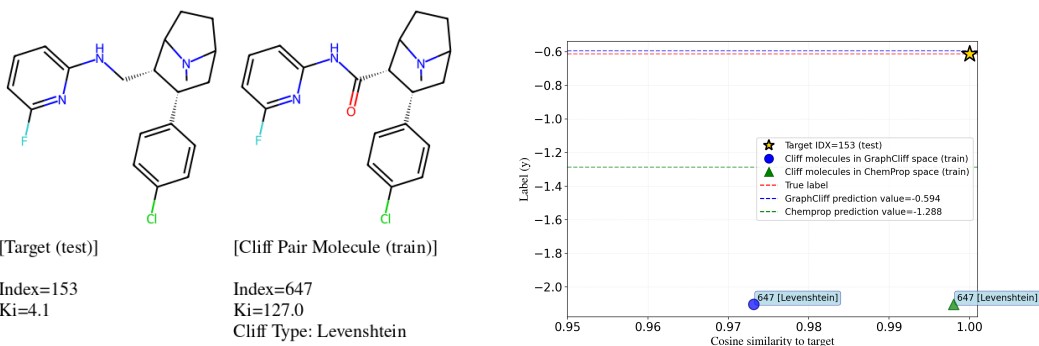

(a) Molecular structures and local modification.  (b) Embedding similarity of target and cliff molecule.

Figure 7: **Levenshtein activity cliff in CHEMBL238 (Ki).** Levenshtein type activity cliff pair showing the target (test) and cliff (train) molecules with their embedding positions and model predictions. ChemProp maps the pair almost identically, whereas GraphCliff assigns a larger separation and provides a more accurate prediction.

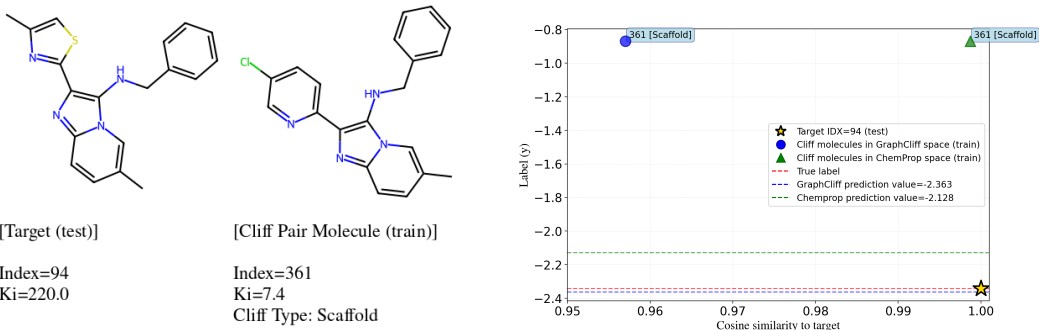

(a) Molecular structures and scaffold-level change.  (b) Embedding similarity of target and cliff molecule.

Figure 8: **Scaffold activity cliff in CHEMBL2047 (EC50).** Scaffold activity cliff pair with embedding positions and predicted activities for the target (test) and cliff (train) molecules. ChemProp places the pair extremely close, while GraphCliff yields a clearer separation and more accurate prediction.

