# OpenReview forum: "GraphCliff: Short-Long Range Gating for Subtle Differences but Critical Changes"
_ICLR.cc/2026/Conference — Submitted to ICLR 2026_

### Official Review · Reviewer_mPRp · 2025-10-16

**Soundness:** 2
**Presentation:** 3
**Contribution:** 2
**Rating:** 2
**Confidence:** 2

**Summary:**

This paper proposes GraphCliff, a graph neural model for QSAR tasks with activity cliffs. The core idea is to explicitly combine short-range message passing (i.e. GINE) with long-range propagation (i.e. Chebnet) via a learnable sigmoid gate, followed by attention-based pooling for graph-level prediction. The authors motivate the work by showing that standard GNN embeddings under-emphasize local substructure differences compared to ECFP dissimilarities; GraphCliff is intended to preserve local sensitivity while providing global context. Experiments show lower RMSE overall and ablations indicate both filters and the gate contribute to performance.

**Strengths:**

S1: The paper targets activity cliffs, a well-known failure mode for deep models and matter for medicinal chemistry that generic molecular GNNs miss.

S2: The empirical study is extensive, with results on 30 MoleculeACE targets, along with targeted ablations such as removing short/long paths and gating, hop-wise sensitivity and Dirichlet energy estimation, which examine not just accuracy but also how information propagates through the network.

S3: The paper is well organized and readable. The modular architecture (short-range path, long-range path, and gating) is explained clearly, and the figures/equations make the design easy to follow.

**Weaknesses:**

W1: Section 5.2’s hop-wise sensitivity (perturbation of u affecting v at k hops) and Jacobian spectrum are closer to an over-squashing analysis than smoothing. Although Dirichlet energy is included for smoothing, the text and framing mix these notions, and it’s unclear whether the proposed gains primarily combat squashing (limited receptive field / information bottlenecks) or smoothing (Laplacian averaging).

W2: Necessity of the mechanism vs. simpler baselines. If over-smoothing were the key issue motivating deep propagation, residual/skip connections, PairNorm/BatchNorm, JK-Net and APPNP/PPNP are standard, lightweight fixes. The paper does not compare GraphCliff against these strong anti over-smoothing baselines or against simple residual GNNs. This weakens the claim that the proposed mechanism is required.

W3: Chebyshev long-range propagation, gated/highway mechanisms, and attention pooling have all been in literature for other application; GINE is standard for molecular graphs. The specific combination is practical but feels like engineering glue more than a new principle.

W4: Formal problem setup & notations are missing. There is no concise notation paragraph formally defining the input graph G=(V,E), node/edge features, and the prediction problem; the method section jumps directly into components/equations, which hurts clarity.

**Questions:**

Q1: Could you clarify why do deep GNNss are needed for this application ?

Q2: What are the training/inference runtime and memory costs of GraphCliff relative to strong baselines? How do these scale with graph size and Chebyshev order?

Q3: Did you try residual connections to the input or JK aggregation in lieu of gated fusion? Even a small study showing why they underperform would clarify the design choice.

Q4: Could you provide a few negative cases where GraphCliff fails on known cliff pairs and analyze whether the failure is due to insufficient local sensitivity, long-range diffusion, or data sparsity? This would sharpen the pros and cons of GraphCliff

---

> ### Author Response · Authors · 2025-11-22
>
> # W1.
>
> Thank you for this insightful comment. We acknowledge the confusion in our analysis and appreciate the opportunity to clarify. You are correct that hop-wise sensitivity and Jacobian spectrum analysis are associated with oversquashing. We agree that our previous framing was misleading. Our primary concern is oversmoothing in the context of activity cliff prediction, where the task is to distinguish subtle local substructural differences between structurally similar molecules. Since this is fundamentally a local feature discrimination problem rather than a long-range information propagation issue, oversquashing is not the primary bottleneck in this particular problem. To better align our analysis with the actual problem, we replaced Section 5.2's hop-wise sensitivity and Jacobian-based metrics with Mean Average Distance (MAD) refer to Chen et al., which directly measure the distinguishability of node representations. Thank you for helping us improve the rigor and clarity of our manuscript.
>
> Chen, Deli, et al. "Measuring and relieving the over-smoothing problem for graph neural networks from the topological view." Proceedings of the AAAI conference on artificial intelligence. Vol. 34. No. 04. 2020.
>
>
> # W2 and Q3.
>
> We appreciate the reviewer’s suggestion and fully agree that it is important to assess whether GraphCliff’s gated fusion is truly necessary beyond standard anti–oversmoothing techniques. To address this, we conducted additional experiments comparing GraphCliff with widely used oversmoothing-mitigation baselines, as well as simplified variants of our model in which the gated fusion is replaced by residual or JK aggregation.
>
> Specifically, we evaluate the following models across the same 30 ChEMBL regression tasks:
> - **GraphCliff** (proposed)
> - **GraphCliff w/ JK-Net** (replacing gated fusion with JK aggregation)
> - **GraphCliff w/ Residual** (adding residual connections instead of gating)
> - **GINE+PairNorm**, **GINE+NodeNorm**, **JK-Net**, **GINE+Residual**, **GCNII**, **GINE+DropEdge**, **APPNP**, **PPNP**
>
> Below, we summarize the average ranks over all datasets. Also, the complete RMSE and RMSE_cliff tables are provided.
>
> ### Average Rank (RMSE)
>
> | Model                    | Avg. Rank |
> |--------------------------|-----------|
> | **GraphCliff**           | **1.83**  |
> | GraphCliff w/ JK-Net     | 2.53      |
> | GraphCliff w/ Residual   | 3.80      |
> | GINE + PairNorm          | 4.40      |
> | GINE + NodeNorm          | 4.90      |
> | JK-Net                   | 4.93      |
> | GINE + Residual          | 6.00      |
> | GCNII                    | 8.27      |
> | GINE + DropEdge          | 8.60      |
> | APPNP                    | 9.73      |
> | PPNP                     | 11.00     |
>
>
> ### Average Rank (RMSE_cliff)
>
> | Model                    | Avg. Rank |
> |--------------------------|-----------|
> | **GraphCliff**           | **2.50**  |
> | GraphCliff w/ JK-Net     | 3.20      |
> | GraphCliff w/ Residual   | 3.90      |
> | GINE + NodeNorm          | 4.57      |
> | JK-Net                   | 4.77      |
> | GINE + PairNorm          | 4.93      |
> | GINE + Residual          | 5.67      |
> | GCNII                    | 7.63      |
> | GINE + DropEdge          | 8.60      |
> | APPNP                    | 9.50      |
> | PPNP                     | 10.73     |
>
> These results show that GraphCliff consistently achieves strong performance in both RMSE and cliff_rmse. While anti-oversmoothing techniques can achieve comparable performance in some cases, they still fall short of GraphCliff. This suggests that alleviating oversmoothing alone is insufficient for activity cliff prediction. The task requires a node-level mechanism that can adaptively regulate long-range propagation based on each atom’s local structural sensitivity, which is precisely what GraphCliff’s gated fusion is designed to achieve. We believe these additional results clarify the necessity of the proposed mechanism and the limitations of simpler alternatives. We will include the full comparison table, incorporating all anti-oversmoothing baselines, in the revised manuscript for completeness.

---

> ### Author Response · Authors · 2025-11-22
>
> ### RMSE
>
>
> | dataset | GraphCliff | GraphCliff w/ JK-Net | GraphCliff w/ Residual | GINE+PairNorm | GINE+NodeNorm | JK-Net | GINE+Residual | GCNII | GINE+DropEdge | APPNP | PPNP |
> |--------|------------|----------------------|-------------------------|----------------|----------------|--------|----------------|--------|----------------|--------|--------|
> | CHEMBL1862_Ki | **0.7810** | 0.8194 | 0.8148 | 0.8408 | 0.9213 | 0.9715 | 0.8660 | 0.9601 | 1.0060 | 1.0084 | 1.4631 |
> | CHEMBL1871_Ki | 0.6279 | **0.6042** | 0.6615 | 0.6301 | 0.6720 | 0.6400 | 0.6778 | 0.7403 | 0.7704 | 0.7624 | 1.1400 |
> | CHEMBL2034_Ki | 0.7473 | 0.7718 | 0.7369 | 0.7245 | 0.7187 | **0.7143** | 0.7300 | 0.7593 | 0.7485 | 0.7650 | 1.0041 |
> | CHEMBL2047_EC50 | 0.5924 | 0.6289 | 0.6177 | 0.6125 | 0.6542 | **0.5921** | 0.6696 | 0.7248 | 0.7415 | 0.7595 | 1.2564 |
> | CHEMBL204_Ki | **0.6914** | 0.6947 | 0.7126 | 0.7451 | 0.7376 | 0.7783 | 0.8000 | 0.8258 | 0.8658 | 0.8719 | 1.5242 |
> | CHEMBL2147_Ki | 0.5600 | **0.5581** | 0.5816 | 0.6933 | 0.7788 | 0.6663 | 0.7111 | 1.0474 | 0.7846 | 1.1075 | 1.9766 |
> | CHEMBL214_Ki | **0.6210** | 0.6302 | 0.6415 | 0.7222 | 0.6806 | 0.6665 | 0.7098 | 0.7543 | 0.7644 | 0.8000 | 1.1338 |
> | CHEMBL218_EC50 | 0.6974 | 0.7252 | 0.7669 | **0.6903** | 0.7653 | 0.7348 | 0.7265 | 0.7576 | 0.8521 | 0.8527 | 1.0523 |
> | CHEMBL219_Ki | 0.6636 | **0.6634** | 0.7052 | 0.7198 | 0.7021 | 0.7400 | 0.7472 | 0.8156 | 0.8325 | 0.8865 | 1.0711 |
> | CHEMBL228_Ki | **0.6513** | 0.6672 | 0.7010 | 0.6931 | 0.7017 | 0.7129 | 0.7080 | 0.7838 | 0.8104 | 0.8442 | 1.1956 |
> | CHEMBL231_Ki | **0.7080** | 0.7292 | 0.7155 | 0.7292 | 0.7566 | 0.7756 | 0.7953 | 0.8574 | 0.7923 | 0.8659 | 1.3204 |
> | CHEMBL233_Ki | 0.7860 | **0.7642** | 0.7929 | 0.7960 | 0.7912 | 0.7766 | 0.7937 | 0.8372 | 0.8611 | 0.9045 | 1.2973 |
> | CHEMBL234_Ki | 0.6175 | **0.5991** | 0.6331 | 0.6819 | 0.6717 | 0.6553 | 0.6940 | 0.7294 | 0.7518 | 0.7938 | 1.1354 |
> | CHEMBL235_EC50 | 0.6504 | 0.6551 | 0.6513 | **0.6473** | 0.6563 | 0.6716 | 0.6882 | 0.6928 | 0.7117 | 0.7608 | 1.0675 |
> | CHEMBL236_Ki | 0.6965 | **0.6873** | 0.7023 | 0.7575 | 0.7300 | 0.7412 | 0.7357 | 0.7863 | 0.8138 | 0.8622 | 1.3322 |
> | CHEMBL237_EC50 | **0.7075** | 0.7294 | 0.7776 | 0.7571 | 0.7381 | 0.7637 | 0.7709 | 1.0908 | 0.8950 | 0.9286 | 1.4430 |
> | CHEMBL237_Ki | 0.7046 | **0.6985** | 0.7174 | 0.7384 | 0.7279 | 0.7474 | 0.7528 | 0.7692 | 0.7821 | 0.8286 | 1.3400 |
> | CHEMBL238_Ki | **0.5966** | 0.6102 | 0.6006 | 0.6578 | 0.6663 | 0.6437 | 0.7026 | 0.7274 | 0.7444 | 0.7846 | 1.1516 |
> | CHEMBL239_EC50 | **0.6701** | 0.6728 | 0.7263 | 0.7331 | 0.6928 | 0.7858 | 0.7095 | 0.7561 | 0.7924 | 0.8233 | 1.0983 |
> | CHEMBL244_Ki | **0.6683** | 0.6830 | 0.6826 | 0.7861 | 0.7576 | 0.7580 | 0.7655 | 0.8574 | 0.8467 | 0.9900 | 1.6043 |
> | CHEMBL262_Ki | 0.7523 | **0.7108** | 0.7886 | 0.7891 | 0.8071 | 0.8369 | 0.8593 | 0.9118 | 0.9186 | 0.9799 | 1.0920 |
> | CHEMBL264_Ki | **0.6189** | 0.6424 | 0.6488 | 0.6431 | 0.6582 | 0.6578 | 0.6628 | 0.7039 | 0.7444 | 0.7942 | 1.0769 |
> | CHEMBL2835_Ki | 0.3959 | 0.3986 | 0.3903 | **0.3840** | 0.4147 | 0.4208 | 0.4287 | 0.4760 | 0.4785 | 0.5020 | 0.9767 |
> | CHEMBL287_Ki | 0.7055 | **0.7016** | 0.7363 | 0.7338 | 0.7331 | 0.7023 | 0.7421 | 0.8044 | 0.8330 | 0.7939 | 1.0419 |
> | CHEMBL2971_Ki | 0.6154 | **0.5893** | 0.6695 | 0.6835 | 0.6468 | 0.6544 | 0.6921 | 0.8598 | 0.6790 | 0.8402 | 1.3838 |
> | CHEMBL3979_EC50 | **0.6226** | 0.6635 | 0.6951 | 0.6308 | 0.6894 | 0.6838 | 0.6784 | 0.7640 | 0.7519 | 0.7956 | 1.1359 |
> | CHEMBL4005_Ki | **0.6174** | 0.6516 | 0.6386 | 0.6671 | 0.6738 | 0.6221 | 0.6498 | 0.8057 | 0.7042 | 0.7212 | 1.0788 |
> | CHEMBL4203_Ki | 0.9000 | 0.8724 | **0.8198** | 0.9310 | 0.9556 | 0.9765 | 0.9403 | 0.9767 | 0.9499 | 1.0061 | 1.0695 |
> | CHEMBL4616_EC50 | **0.6342** | 0.6631 | 0.6973 | 0.6916 | 0.6746 | 0.7140 | 0.6603 | 0.7375 | 0.7476 | 0.7972 | 0.9291 |
> | CHEMBL4792_Ki | 0.6351 | 0.6401 | **0.6332** | 0.6760 | 0.7496 | 0.6805 | 0.7617 | 0.7916 | 0.8374 | 0.8130 | 1.1759 |

---

> ### Author Response · Authors · 2025-11-22
>
> ### RMSE_cliff
>
>
> | dataset | GraphCliff | GraphCliff w/ JK-Net | GraphCliff w/ Residual | GINE+NodeNorm | JK-Net | GINE+PairNorm | GINE+Residual | GCNII | GINE+DropEdge | APPNP | PPNP |
> |--------|------------|----------------------|-------------------------|----------------|--------|----------------|----------------|--------|----------------|--------|--------|
> | CHEMBL1862_Ki | **0.6740** | 0.7496 | 0.7696 | 0.8562 | 0.9612 | 0.7460 | 0.7848 | 0.9193 | 0.9994 | 0.9476 | 1.4367 |
> | CHEMBL1871_Ki | 0.7975 | **0.7342** | 0.7796 | 0.8341 | 0.8227 | 0.8670 | 0.7810 | 0.9383 | 0.9536 | 0.9685 | 1.3332 |
> | CHEMBL2034_Ki | 0.8580 | 0.9196 | 0.8556 | 0.8524 | 0.8245 | 0.8453 | 0.8964 | 0.8566 | 0.9009 | 0.8998 | 0.9308 |
> | CHEMBL2047_EC50 | 0.6549 | 0.7330 | 0.6779 | 0.7203 | **0.5779** | 0.5953 | 0.6731 | 0.7001 | 0.7301 | 0.7709 | 0.9284 |
> | CHEMBL204_Ki | **0.8212** | 0.8226 | 0.8479 | 0.8635 | 0.8997 | 0.8983 | 0.9378 | 0.9342 | 0.9784 | 0.9734 | 1.7086 |
> | CHEMBL2147_Ki | **0.5790** | 0.6186 | 0.6362 | 0.6964 | 0.6807 | 0.6793 | 0.7330 | 0.9713 | 0.8020 | 1.0046 | 1.8805 |
> | CHEMBL214_Ki | **0.7260** | 0.7485 | 0.7612 | 0.7934 | 0.7683 | 0.8089 | 0.8114 | 0.8342 | 0.8741 | 0.8887 | 1.1517 |
> | CHEMBL218_EC50 | 0.7811 | 0.7997 | 0.8055 | 0.7763 | 0.7825 | **0.7711** | 0.7927 | 0.8331 | 0.8753 | 0.8863 | 0.9986 |
> | CHEMBL219_Ki | 0.7435 | **0.7328** | 0.7818 | 0.7969 | 0.8127 | 0.7691 | 0.8188 | 0.8606 | 0.8942 | 0.9236 | 1.0656 |
> | CHEMBL228_Ki | **0.6739** | 0.6902 | 0.6994 | 0.7565 | 0.7710 | 0.7139 | 0.7927 | 0.8280 | 0.8577 | 0.8787 | 1.1603 |
> | CHEMBL231_Ki | 0.8660 | 0.8745 | 0.8663 | **0.8443** | 0.8577 | 0.7855 | 0.8026 | 0.7778 | 0.8716 | 0.8776 | 1.2262 |
> | CHEMBL233_Ki | 0.8803 | **0.8663** | 0.8847 | 0.8769 | 0.8589 | 0.8899 | 0.8812 | 0.9122 | 0.9405 | 0.9578 | 1.3029 |
> | CHEMBL234_Ki | 0.6316 | **0.5979** | 0.6409 | 0.6545 | 0.6557 | 0.7028 | 0.6995 | 0.7103 | 0.7418 | 0.7788 | 1.0602 |
> | CHEMBL235_EC50 | **0.7544** | 0.7642 | 0.7766 | 0.7793 | 0.7849 | 0.7946 | 0.8191 | 0.8318 | 0.8541 | 0.9047 | 1.1460 |
> | CHEMBL236_Ki | 0.7850 | 0.7958 | **0.7592** | 0.8169 | 0.8329 | 0.8165 | 0.7980 | 0.8299 | 0.9103 | 0.9160 | 1.3214 |
> | CHEMBL237_EC50 | **0.7668** | 0.8026 | 0.8797 | 0.8174 | 0.8654 | 0.8735 | 0.8544 | 1.0403 | 0.9487 | 0.9318 | 1.3491 |
> | CHEMBL237_Ki | 0.7833 | **0.7567** | 0.7876 | 0.7953 | 0.8238 | 0.8103 | 0.8101 | 0.8311 | 0.8438 | 0.8655 | 1.4205 |
> | CHEMBL238_Ki | 0.7293 | 0.7264 | **0.6958** | 0.7469 | 0.7385 | 0.7359 | 0.7650 | 0.7441 | 0.8308 | 0.8673 | 1.1805 |
> | CHEMBL239_EC50 | **0.7919** | 0.8019 | 0.8391 | 0.8229 | 0.9106 | 0.8636 | 0.8393 | 0.8674 | 0.9296 | 0.9480 | 1.1795 |
> | CHEMBL244_Ki | 0.7516 | 0.7683 | **0.7485** | 0.8361 | 0.8343 | 0.8591 | 0.8153 | 0.9347 | 0.9104 | 0.9900 | 1.5392 |
> | CHEMBL262_Ki | **0.7019** | 0.7118 | 0.9097 | 0.8037 | 0.8611 | 0.9355 | 0.8792 | 0.9378 | 0.8750 | 0.9536 | 1.1605 |
> | CHEMBL264_Ki | **0.6715** | 0.6957 | 0.7051 | 0.7159 | 0.7186 | 0.7241 | 0.7499 | 0.7747 | 0.7897 | 0.8439 | 1.0153 |
> | CHEMBL2835_Ki | 0.7953 | 0.8506 | 0.8178 | **0.7563** | 0.7796 | **0.7499** | 0.8235 | 0.9362 | 0.9170 | 0.9794 | 1.3259 |
> | CHEMBL287_Ki | 0.7977 | **0.7356** | 0.8150 | 0.7663 | 0.7269 | 0.7459 | 0.7571 | 0.8662 | 0.8733 | 0.8640 | 1.1170 |
> | CHEMBL2971_Ki | 0.7777 | **0.7058** | 0.7589 | 0.7698 | 0.7893 | 0.8511 | 0.7991 | 0.8850 | 0.8311 | 0.8530 | 1.3053 |
> | CHEMBL3979_EC50 | **0.6536** | 0.7087 | 0.7653 | 0.7025 | 0.7149 | 0.6898 | 0.7153 | 0.7988 | 0.7878 | 0.7956 | 1.0032 |
> | CHEMBL4005_Ki | **0.7122** | 0.7400 | 0.7277 | 0.7833 | 0.7735 | 0.7668 | 0.7413 | 0.8503 | 0.8006 | 0.8305 | 1.0877 |
> | CHEMBL4203_Ki | 1.1768 | 1.1417 | **1.0360** | 1.2305 | 1.0937 | 1.2276 | 1.2689 | 1.1498 | 1.1955 | 1.2026 | 1.0985 |
> | CHEMBL4616_EC50 | 0.7186 | 0.7244 | 0.7425 | **0.6998** | 0.7502 | 0.7698 | 0.7098 | 0.7396 | 0.7609 | 0.7732 | 0.8418 |
> | CHEMBL4792_Ki | **0.6513** | 0.6595 | 0.6818 | 0.7339 | 0.6948 | 0.6968 | 0.7714 | 0.8003 | 0.8613 | 0.8130 | 1.1685 |

---

> > ### Author Response · Authors · 2025-11-22
> >
> > # W3.
> > We acknowledge the reviewer’s concern. While components such as GINE, Chebyshev propagation, and gating have each appeared in prior work, GraphCliff is not an arbitrary assembly of existing modules. Rather, the architecture is directly motivated by the unique structural demands of the activity cliff problem. Activity cliffs impose a dual requirement that standard GNNs and hybrid models do not simultaneously satisfy: extreme sensitivity to local modifications, and at the same time, stable and meaningful long-range propagation of these local changes across the graph. Existing architectures typically provide either strong locality (as in GINE and other MPNNs) or strong global propagation (as in spectral or Transformer-based models), but not both in a controlled manner. As a result, they either oversmooth when deepened or fail to preserve the local discriminability required to detect cliffs.
> >
> > In contrast, GraphCliff is designed as a functionally motivated architecture in which each component is selected for a specific reason: the short-range GINE pathway preserves local atomic sensitivity, the Chebyshev long-range pathway stabilizes broader structural context propagation, and the gating mechanism determines, based on node context, when long-range propagation should be suppressed or amplified. This node-level adaptive control is precisely what the activity-cliff task requires, and it is not provided by existing anti-oversmoothing baselines.
> >
> > Our empirical analyses further support this interpretation. When compared directly against strong anti-oversmoothing baselines such as PairNorm, NodeNorm, DropEdge, GCNII, and APPNP/PPNP, GraphCliff consistently achieves the best average rank in both RMSE and RMSE_cliff. Complementary analyses using MAD and Dirichlet energy in the revised Figure 2 further show that GraphCliff effectively mitigates oversmoothing while maintaining discriminative, activity-aligned representations across layers. Together, these findings indicate that GraphCliff is not an ad hoc combination of modules, but an architecture whose components play clearly differentiated roles aligned with the structural challenges of activity-cliff prediction, particularly the need to adaptively modulate long-range propagation based on local chemical context.
> >
> >
> > # W4.
> > Thank you for pointing this out. We have added a new subsection, “Problem Setup and Notation”, in Section 3.2. The added subsection is highlighted in yellow in the revised manuscript.
> >
> >
> > # Q1.
> >
> > We interpret the reviewer’s question in two complementary ways: why long-range information is needed, and whether a Transformer alone could replace a deep GNN.
> >
> > First, models in molecular prediction must capture the general trend that structurally similar molecules often have similar properties, while also accounting for cases where small local changes interact with distant parts of the molecule to produce different outcomes. This requires propagation of information across the entire graph. In many ChEMBL targets, for example, the functional effect of a local substitution can depend on ring systems or substituents located far away, which shallow GNNs cannot model due to their limited receptive field.
> >
> > Second, using a Transformer alone is also insufficient: global mixing can dilute or overshadow the very local signals that define an activity cliff. Empirically, our GraphTransformer baseline underperforms for this reason (see also m4Bd Q1). An effective architecture must capture long-range dependencies without washing out fine-grained local distinctions.
> >
> > GraphCliff achieves this balance by combining short-range GINE message passing, long-range Chebyshev propagation, and adaptive gating. This allows the model to preserve local perturbations while still incorporating the influence of distant structural context. For these reasons, deep propagation is not optional but necessary for accurately modeling the combination of local and global factors.

---

> ### Author Response · Authors · 2025-11-22
>
> # Q2.
>
> GraphCliff exhibits moderate computational cost relative to strong baselines. Its parameter count (~6M) is larger than lightweight GNN variants but remains far smaller than pretrained hybrid models such as GROVER (48–100M parameters). As summarized in Table 1, end-to-end runtime averaged over all datasets is also reasonable. GraphCliff requires approximately 262 seconds for 100 epochs, while ChemProp requires around 126 seconds for 30 epochs. Inference is similarly efficient at roughly 0.11 seconds per batch.
>
> With respect to scaling behavior, both training and inference time increase only slightly with the Chebyshev order K. As shown in Table 2, when measured on a per-epoch basis, training time increases by approximately 4.5% from K=2 to K=6, and inference time shows a similarly small variation. This mild increase is expected because GraphCliff processes small molecular graphs (typically under 200 nodes), and GPU kernel launch overhead dominates the computation cost.
>
> Graph size also has limited effect on runtime. Table 3 reports per-graph inference time across node-count bins. Even when node counts vary from under 20 nodes to over 150 nodes, inference time stays within a narrow band (approximately 0.015–0.020 seconds per graph), with deviations mainly observed in bins containing very few samples. This pattern reflects that most operations remain memory-bound rather than compute-bound in the small-molecule regime.
>
> Overall, GraphCliff achieves a favorable trade-off between efficiency and accuracy: it is slightly more expensive than shallow GNNs, substantially lighter than pretrained hybrids, and exhibits stable scaling with both graph size and Chebyshev order.
>
> **Table 1. Comparison of parameter sizes and average end-to-end runtime across all datasets.**
>
>
> | Model               | #Params      | Average Runtime (s) |
> |--------------------|-------------|------------------|
> | **GraphCliff**     | 6.02M          | 261.8      |
> | SVM-ECFP          | -       | 1.66        |
> | ChemProp        | 0.31M      | 125.6 |
> | GINE + PairNorm    | 0.68M           | 152.8            |
> | GINE + NodeNorm    | 0.68M           | 176.6            |
> | JK-Net             | 0.81M        | 176.1            |
> | GCNII              | 0.27M           | 173.9            |
> | APPNP              | 0.27M           | 200.0            |
> | PPNP               | 0.27M       | 101.0            |
> | GINE + DropEdge    | 0.68M         | 169.7            |
> | GINE + ResNet      | 0.68M         | 173.8            |
> | GraphCliff w/ JK-Net    | 6.02M          | 153.9            |
> | GraphCliff w/ Residual   | 6.02M             | 174.7            |
>
>
> **Table 2. Mean and standard deviation of per-epoch training and inference time (in seconds) across different Chebyshev orders K.**
> | mid_K | mean train time (s) | std_train_time (s) | mean inference time (s) | std inference time (s) |
> |-------|----------------------|---------------------|---------------------------|--------------------------|
> | 2     | 2.5452               | 1.2114        | 0.1056         | 0.0528                   |
> | 3     | 2.6012               | 1.2997              | 0.1062          | 0.0540                   |
> | 4     | 2.6421               | 1.3245              | 0.1134         | 0.0661                   |
> | 5     | 2.6356               | 1.3248              | 0.1101                    | 0.0582                   |
> | 6     | 2.6610               | 1.3430              | 0.1121                    | 0.0598                   |
>
>
> **Table 3. Mean per-graph inference time (in seconds) and node statistics across graph size bins.**
>
> | node_bin     | mean #nodes | mean time (s) | std time (s) | # samples |
> |--------------|----------------|-------------|-------------|-------------|
> | [8, 31)      | 14.5333        | 0.0161      | 0.0019      | 150         |
> | [31, 55)     | 47.1176        | 0.0160      | 0.0016      | 85          |
> | [55, 78)     | 76.0000        | 0.0160      | 0.0025      | 5           |
> | [78, 102)    | 82.0000        | 0.0157      | 0.0016      | 20          |
> | [102, 126)   | 114.0000       | 0.0154      | 0.0010      | 5           |
> | [126, 149)   | 138.0000       | 0.0166      | 0.0022      | 15          |
> | [149, 173)   | 157.0000       | 0.0208      | 0.0066      | 5           |
> | [173, 196)   | 182.0000       | 0.0168      | 0.0014      | 10          |
> | [196, 220)   |                |             |           | 0           |
> | [220, 244)   | 244.0000       | 0.0165      | 0.0021      | 5       |
>
>
>  # Q4.
>
>  We appreciate the reviewer’s insightful comments. In the datasets where GraphCliff underperformed, the primary contributing factor was reduced local sensitivity, which made it difficult to capture certain fine-grained structural differences relative to ECFP-based models. We have added a detailed analysis together with illustrative figures in **Appendix Section 8.7**. To improve clarity, we will also include an explicit pointer to this section in the main text.

---

> ### Comment · Reviewer_mPRp · 2025-11-24
>
> I thank the authors for their detailed rebuttal and the additional experimental results, which have resolved my technical concerns. However, I still find the architectural contribution to be somewhat limited. While the authors argue that the design is functionally motivated for the activity cliff problem, the solution remains a combination of well-established modules, which are known to help with different aspects of the problem. Given the additional experiments and improvements to the manuscript, I am raising my rating to 4.

---

> > ### Author Response · Authors · 2025-11-25
> >
> > We sincerely thank the reviewer for reconsidering our work and for acknowledging that our additional experiments and revisions have addressed the technical concerns. We would like to offer one final clarification regarding the architectural contribution, as we believe there may be a subtle but important distinction that we have not fully conveyed.
> >
> >
> > We agree that GraphCliff employs well-established modules; GINE for local propagation and Chebyshev polynomials for spectral propagation. However, we respectfully suggest that the contribution lies not in the novelty of the individual components, but in the specific way they are integrated to address a problem that existing architectures cannot solve effectively. The main contribution is the introduction of node-level, scale-adaptive gating that dynamically selects between short-range and long-range propagation within a single message-passing step, enabling each node to adapt its receptive field based on its structural role.
> >
> >
> > To clarify this distinction more concretely, existing approaches relate to GraphCliff along three distinct dimensions: existing gating mechanisms, hybrid GNN–Transformer architectures, and anti-oversmoothing techniques. We therefore summarize our contributions by clarifying how GraphCliff differs from each of these three directions.
> >
> >
> > **Difference from Existing Gating Mechanisms.**
> >
> > We recognize that gating mechanisms have appeared in several prior graph models, and we would like to clarify how GraphCliff's gating differs from these established approaches. In GGS-NN (Li et al., 2015), GRU-style gating is applied across recurrent iterations to stabilize the update of node states. This gating controls how much of the previous hidden state is preserved over time, but it is not designed to regulate different propagation ranges and therefore does not distinguish between short-range and long-range structural signals. GaAN (Zhang et al., 2018) implements head-level gating within multi-head attention, focusing on reducing noisy attention heads rather than modulating propagation scales. GFGN (Jin et al., 2021) also employs gating, but its gates operate at the feature-channel level and do not interact with the graph topology.
> >
> > In contrast, GraphCliff gates the propagation scale by choosing between 1-hop and multi-hop spectral messages, each reflecting different aspects of the graph structure. Because these signals change with a node's structural position, the gate provides topology-aware control at the node level, which existing gating mechanisms do not offer.
> >
> > - Li, Yujia, et al. "Gated graph sequence neural networks." arXiv preprint arXiv:1511.05493 (2015).
> > - Zhang, Jiani, et al. "Gaan: Gated attention networks for learning on large and spatiotemporal graphs." arXiv preprint arXiv:1803.07294 (2018).
> > - Jin, Wei, et al. "Graph feature gating networks." arXiv preprint arXiv:2105.04493 (2021).
> >
> > **Difference from Hybrid GNN–Transformer Architectures.**
> >
> > GraphCliff also differs from hybrid models such as GROVER and GraphTrans in a fundamental way. Hybrid GNN–Transformer architectures typically incorporate a separate Transformer block that computes global attention independently of the local GNN encoder, resulting in dense pairwise mixing rather than controlled multi-hop propagation. These models do not perform node-wise scale gating; rather, they compute node–node attention scores and stack the global module on top of the local encoder.
> >
> > In contrast, GraphCliff modifies the node-update rule directly, integrating short- and long-range propagation within a single message-passing step. This design enables each node to adaptively determine whether local cues or global structural context should dominate its update. We believe the key distinction lies not in simply combining local and global operators, but in GraphCliff's node-specific and scale-specific fusion, which is a form of integration that GNN–Transformer hybrids do not provide in the same manner.
> >
> > **Difference from Anti-Oversmoothing Techniques.**
> >
> > Anti-oversmoothing methods such as PairNorm, NodeNorm, DropEdge, APPNP/PPNP, GCNII, and JK-Net have been effective in preventing representation collapse by normalizing features, injecting residual signals, perturbing edges, or aggregating multiple layers. These techniques typically operate uniformly at the global or layer-wide level, applying the same adjustment to all nodes regardless of their structural role. While effective at reducing overall smoothing, they do not distinguish between nodes that require strong local discrimination, such as those responsible for cliff-defining perturbations, and nodes that benefit from broader structural context. As a result, these approaches primarily adjust the depth of propagation rather than the scale at each node.
> >
> > In contrast, GraphCliff regulates the propagation scale at the node level, which provides the context-dependent behavior needed for activity-cliff prediction.

---

> > > ### Author Response · Authors · 2025-11-25
> > >
> > > **Summary.**
> > >
> > > In summary, we acknowledge that GraphCliff builds on established modules such as GINE and Chebyshev propagation. However, we believe that the way these components are integrated represents a meaningful architectural contribution tailored to the activity-cliff prediction problem. The model introduces a form of per-node, per-layer, scale-adaptive gating that controls how short-hop and multi-hop information are fused directly within the message-passing update. This integration differs from temporal gating, attention-head gating, and stacked global modules in that it embeds scale selection into the propagation rule itself.
> > >
> > > The design is motivated by the specific demands of activity-cliff prediction, which requires simultaneously preserving atom-level sensitivity and maintaining global structure-activity relationship (SAR) consistency. By regulating propagation scale at the node level, GraphCliff aims to provide a form of adaptive behavior that, to our understanding, existing gating architectures, hybrid Transformer models, and anti-oversmoothing methods do not support in the same manner.
> > >
> > >
> > > We hope this clarification better conveys the distinction we are making. If there are any remaining questions or concerns, we would be happy to address them.

---

### Official Review · Reviewer_m4Bd · 2025-10-28

**Soundness:** 3
**Presentation:** 3
**Contribution:** 3
**Rating:** 6
**Confidence:** 4

**Summary:**

This paper proposes GraphCliff, a gated graph neural network that integrates short-range (GINE) and long-range (Chebyshev) filters to capture subtle structural variations, known as activity cliffs, in molecular property prediction. The model explicitly balances local substructural sensitivity and global molecular context, addressing over-smoothing issues in conventional GNNs. Extensive experiments across 30 MoleculeACE datasets and small-scale LSSNS benchmarks demonstrate consistent improvements and enhanced discriminative node embeddings.

**Strengths:**

- The paper tackles an important and challenging domain-specific issue—*activity cliffs*—that conventional GNNs struggle with.
- The proposed gating design effectively balances local and global information, reducing over-smoothing while preserving local sensitivity.
- The experiments are thorough, including 30 datasets, multiple baselines, ablation and interpretability analyses (e.g., Hop-wise sensitivity, Dirichlet energy, …).

**Weaknesses:**

- Although the overall idea of integrating short- and long-range information is reasonable, the novelty of the approach is somewhat limited, as similar hybrid architectures (e.g., GROVER, GraphTrans) have already been proposed. The paper should more clearly articulate how GraphCliff’s gating design provides advantages specific to molecular *activity cliff* prediction.
- Minor issues in figure and notation:
    - Figure 1 (Overall architecture of GraphCliff) is visually unrefined and lacks clear correspondence between visual components and the mathematical formulation, such as X, h, z.
    - Some notations and dimensional definitions are missing or ambiguous

GROVER : Rong, Yu, et al. "Self-supervised graph transformer on large-scale molecular data." *Advances in neural information processing systems* 33 (2020): 12559-12571.

Graphtrans : Wu, Zhanghao, et al. "Representing long-range context for graph neural networks with global attention." *Advances in neural information processing systems* 34 (2021): 13266-13279.

**Questions:**

The paper emphasizes integrating short- and long-range information. However, prior works such as GROVER and GraphTrans have already explored combining local message-passing GNNs with Transformer-based long-range modeling.
- Does it achieve superior results even when compared to such hybrid models?
- What are the expected advantages of such hybrid models in terms of sensitivity, interpretability, and over-smoothing analyses?

---

> ### Author Response · Authors · 2025-11-22
>
> # W1.
>
> We thank the reviewer for raising this important point. While prior hybrid architectures such as GROVER and GraphTrans also combine local and global components, GraphCliff differs fundamentally in both architectural formulation and problem-specific motivation.
>
> **1. Architectural Differences.**
>
> GROVER employs a dyMPN local encoder and a separate Transformer module for global reasoning, requiring large-scale pretraining on 10M molecules and 48–100M parameters, whereas GraphCliff directly restructures message passing itself. By introducing 1-hop GINE as a short-range channel and Chebyshev polynomial propagation as a long-range channel, GraphCliff captures multi-hop dependencies within a unified GNN operator without pretraining. Unlike GraphTrans, which simply stacks a Transformer on top of a GNN with a <CLS> token, GraphCliff modifies the propagation rule at the node-update level. Our gating mechanism adaptively fuses short- and long-range signals per node, enabling the representation to focus on the relevant scale (local vs. global) depending on structural context.
>
> **2. Problem-Specific Design.**
> Existing hybrid models are designed for general purpose molecular representation learning, while GraphCliff is specifically tailored to activity cliff prediction. Distinguishing small atomic perturbations while maintaining global SAR continuity is central to this task. The gating mechanism explicitly balances these goals: it preserves local atom-level sensitivity essential for detecting cliffs, and simultaneously incorporates long-range Chebyshev updates when global structure is more informative.
>
> **3. Interpretability.**
> While both GraphTrans and GraphCliff compute node-level importance weights, their functions differ fundamentally: GraphTrans uses attention to weight interactions between nodes, whereas GraphCliff’s gating modulates the contribution of short-range and long-range propagation channels, highlighting atoms where local perturbations are critical for activity cliffs.
>
> In summary, GraphCliff introduces a unified propagation mechanism, dynamic gating, and task-specific design choices that distinguish it from previous hybrid architectures.
>
>
> # W2.
>
> We appreciate the reviewer’s comments regarding figure clarity and notation. We revised Figure 1 to ensure that each visual component directly corresponds to the mathematical elements. We also added Problem Setup and Notation subsection in section 3.2 and specify tensor dimensionalities in the method section for clearer alignment with the architectural description. These revisions will improve consistency and readability across the manuscript.

---

> ### Author Response · Authors · 2025-11-22
>
> # Q1.
>
> As shown in the table, GraphCliff consistently outperforms both GROVER and GraphTrans across a wide range of CHEMBL datasets. While GraphCliff achieves the best RMSE on every dataset, there are three cases where its RMSE_cliff is slightly higher than that of GROVER models:
>
> • **CHEMBL2034_Ki**
>   – GraphCliff: RMSE_cliff 0.8580
>   – GROVER_base: RMSE_cliff 0.8345 (better)
>
> • **CHEMBL2835_Ki**
>   – GraphCliff: 0.7953
>   – GROVER_base: 0.7744 (better)
>
> • **CHEMBL4203_Ki**
>   – GraphCliff: 1.1768
>   – GROVER_base: 1.0742 (better)
>   – GROVER_large: 1.0565 (better)
>
> These are the only datasets where GraphCliff does not obtain the lowest RMSE_cliff. Deep GNNs tend to collapse to uniform representations, and global Transformers may overshadow important local information. GraphCliff mitigates both issues by dynamically adjusting the contributions of short-range and long-range propagation. Nodes that require sharper local distinctions downweight long-range signals, while others make greater use of broader structural context. This adaptive balancing helps maintain expressive yet stable representations. Overall, the results demonstrate that simply combining GNNs with Transformers is not sufficient, and that GraphCliff’s task-specific propagation design provides consistent and substantial performance gains. For completeness, we will include GraphTrans and GROVER in the full per-dataset comparison table in the revised manuscript.

---

> > ### Author Response · Authors · 2025-11-22
> >
> > **Table 1: Performance Comparison among GraphTrans, GROVER, and GraphCliff**
> > | Dataset Name       | GraphCliff (RMSE/RMSE_cliff)      | GraphTrans (RMSE/RMSE_cliff)        | GROVER_base (RMSE/RMSE_cliff)        | GROVER_large (RMSE/RMSE_cliff)       |
> > |--------------------|------------------------------|---------------------------------|----------------------------------|----------------------------------|
> > | CHEMBL1862_Ki      | **0.7810** / **0.6740**    | 1.4717 / 1.4065     | 1.0264 / 1.0200                  | 1.0473 / 0.9717                  |
> > | CHEMBL1871_Ki      | **0.6279** / **0.7975**      | 1.4495 / 1.7910    | 0.7630 / 1.0915                  | 0.8335 / 1.1063                  |
> > | CHEMBL2034_Ki      | **0.7473** / 0.8580          | 1.8533 / 1.9526   | 0.7815 / **0.8345**              | 0.8197 / 0.8456                  |
> > | CHEMBL2047_EC50    | **0.5924** / **0.6549**      | 1.3639 / 1.5342    | 0.7590 / 0.7392                  | 0.7724 / 0.7329                  |
> > | CHEMBL204_Ki       | **0.6914** / **0.8212**      | 1.3063 / 1.5009     | 1.0615 / 1.1886                  | 1.1073 / 1.2617                  |
> > | CHEMBL2147_Ki      | **0.5600** / **0.5790**      | 1.6622 / 1.8431     | 0.8723 / 0.9109                  | 1.0308 / 0.9899                  |
> > | CHEMBL214_Ki       | **0.6210** / **0.7260**      | 1.1824 / 1.1384                 | 0.8259 / 0.8841                  | 1.0456 / 1.1307                  |
> > | CHEMBL218_EC50     | **0.6974** / **0.7811**      | 1.1514 / 1.2525                 | 0.8548 / 0.8415                  | 0.8424 / 0.8587                  |
> > | CHEMBL219_Ki       | **0.6636** / **0.7435**      | 1.6714 / 1.6429   | 0.8536 / 0.8839     | 1.0437 / 1.0500                  |
> > | CHEMBL228_Ki       | **0.6513** / **0.6739**      | 1.2219 / 1.2524   | 0.9193 / 0.9345    | 0.9041 / 0.9060                  |
> > | CHEMBL231_Ki       | **0.7080** / **0.8660**          | 1.3378 / 1.2956      | 0.9542 / 0.9576                  | 0.9736 / 0.9418              |
> > | CHEMBL233_Ki       | **0.7860** / 0.8803          | 1.2141 / 1.2698    | 1.0026 / 1.0392                  | 1.2364 / 1.2553                  |
> > | CHEMBL234_Ki       | **0.6175** / **0.6316**      | 1.1861 / 1.1466                 | 0.9636 / 0.9056                  | 1.0231 / 1.0138                  |
> > | CHEMBL235_EC50     | **0.6504** / **0.7544**      | 1.1308 / 1.3455                 | 0.8132 / 0.9182                  | 0.8241 / 0.9246                  |
> > | CHEMBL236_Ki       | **0.6965** / **0.7850**      | 1.2163 / 1.3333                 | 1.0055 / 1.0278                  | 1.0207 / 1.0961                  |
> > | CHEMBL237_EC50     | **0.7075** / **0.7668**      | 1.4189 / 1.4467                 | 1.0624 / 1.0696                  | 1.0854 / 1.1021                  |
> > | CHEMBL237_Ki       | **0.7046** / **0.7833**      | 1.4224 / 1.5063                 | 1.0947 / 1.1121                  | 1.0363 / 1.0702                  |
> > | CHEMBL238_Ki       | **0.5966** / **0.7293**      | 1.1488 / 1.3004                 | 0.8687 / 0.9169                  | 0.9942 / 1.0283                  |
> > | CHEMBL239_EC50     | **0.6701** / **0.7919**      | 1.2363 / 1.5022                 | 0.8651 / 0.9888                  | 1.0725 / 1.1241                  |
> > | CHEMBL244_Ki       | **0.6683** / **0.7516**      | 1.4521 / 1.4311                 | 1.0942 / 1.0873                  | 1.2766 / 1.2638                  |
> > | CHEMBL262_Ki       | **0.7523** / **0.7019**      | 1.1209 / 1.1434                 | 0.9039 / 0.9977                  | 0.9045 / 0.9916                  |
> > | CHEMBL264_Ki       | **0.6189** / **0.6715**      | 1.3226 / 1.4438                 | 0.9726 / 0.9946                  | 0.9520 / 1.0046                  |
> > | CHEMBL2835_Ki      | **0.3959** / 0.7953          | 1.1739 / 1.8163                 | 0.4580 / **0.7744**              | 0.4683 / 0.8275                  |
> > | CHEMBL287_Ki       | **0.7055** / **0.7977**      | 1.0453 / 1.0965                 | 0.8230 / 0.8870                  | 0.8689 / 0.9457                  |
> > | CHEMBL2971_Ki      | **0.6154** / **0.7777**      | 2.2307 / 1.8546                 | 0.7740 / 0.7891                  | 0.9376 / 0.9507                  |
> > | CHEMBL3979_EC50    | **0.6226** / **0.6536**      | 1.1292 / 1.0235                 | 0.8115 / 0.8202                  | 0.8211 / 0.8466                  |
> > | CHEMBL4005_Ki      | **0.6174** / **0.7122**      | 1.1866 / 1.1346                 | 0.8401 / 0.9039                  | 0.8612 / 0.9224                  |
> > | CHEMBL4203_Ki      | **0.9000** / 1.1768          | 1.0695 / 1.0509                 | 0.9263 / **1.0742**              | 0.9733 / **1.0565**              |
> > | CHEMBL4616_EC50    | **0.6342** / **0.7186**      | 1.0185 / 1.0174                 | 0.7758 / 0.7953                  | 0.7817 / 0.7797                  |
> > | CHEMBL4792_Ki      | **0.6351** / **0.6513**      | 1.2683 / 1.2347                 | 0.8415 / 0.8384                  | 0.9262 / 0.9393                  |

---

> ### Author Response · Authors · 2025-11-22
>
> # Q2.
>
> GraphCliff’s hybrid design yields several advantages tailored to activity cliff prediction:
>
> **Local Sensitivity.**
> Activity cliffs arise from minimal structural changes. Standard GNNs smooth these differences away, and pure Transformers dilute local signals via global mixing. GraphCliff’s short-range channel preserves local distinctions and the gating mechanism selectively amplifies local information when cliff-like perturbations are detected.
>
> **Interpretability.**
> Whereas Transformer provides global attention scores, GraphCliff produces atom-level gating weights that directly quantify each atom’s reliance on long-range or short-range information. This results in interpretable attribution maps that correctly identify the atoms responsible for activity cliffs, offering more chemically meaningful insight.
>
> **Over-smoothing Mitigation.**
> Deep GNNs collapse to uniform embeddings, while global Transformers may dominate the representation and suppress important local structural cues. GraphCliff avoids both extremes by using its gating mechanism to dynamically balance short-range and long-range information. Nodes that require fine-grained local discrimination naturally downweight long-range updates, whereas nodes for which broader structural context is useful place more emphasis on Chebyshev propagation. This adaptivity prevents representational collapse and produces more stable and reliable predictions.
>
> In summary, GraphCliff improves local sensitivity, interpretability, and robustness against over-smoothing in a manner directly aligned with the challenges of activity-cliff prediction.

---

### Official Review · Reviewer_b6d2 · 2025-10-30

**Soundness:** 3
**Presentation:** 3
**Contribution:** 2
**Rating:** 6
**Confidence:** 3

**Summary:**

This paper prosposes GraphClif,  a short–long range gating mechanism to explicitly integrate local substructural sensitivity and global molecular context, mitigating over-smoothing while preserving expressive molecular graph representations.

**Strengths:**

1. The empirical results are strong and were done on 30 benchmarks, nonetheless all of them stem from the same dataset.
2. The visualization obtained from the gating mechanism seems to provide interesting insights that align with domain priors
3. The approach to combine high and low frequency signals with gating is intuitive and simple.
4. The authors provide ablation studies.

**Weaknesses:**

1. The approach is tailored and demonstrated to molecules, and it is not clear whether other domain can benefit from it.
2. The empirical evaluation although very extensive, focuses only on ChEMBL, and it remains unknown if this method is also beneficial to other domains or datasets. Evaluating it on other diverse benchmarks from other domains and other tasks may be more convincing on the merits of this work.
3. Based on the two above comments, it is possible the contribution is incremental as it is beneficial only for very specific tasks and types of data. Nonetheless it is possible that this problem of its own is important enough to justify a tailored architecture. As I am not from the molecular field, I lack the ability to judge on the importance of this problem of its own, but rather commenting on the broad contribution of the method to the graph community.
4. One thing that can strengthen the method is some theoretical example where it is provable that without this combination of high and low rank, the task is unrealizable e.g. with 1-WL GNN, but it is realizable with your method. This would motivate the generality of tis approach to solve general cases as shown here empirically, from the theoretical side.

**Questions:**

1. Could you provide other examples rather than molecular predictions where this approach is critical ?

---

> ### Author Response · Authors · 2025-11-22
>
> # W1, W2, W3, and Q1.
>
> We appreciate the reviewer’s thoughtful and constructive comments.
> Our focus on the molecular domain is an intentional choice. The activity cliff problem possesses characteristics unique to molecular science. Activity cliffs describe phenomena where structurally very similar molecules exhibit dramatic differences in biological activity, representing important exceptions that violate the fundamental assumption of QSAR modeling (similar structure → similar activity). We have not found problems with similar characteristics to activity cliffs in other domains such as social networks, knowledge graphs, or recommendation systems. We had difficulty identifying other domains where small structural modifications result in large shifts in target properties, or where such exceptional cases offer insights of comparable significance.
>
> Although specialized to the molecular domain, the importance of this problem itself is substantial. This problem is directly related to core drug discovery processes. Hit-to-lead optimization is the process of optimizing initially discovered active compounds (hits) into drug candidates (leads). Activity cliffs reveal which structural changes dramatically increase or decrease activity, providing essential guidance for molecular design that maximizes efficacy. A single atom or functional group change can result in a 10-fold or greater difference in potency, and such information is critical for efficient lead optimization.
>
> The industrial impact of solving this problem is significant. Accurate prediction of activity cliffs through virtual screening before experimentation allows early elimination of high-failure-probability candidates and concentration on promising compounds, resulting in substantial cost and time savings. Drug development takes an average of over 10 years and costs approximately $2.6 billion (DiMasi et al., 2016), and improved accuracy in activity cliff prediction can significantly reduce these costs and timelines.
>
> Nevertheless, our methodological contributions have broader applicability. If problems are discovered in other domains with characteristics such as cases where minor differences within graph structures have large impacts, cases requiring sensitive modeling of local structural changes, or cases needing to handle exceptional cases in similarity-based prediction, our approach could be effectively applied.
>
> DiMasi, Joseph A., Henry G. Grabowski, and Ronald W. Hansen. "Innovation in the pharmaceutical industry: new estimates of R&D costs." Journal of health economics 47 (2016): 20-33.
>
>
> # W4.
>
> We appreciate for this valuable suggestion. To address this point, we have added a new subsection in the **Appendix Section 8.6** that provides both a theoretical example and empirical illustrations of cases where 1-WL GNNs are unable to distinguish activity-cliff pairs. These examples show that when two molecules differ only by a localized substitution that leaves their neighborhood multisets unchanged, standard message-passing GNNs produce identical updates and cannot capture the resulting large activity difference.

---

### Official Review · Reviewer_f4qn · 2025-10-31

**Soundness:** 2
**Presentation:** 3
**Contribution:** 2
**Rating:** 4
**Confidence:** 2

**Summary:**

This paper aims to solve the discontinuity problem known as the activity cliff, where small structural differences significantly impact activity in QSAR. This paper proposes a new graph neural network called GraphCliff. The proposed method is inspired by StripedHyena2 and integrates local interactions with global interactions using gating mechanisms. The proposed method is applied to 39 QSAR tasks, including those addressing the activity cliff problem, and its predictive performance is evaluated. Additionally, ablation studies are conducted to analyze the impact of local structure on activity cliffs.

**Strengths:**

1. (Originality) As noted in this paper, methods for modeling long-range dependencies have been studied in several fields, such as genome language models. Approaches integrating local and global interactions have been studied, including the Hyena Hierarchy. However, to my knowledge, this paper is the first to apply this idea to the long-range dependency problem in graph learning.
2. (Quality) The paper deals with the important and specific problem of the activity cliff. Furthermore, the numerical experiments cover a wide range of experimental settings, including 16 baseline methods and 39 datasets, which strengthens the credibility of its claims.
3. (Clarity) The writing is clear. The paper's structure is appropriate, and the explanations in each section are straightforward. I had no significant difficulty in understanding the paper's main points.

**Weaknesses:**

1. The discussion of the numerical experiment results has room for improvement. Specifically, I question whether the presentation of Table 1 is appropriate. This table only highlights the datasets where the proposed method achieves the highest accuracy. Results for the remaining datasets are provided in the appendix. However, if I do not miss any information, no validation or discussion regarding them is presented.
2. While the paper claims the proposed method achieves the best overall performance (L.309), the basis for this claim is unclear. This claim should be substantiated through a quantitative evaluation using all 39 datasets.
3. Section 4 provides a detailed analysis of existing methods other than the proposed one (L.310--329). While this analysis is valuable, it deviates from the main focus of this paper, that is, validating the accuracy of the proposed method, and is therefore less important.
4. If I have not missed any information, the method for determining hyperparameters is not described.

**Questions:**

1. I would like to clarify the basis for claiming that the proposed method achieved the best overall performance.
2. I would like the authors to deepen the analysis of the causes for the poor prediction accuracy of the proposed method on certain datasets.

**Details Of Ethics Concerns:**

N.A.

---

> ### Author Response · Authors · 2025-11-22
>
> # W1 and Q2.
>
> We appreciate the reviewer’s insightful comments. In the datasets where GraphCliff underperformed, the primary factor was limited local sensitivity, which made certain fine-grained structural differences harder to capture compared to ECFP-based models. A more detailed analysis, along with supporting figures, has been added to **Section 5.3**.
>
> # W2 and Q1.
>
> We appreciate your feedback. The claim of "best overall performance" (L.309) was based on achieving the top rank when averaging the rankings across 30 datasets, where we evaluated the rank on individual datasets. We acknowledge that this phrasing may have been unclear and could lead to misunderstanding. We revised the statement to clarify that our method "achieves the top rank based on average rankings across 30 datasets" to better reflect the evaluation methodology and avoid ambiguity.
>
> # W3.
>
> Thank you for your thoughtful feedback. We agree that, although the detailed analysis of existing baseline methods in Section 4 (L.310–329) provides useful background, it is not essential to the core validation of our proposed method. To improve the clarity and focus of the main text, we will relocate this content to the Appendix while still making it available for readers who wish to refer to it.
>
> # W4.
>
> Thank you for the reviewer’s comment. We have clarified our hyperparameter tuning procedure and will add this information to the revised manuscript. Specifically, we performed grid search over the ranges summarized in the table below and selected the final hyperparameters based on validation performance.
>
> **Table 1. Hyperparameter tuning search space and final selected values**
> | **Category**        | **Hyperparameter** | **Search Space**            | **Final Value** |
> |---------------------|--------------------|-----------------------------|------------------|
> | **Training setup**  | Batch size         | {64, 128, 256}              | 128              |
> |                     | Epochs             | {50, 100, 150, 200}         | 100              |
> |                     | Learning rate      | {5e-4, 1e-4, 5e-5}          | 1e-4             |
> |                     | Patience (Early Stop)      | {0, 5, 10, 15, 20}          | 15               |
> | **Model**  | Hidden size        | {128, 256, 512}             | 256              |
> |                     | Number of layers   | {1, 2, 3, 4, 5, 6}          | 3                |
> |   | Long-range order   | {2, 3, 4, 5, 6}             | 3                |
> |                     | Dropout rate       | {0, 0.1, 0.2}               | 0                |

---

> > ### Comment · Reviewer_f4qn · 2025-11-23
> >
> > I thank the authors for their detailed responses and the planned revisions to the manuscript. The rebuttal has mostly cleared up the technical ambiguities in the paper. I will re-read the updated manuscript and take all these clarifications into account to decide my final recommendation. Let me ask questions to the authors if necessary.

---

> > > ### Author Response · Authors · 2025-11-25
> > >
> > > We sincerely thank the reviewer for the thoughtful feedback. We would like to share the key updates we have made to the manuscript based on the reviewers' comments:
> > >
> > > **Section 3 (Method):**
> > > - Aligned notation in Figure 1 with the equations for consistency
> > > - Added detailed notation and clarification in Section 3.2 (Problem Setup and Notation)
> > >
> > > **Section 4 (Results):**
> > > - Updated Table 1 to include anti-oversmoothing methods (PairNorm, NodeNorm, DropEdge, APPNP, GCNII, JK-Net) for more comprehensive comparison
> > >
> > > **Section 5.2 (Analysis):**
> > > - Revised analysis to include additional baseline models
> > > - Replaced previous metric with MAD (Mean Absolute Deviation) for clearer interpretation
> > >
> > > **Section 5.3 (Error Analysis):**
> > > - Added new error analysis subsection examining why GraphCliff shows lower performance in some cases
> > >
> > > **Appendix 8.3:**
> > > - Added detailed insights and interpretation of the results across all 30 datasets
> > >
> > > **Appendix 8.6:**
> > > - Added theoretical and empirical examples illustrating the limitations of 1-WL GNNs in activity-cliff prediction
> > >
> > > We remain available to answer any additional questions you may have.

---

### Author Response · Authors · 2025-12-01
**Reviewer Comments Compilation**

We sincerely thank all reviewers for their thoughtful and constructive feedback.  We have carefully revised the manuscript according to the suggestions, and the key updates are summarized as follows.

**Section 3 (Method)**
* Aligned the notation used in Figure 1 with the mathematical formulation for clarity.
* Added missing notation and improved explanations in Section 3.2 (Problem Setup and Notation).

**Section 4 (Results)**
* Updated Table 1 to include additional anti-oversmoothing methods (PairNorm, NodeNorm, DropEdge, APPNP, GCNII, JK-Net) for a more comprehensive comparison.

**Section 5.2 (Analysis of over-smoothing mitigation in GraphCliff)**
* Expanded the analysis to incorporate additional baseline models.
* Replaced the previous metric with MAD (Mean Absolute Deviation) for clearer interpretation of smoothing behavior.

**Section 5.3 (Error Analysis)**
* Added a new subsection analyzing failure cases and explaining why GraphCliff underperforms on certain datasets.

**Appendix 8.3 (All results of 30 datasets in MoleculeACE)**
* Added detailed insights and interpretation for all 30 MoleculeACE datasets.

**Appendix 8.6 (Theoretical and empirical examples of 1-WL GNN limitations)**
* Added theoretical and empirical examples illustrating the limitations of 1-WL GNNs and how GraphCliff overcomes these limitations in activity-cliff prediction.

Below, we continue with the detailed, comment-by-comment responses.

---

> ### Author Response · Authors · 2025-12-01
> **Methods**
>
> ## 1. Limited Technical Contribution
>
> > [reviewer mPRp. W3] Chebyshev long-range propagation, gated/highway mechanisms, and attention pooling have all been in literature for other application; GINE is standard for molecular graphs. The specific combination is practical but feels like engineering glue more than a new principle.
>
>
> We agree that GraphCliff employs well-established modules; GINE for local propagation and Chebyshev polynomials for spectral propagation. However, we respectfully suggest that the contribution lies not in the novelty of the individual components, but in the specific way they are integrated to address a problem that existing architectures cannot solve effectively. The main contribution is the introduction of node-level, scale-adaptive gating that dynamically selects between short-range and long-range propagation within a single message-passing step, enabling each node to adapt its receptive field based on its structural role.
>
> To clarify this distinction more concretely, existing approaches relate to GraphCliff along three distinct dimensions: existing gating mechanisms, hybrid GNN–Transformer architectures, and anti-oversmoothing techniques. We therefore summarize our contributions by clarifying how GraphCliff differs from each of these three directions.
>
> **Difference from Existing Gating Mechanisms.**
>
> We recognize that gating mechanisms have appeared in several prior graph models, and we would like to clarify how GraphCliff's gating differs from these established approaches. In GGS-NN (Li et al., 2015), GRU-style gating is applied across recurrent iterations to stabilize the update of node states. This gating controls how much of the previous hidden state is preserved over time, but it is not designed to regulate different propagation ranges and therefore does not distinguish between short-range and long-range structural signals. GaAN (Zhang et al., 2018) implements head-level gating within multi-head attention, focusing on reducing noisy attention heads rather than modulating propagation scales. GFGN (Jin et al., 2021) also employs gating, but its gates operate at the feature-channel level and do not interact with the graph topology.
>
> In contrast, GraphCliff gates the propagation scale by choosing between 1-hop and multi-hop spectral messages, each reflecting different aspects of the graph structure. Because these signals change with a node's structural position, the gate provides topology-aware control at the node level, which existing gating mechanisms do not offer.
>
> Li, Yujia, et al. "Gated graph sequence neural networks." arXiv preprint arXiv:1511.05493 (2015).
> Zhang, Jiani, et al. "Gaan: Gated attention networks for learning on large and spatiotemporal graphs." arXiv preprint arXiv:1803.07294 (2018).
> Jin, Wei, et al. "Graph feature gating networks." arXiv preprint arXiv:2105.04493 (2021).
>
> **Difference from Hybrid GNN–Transformer Architectures.**
>
> GraphCliff also differs from hybrid models such as GROVER and GraphTrans in a fundamental way. Hybrid GNN–Transformer architectures typically incorporate a separate Transformer block that computes global attention independently of the local GNN encoder, resulting in dense pairwise mixing rather than controlled multi-hop propagation. These models do not perform node-wise scale gating; rather, they compute node–node attention scores and stack the global module on top of the local encoder.
>
> In contrast, GraphCliff modifies the node-update rule directly, integrating short- and long-range propagation within a single message-passing step. This design enables each node to adaptively determine whether local cues or global structural context should dominate its update. We believe the key distinction lies not in simply combining local and global operators, but in GraphCliff's node-specific and scale-specific fusion, which is a form of integration that GNN–Transformer hybrids do not provide in the same manner.
>
> **Difference from Anti-Oversmoothing Techniques.**
>
> Anti-oversmoothing methods such as PairNorm, NodeNorm, DropEdge, APPNP/PPNP, GCNII, and JK-Net have been effective in preventing representation collapse by normalizing features, injecting residual signals, perturbing edges, or aggregating multiple layers. These techniques typically operate uniformly at the global or layer-wide level, applying the same adjustment to all nodes regardless of their structural role. While effective at reducing overall smoothing, they do not distinguish between nodes that require strong local discrimination, such as those responsible for cliff-defining perturbations, and nodes that benefit from broader structural context. As a result, these approaches primarily adjust the depth of propagation rather than the scale at each node.
>
> In contrast, GraphCliff regulates the propagation scale at the node level, which provides the context-dependent behavior needed for activity-cliff prediction.

---

> > ### Author Response · Authors · 2025-12-01
> > **Methods (continued)**
> >
> > **Summary.**
> >
> > In summary, we acknowledge that GraphCliff builds on established modules such as GINE and Chebyshev propagation. However, we believe that the way these components are integrated represents a meaningful architectural contribution tailored to the activity-cliff prediction problem. The model introduces a form of per-node, per-layer, scale-adaptive gating that controls how short-hop and multi-hop information are fused directly within the message-passing update. This integration differs from temporal gating, attention-head gating, and stacked global modules in that it embeds scale selection into the propagation rule itself.
> >
> > The design is motivated by the specific demands of activity-cliff prediction, which requires simultaneously preserving atom-level sensitivity and maintaining global structure-activity relationship (SAR) consistency. By regulating propagation scale at the node level, GraphCliff aims to provide a form of adaptive behavior that, to our understanding, existing gating architectures, hybrid Transformer models, and anti-oversmoothing methods do not support in the same manner.
> >
> >
> >
> > ## 2. Justification for the Use of Deep GNNs
> >
> > > [reviewer mPRp. Q1] Could you clarify why do deep GNNss are needed for this application ?
> >
> > We interpret the reviewer’s question in two complementary ways: why long-range information is needed, and whether a Transformer alone could replace a deep GNN.
> >
> > First, models in molecular prediction must capture the general trend that structurally similar molecules often have similar properties, while also accounting for cases where small local changes interact with distant parts of the molecule to produce different outcomes. This requires propagation of information across the entire graph. In many ChEMBL targets, for example, the functional effect of a local substitution can depend on ring systems or substituents located far away, which shallow GNNs cannot model due to their limited receptive field.
> >
> > Second, using a Transformer alone is also insufficient: global mixing can dilute or overshadow the very local signals that define an activity cliff. Empirically, our GraphTransformer baseline underperforms for this reason (see also m4Bd Q1). An effective architecture must capture long-range dependencies without washing out fine-grained local distinctions.
> >
> > GraphCliff achieves this balance by combining short-range GINE message passing, long-range Chebyshev propagation, and adaptive gating. This allows the model to preserve local perturbations while still incorporating the influence of distant structural context. For these reasons, deep propagation is not optional but necessary for accurately modeling the combination of local and global factors.

---

> ### Author Response · Authors · 2025-12-01
> **Results**
>
> ## 1. Comparison with Standard Anti-Oversmoothing Baselines
>
> > [reviewer mPRp. W2] Necessity of the mechanism vs. simpler baselines. If over-smoothing were the key issue motivating deep propagation, residual/skip connections, PairNorm/BatchNorm, JK-Net and APPNP/PPNP are standard, lightweight fixes. The paper does not compare GraphCliff against these strong anti over-smoothing baselines or against simple residual GNNs. This weakens the claim that the proposed mechanism is required.
>
> > [reviewer mPRp. Q3] Did you try residual connections to the input or JK aggregation in lieu of gated fusion? Even a small study showing why they underperform would clarify the design choice.
>
>
> We fully agree that it is important to assess whether GraphCliff’s gated fusion is truly necessary beyond standard anti–oversmoothing techniques. To address this, we conducted additional experiments comparing GraphCliff with widely used oversmoothing-mitigation baselines, as well as simplified variants of our model in which the gated fusion is replaced by residual or JK aggregation.
>
> Specifically, we evaluate the following models across the same 30 ChEMBL regression tasks:
>
> * GraphCliff (proposed)
> * GraphCliff w/ JK-Net (replacing gated fusion with JK aggregation)
> * GraphCliff w/ Residual (adding residual connections instead of gating)
> * GINE+PairNorm, GINE+NodeNorm, JK-Net, GINE+Residual, GCNII, GINE+DropEdge, APPNP, PPNP
>
> Below, we summarize the average ranks over all datasets. Also, the complete RMSE and RMSE_cliff tables are provided in the `reviewer-response section`.
>
> | Model                   | Avg. Rank (RMSE) | Avg. Rank (RMSE_cliff) |
> |-------------------------|------------------|--------------------------|
> | GraphCliff              | 1.83             | 2.50                     |
> | GraphCliff w/ JK-Net   | 2.53             | 3.20                     |
> | GraphCliff w/ Residual | 3.80             | 3.90                     |
> | GINE + PairNorm        | 4.40             | 4.93                     |
> | GINE + NodeNorm        | 4.90             | 4.57                     |
> | JK-Net                 | 4.93             | 4.77                     |
> | GINE + Residual        | 6.00             | 5.67                     |
> | GCNII                  | 8.27             | 7.63                     |
> | GINE + DropEdge        | 8.60             | 8.60                     |
> | APPNP                  | 9.73             | 9.50                     |
> | PPNP                   | 11.00            | 10.73                    |
>
> These results show that GraphCliff consistently achieves strong performance in both RMSE and RMSE_cliff. While anti-oversmoothing techniques can achieve comparable performance in some cases, they still fall short of GraphCliff. This suggests that alleviating oversmoothing alone is insufficient for activity cliff prediction. The task requires a node-level mechanism that can adaptively regulate long-range propagation based on each atom’s local structural sensitivity, which is precisely what GraphCliff’s gated fusion is designed to achieve. We believe these additional results clarify the necessity of the proposed mechanism and the limitations of simpler alternatives. We include the full comparison table, incorporating all anti-oversmoothing baselines, in the revised manuscript for completeness.

---

> ### Author Response · Authors · 2025-12-01
> **Analysis**
>
> ## 1. Clarifying Oversmoothing vs. Oversquashing in Section 5.2(Analysis of over-smoothing mitigation in GraphCliff)
>
> > [reviewer mPRp. W1] Section 5.2’s hop-wise sensitivity (perturbation of u affecting v at k hops) and Jacobian spectrum are closer to an over-squashing analysis than smoothing. Although Dirichlet energy is included for smoothing, the text and framing mix these notions, and it’s unclear whether the proposed gains primarily combat squashing (limited receptive field / information bottlenecks) or smoothing (Laplacian averaging).
>
> We agree that our previous framing was misleading. Our primary concern is oversmoothing in the context of activity cliff prediction, where the task is to distinguish subtle local substructural differences between structurally similar molecules. Since this is fundamentally a local feature discrimination problem rather than a long-range information propagation issue, oversquashing is not the primary bottleneck in this particular problem. To better align our analysis with the actual problem, we replaced Section 5.2's hop-wise sensitivity and Jacobian-based metrics with Mean Average Distance (MAD) refer to Chen et al., which directly measure the distinguishability of node representations.
>
> Chen, Deli, et al. "Measuring and relieving the over-smoothing problem for graph neural networks from the topological view." Proceedings of the AAAI conference on artificial intelligence. Vol. 34. No. 04. 2020.
>
> ## 2. Failure Case Analysis
>
> > [reviewer f4qn. W1] The discussion of the numerical experiment results has room for improvement. Specifically, I question whether the presentation of Table 1 is appropriate. This table only highlights the datasets where the proposed method achieves the highest accuracy. Results for the remaining datasets are provided in the appendix. However, if I do not miss any information, no validation or discussion regarding them is presented.
>
> > [reviewer f4qn. Q2] I would like the authors to deepen the analysis of the causes for the poor prediction accuracy of the proposed method on certain datasets.
>
> > [reviewer mPRp. Q4] Could you provide a few negative cases where GraphCliff fails on known cliff pairs and analyze whether the failure is due to insufficient local sensitivity, long-range diffusion, or data sparsity? This would sharpen the pros and cons of GraphCliff
>
> In the datasets where GraphCliff underperformed, the primary factor was limited local sensitivity, which made certain fine-grained structural differences harder to capture compared to ECFP-based models. A more detailed analysis, along with supporting figures, has been added to **Section 5.3 (Error Analysis)**.
>
>
> ## 3. Theoretical Motivation for the High–Low Rank Combination
>
> > [reviewer b6d2. Q4] One thing that can strengthen the method is some theoretical example where it is provable that without this combination of high and low rank, the task is unrealizable e.g. with 1-WL GNN, but it is realizable with your method. This would motivate the generality of tis approach to solve general cases as shown here empirically, from the theoretical side.
>
> To address this point, we have added a new subsection in the **Appendix Section 8.6 (Thoretical and empirical of 1-WL GNN limitations)** that provides both a theoretical example and empirical illustrations of cases where 1-WL GNNs are unable to distinguish activity-cliff pairs. These examples show that when two molecules differ only by a localized substitution that leaves their neighborhood multisets unchanged, standard message-passing GNNs produce identical updates and cannot capture the resulting large activity difference.

---

> > ### Author Response · Authors · 2025-12-01
> > **Writing**
> >
> > ## 1. Clarifications on Figure 1 and Notation
> >
> > > [reviewer m4Bd. W2] Minor issues in figure and notation: a) Figure 1 (Overall architecture of GraphCliff) is visually unrefined and lacks clear correspondence between visual components and the mathematical formulation, such as X, h, z. b)Some notations and dimensional definitions are missing or ambiguous
> >
> > > [reviewr mPRp. W4] Formal problem setup & notations are missing. There is no concise notation paragraph formally defining the input graph G=(V,E), node/edge features, and the prediction problem; the method section jumps directly into components/equations, which hurts clarity.
> >
> > We revised Figure 1 to ensure that each visual component directly corresponds to the mathematical elements. We also added **Problem Setup and Notation subsection in section 3.2** and specify tensor dimensionalities in the method section for clearer alignment with the architectural description.
> >
> >
> > ## 2. Clarification on Overall Performance Claims
> >
> > > [reviewer f4qn. W2] The discussion of the numerical experiment results has room for improvement. Specifically, I question whether the presentation of Table 1 is appropriate. This table only highlights the datasets where the proposed method achieves the highest accuracy. Results for the remaining datasets are provided in the appendix. However, if I do not miss any information, no validation or discussion regarding them is presented.
> >
> > > [reviewer f4qn. Q1] I would like to clarify the basis for claiming that the proposed method achieved the best overall performance.
> >
> > The claim of "best overall performance" (L.309) was based on achieving the top rank when averaging the rankings across 30 datasets, where we evaluated the rank on individual datasets. We acknowledge that the phrasing may have been unclear and could lead to misunderstanding. We revised the statement to clarify that our method "achieves the top rank based on average rankings across 30 datasets" to better reflect the evaluation methodology and avoid ambiguity.
> >
> >
> >
> > ## 3. Relevance of  Detailed Anlaysis of Existing Methods
> >
> > > [reviewer f4qn. W3] Section 4 provides a detailed analysis of existing methods other than the proposed one (L.310--329). While this analysis is valuable, it deviates from the main focus of this paper, that is, validating the accuracy of the proposed method, and is therefore less important.
> >
> > In the revised manuscript, Section 4 has been reorganized. The original content has been moved to **Appendix Section 8.3 (All Results of 30 Datasets in MoleculeACE)**, and Section 4 now focuses on the newly added descriptions of the anti-oversmoothing methods (PairNorm, NodeNorm, DropEdge, APPNP, GCNII, and JK-Net) used in our comparison.

---

> > > ### Author Response · Authors · 2025-12-01
> > > **Additional**
> > >
> > > ## 1. Generalization Beyond Molecular Domains
> > >
> > > > [reviewer b6d2. W1] The approach is tailored and demonstrated to molecules, and it is not clear whether other domain can benefit from it.
> > >
> > > > [reviewr b6d2. W2] The empirical evaluation although very extensive, focuses only on ChEMBL, and it remains unknown if this method is also beneficial to other domains or datasets. Evaluating it on other diverse benchmarks from other domains and other tasks may be more convincing on the merits of this work.
> > >
> > > > [reviewer b6d2. W3] Based on the two above comments, it is possible the contribution is incremental as it is beneficial only for very specific tasks and types of data. Nonetheless it is possible that this problem of its own is important enough to justify a tailored architecture. As I am not from the molecular field, I lack the ability to judge on the importance of this problem of its own, but rather commenting on the broad contribution of the method to the graph community.
> > >
> > > > [reviewer b6d2. Q1] Could you provide other examples rather than molecular predictions where this approach is critical ?
> > >
> > > Our focus on the molecular domain is an intentional choice. The activity cliff problem possesses characteristics unique to molecular science. Activity cliffs describe phenomena where structurally very similar molecules exhibit dramatic differences in biological activity, representing important exceptions that violate the fundamental assumption of QSAR modeling (similar structure → similar activity). We have not found problems with similar characteristics to activity cliffs in other domains such as social networks, knowledge graphs, or recommendation systems. We had difficulty identifying other domains where small structural modifications result in large shifts in target properties, or where such exceptional cases offer insights of comparable significance.
> > >
> > > Although specialized to the molecular domain, the importance of this problem itself is substantial. This problem is directly related to core drug discovery processes. Hit-to-lead optimization is the process of optimizing initially discovered active compounds (hits) into drug candidates (leads). Activity cliffs reveal which structural changes dramatically increase or decrease activity, providing essential guidance for molecular design that maximizes efficacy. A single atom or functional group change can result in a 10-fold or greater difference in potency, and such information is critical for efficient lead optimization.
> > >
> > > The industrial impact of solving this problem is significant. Accurate prediction of activity cliffs through virtual screening before experimentation allows early elimination of high-failure-probability candidates and concentration on promising compounds, resulting in substantial cost and time savings. Drug development takes an average of over 10 years and costs approximately $2.6 billion (DiMasi et al., 2016), and improved accuracy in activity cliff prediction can significantly reduce these costs and timelines.
> > >
> > > Nevertheless, our methodological contributions have broader applicability. If problems are discovered in other domains with characteristics such as cases where minor differences within graph structures have large impacts, cases requiring sensitive modeling of local structural changes, or cases needing to handle exceptional cases in similarity-based prediction, our approach could be effectively applied.
> > >
> > > DiMasi, Joseph A., Henry G. Grabowski, and Ronald W. Hansen. "Innovation in the pharmaceutical industry: new estimates of R&D costs." Journal of health economics 47 (2016): 20-33.

---

> ### Author Response · Authors · 2025-12-01
> **Additional (continued)**
>
> ## 2. Distinction from Existing Hybrid Models (GROVER, GraphTrans)
>
> > [reviewer m4Bd. W1] Although the overall idea of integrating short- and long-range information is reasonable, the novelty of the approach is somewhat limited, as similar hybrid architectures (e.g., GROVER, GraphTrans) have already been proposed. The paper should more clearly articulate how GraphCliff’s gating design provides advantages specific to molecular activity cliff prediction.
>
> > [reviewer m4Bd. Q1 and Q2] The paper emphasizes integrating short- and long-range information. However, prior works such as GROVER and GraphTrans have already explored combining local message-passing GNNs with Transformer-based long-range modeling. 1) Does it achieve superior results even when compared to such hybrid models? 2) What are the expected advantages of such hybrid models in terms of sensitivity, interpretability, and over-smoothing analyses?
>
>
>
>
> A detailed response has been provided in the corresponding `reviewer-response` section. Here, we summarize the key points. GraphCliff differs fundamentally from existing hybrid models in architectural design, task-specific motivation, and interpretability. Unlike GROVER and GraphTrans which combine separate local and global modules, GraphCliff restructures message passing itself by integrating a short-range GINE channel and a long-range Chebyshev channel within a unified propagation operator. More importantly, the node-level gating mechanism adaptively modulates the contribution of both channels based on local structural context, providing advantages specifically aligned with activity cliff prediction. This design yields three practical benefits: i) Improved local sensitivity to subtle atomic perturbations, ii) Direct interpretability through atom-wise gating coefficients, and iii) Robustness against over-smoothing by balancing short- and long-range updates. Empirically, GraphCliff shows consistent performance advantages over both GROVER and GraphTrans across the CHEMBL benchmarks. A complete comparison table, including GROVER and GraphTrans, is provided in the `reviewer-response` section.
>
>
> ## 3. Runtime, Memory Cost, and Scalability Analysis
>
> > [reviwer mPRp. Q2] What are the training/inference runtime and memory costs of GraphCliff relative to strong baselines? How do these scale with graph size and Chebyshev order?
>
> GraphCliff exhibits moderate computational cost relative to strong baselines. Its parameter count (~6M) is larger than lightweight GNN variants but remains far smaller than pretrained hybrid models such as GROVER (48–100M parameters). As summarized in Table 1, end-to-end runtime averaged over all datasets is also reasonable. GraphCliff requires approximately 262 seconds for 100 epochs, while ChemProp requires around 126 seconds for 30 epochs. Inference is similarly efficient at roughly 0.11 seconds per batch. The training and inference runtime, memory costs, and their scaling with graph size and Chebyshev order are provided in the `reviewer-response` section.
>
>
> ## 4. Hyperparameter Selection
>
> > [reviewer f4qn. W4] If I have not missed any information, the method for determining hyperparameters is not described.
>
> We have clarified our hyperparameter tuning procedure and will add this information to the revised manuscript. Specifically, we performed grid search over the ranges summarized in the table below and selected the final hyperparameters based on validation performance.
>
> ### Table 1. Hyperparameter tuning search space and final selected values
>
> | Category        | Hyperparameter      | Search Space                | Final Value |
> |-----------------|----------------------|------------------------------|-------------|
> | Training setup  | Batch size          | {64, 128, 256}              | 128         |
> |                 | Epochs              | {50, 100, 150, 200}         | 100         |
> |                 | Learning rate       | {5e-4, 1e-4, 5e-5}          | 1e-4        |
> |                 | Patience (Early Stop)| {0, 5, 10, 15, 20}         | 15          |
> | Model           | Hidden size         | {128, 256, 512}             | 256         |
> |                 | Number of layers    | {1, 2, 3, 4, 5, 6}          | 3           |
> |                 | Long-range order    | {2, 3, 4, 5, 6}             | 3           |
> |                 | Dropout rate        | {0, 0.1, 0.2}               | 0           |

---

### Meta-Review · Area_Chair_27tj · 2026-01-06

**Summary:**

This paper introduces GraphCliff, a molecular GNN designed for activity cliff prediction, where tiny structural changes can cause large shifts in activity. The core idea is to explicitly combine a short-range GINE channel with a long-range Chebyshev (spectral) propagation channel, and let a learnable sigmoid gate decide—at the node level—how much local vs. global information to use. The reviewers generally agree the problem is meaningful for QSAR and medicinal chemistry, and several of them found the approach intuitive and practically useful, with thorough experiments and interpretability hooks (e.g., gating visualizations).

**Reviewer Concerns:**

The most critical reviewer argued the paper initially lacked key comparisons against strong anti-oversmoothing baselines / simple residual variants, and also flagged clarity issues in the formal setup and framing. In response, the authors added those comparisons and provided a more complete table of results showing GraphCliff ranking strongly on both RMSE and RMSE on cliff subsets, and they also revised the analysis section to avoid conflating oversquashing-style diagnostics with oversmoothing, switching to a more directly interpretable measure (MAD).

**Reviewer Scores:**

Reviewer f4qn (score 4): likely change, since they explicitly say the rebuttal cleared most technical ambiguities and they plan to re-evaluate with the revisions in mind.

Reviewer b6d2: likely no change. Their main hesitations are about scope/generalization beyond molecules and how broadly the contribution transfers, which discussion clarifications don’t fully resolve.

Reviewer m4Bd: likely no change

Reviewer mPRp (score 2): likely change

---

### Decision · Program_Chairs · 2026-01-26

Reject